# Certificates for Complex-Compatible Learned Cochain Laplacians

Nivar Anwer [1]   Marien Chenaud [2,3]   David Elizondo [4]

## Abstract

Learning mesh-based operators from data can match training objectives while implicitly violating algebraic consistency constraints that classical discretizations satisfy by construction. Such violations can introduce near-kernel directions, degrade conditioning as resolution increases, and distort the low-frequency spectral structure on which downstream solvers and diagnostics rely. This work introduces a low-overhead compatibility certificate for learned operator pairs, together with a closed-form projection that maps a learned pair to its Frobenius-nearest chain-compatible operator. The certificate provides an explicit distance-to-compatibility and yields perturbation bounds for the discrete operator. These bounds imply stability guarantees for elliptic solves and for low-frequency spectral counts, provided a spectral gap separates the kernel from the rest of the spectrum and boundary treatments are well posed. Experiments on standard elliptic problems show that defect-aware training prevents condition-number blow-up at higher resolutions, improves robustness under mesh and topological distribution shifts, and maintains predictive accuracy relative to unconstrained learning. Overall, these results support the use of deployment-neutral, computable algebraic consistency checks to detect and control failure modes that are not revealed by loss values alone.

[1]School of Computer Science, Georgia Institute of Technology, Atlanta, GA, USA [2]MICS, CentraleSupélec, Université Paris-Saclay, Gif-sur-Yvette, France [3]Transvalor SA, 955 Avenue Roumanille, Biot, 06904 Biot, France [4]School of Computer Science and Informatics, De Montfort University, Leicester, United Kingdom. Correspondence to: Nivar Anwer <nanwer3@gatech.edu>.

*Proceedings of the 43rd International Conference on Machine Learning*, Seoul, South Korea. PMLR 306, 2026. Copyright 2026 by the author(s).

## 1. Introduction

Modern scientific machine learning increasingly focuses on learning discrete operators defined on graphs, meshes, and higher-order cell complexes, which serve as surrogates for partial differential equation solvers and implicit time-stepping schemes (Kovachki et al., 2023; Li et al., 2023; Wang et al., 2024; Lam et al., 2023; Bi et al., 2023; Zia et al., 2024). Across Poisson/Darcy benchmarks and many implicit time-stepping pipelines, deployment repeatedly applies or inverts a Laplace-type *stiffness operator*, whose low-frequency spectrum governs conditioning, iterative convergence, and the numerical-solver and topological diagnostics used downstream. Motivated by the central role of Laplace-type operators in these pipelines, this work focuses on *learned cochain Laplacians*. These are discrete Hodge Laplacians assembled from learned incidence-like maps and symmetric positive definite metric terms (Definition 3.10), and they form the interface between learned components and numerical PDE solves. In particular, the stability analysis developed below addresses the regime that regularly occurs in practice: cases in which a learned discretisation achieves low prediction loss, yet yields an operator that is numerically fragile when embedded within a linear solve. The considered setting is a learned degree-$0 \rightarrow 1 \rightarrow 2$ cochain complex ($C^0 \xrightarrow{D_0} C^1 \xrightarrow{D_1} C^2$), where the spaces $C^k$ represent discrete $k$-cochains and the operators $D_k$ act as learned incidence maps between them. Compatibility is encoded by the discrete identity $D_1 D_0 = 0$, the counterpart of the continuous relation $\mathrm{curl}, \mathrm{grad} = 0$, and ensures that the learned operators respect the underlying topological structure used in PDE discretisations. In compatible discretisation frameworks such as Finite Element Exterior Calculus (FEEC) and Discrete Exterior Calculus (DEC), operators are organised into cochain complexes that mirror the structure of the underlying differential operators. In this setting, the *cochain-complex condition* is a structural property that underpins stable Hodge decompositions, controls the dimension of operator kernels, and yields well-behaved Laplacians. When this identity is violated, the resulting failure modes can remain largely invisible to standard regression losses. Learned operators may continue to fit training targets while developing spurious near-kernel directions, drifting Laplacian spectra, and severe conditioning instabilities as the resolution changes (McGreivy & Hakim,

2024; Kovachki et al., 2023; Li et al., 2023). Such pathologies can destabilise elliptic solvers and corrupt topology-sensitive low-frequency spectral multiplicities, which are commonly interpreted through Hodge-theoretic lenses (Isufi et al., 2025b; Davies et al., 2023; Zia et al., 2024). The cochain-complex condition $D_1 D_0 = 0$ is automatically satisfied when the coboundary or incidence operators are fixed by the underlying cell complex and only compatible metric terms (e.g., discrete Hodge stars) are learned (Arnold et al., 2010; Desbrun et al., 2005; Isufi et al., 2025b). However, this condition can be violated when incidence-like maps are themselves learned or adapted: for example, to absorb modelling errors or heterogeneity, to transfer across different mesh families or under remeshing, to implement restriction or transport in sheaf-based architectures, or to calibrate boundary regimes. These challenges motivate the development of a quantitative certificate capable of detecting and controlling violations of cochain compatibility (Li et al., 2023; Kovachki et al., 2023; Wang et al., 2024; Hernandez Caralt et al., 2024; Duța et al., 2023; Zia et al., 2024; McGreivy & Hakim, 2024; Isufi et al., 2025b). For a pair of learned operators $D_{0,\phi}, D_{1,\phi}$, this work defines the *cochain-compatibility defect*

$$D_{\text{comp}}(\phi) \;=\; \|D_{1,\phi} D_{0,\phi}\|_F, \qquad (1)$$

which directly measures violation of the cochain-complex condition (Definition 3.8). When $D_{1,\phi}$ is fixed, the Euclidean-orthogonal projection of $D_{0,\phi}$ onto $\ker(D_{1,\phi})$ yields the nearest chain-compatible reference (Definition 4.1). We show that this projection is *Frobenius-nearest* among complexes with fixed $D_{1,\phi}$, so the residual provides an explicit distance-to-compatibility certificate (Theorem 4.2). This defect-to-distance control propagates to *explicit* Laplacian-drift budgets and, under a low-frequency gap in the relevant boundary regime, to solver sensitivity and stable small-eigenvalue counts; these checks are summarised by the operational criterion $\rho(\phi) < 1$ (Sections 4–4.4, Paragraph 4.4).

The projected reference plays a specific analytical role. It does not claim that $\widetilde{D}_{0,\phi}$ is itself the deployed discretisation or that projection solves the learning problem. Rather, it isolates the component of the learned operator pair that is attributable solely to violation of the cochain-complex condition, with $D_{1,\phi}$ and $W_{1,\phi}$ held fixed. This separation is important because it distinguishes two different sources of downstream error: incompatibility, measured by the gap between $D_{0,\phi}$ and $\widetilde{D}_{0,\phi}$, and reference-family deviation, measured later by $\mathcal{R}_E(\phi)$. The projection therefore provides the correct anchor for certification: instability measured relative to $\widetilde{L}_{0,\phi}$ can be attributed to compatibility violation rather than to a change of target discretisation.

**Contributions.**

- **Defect certificate and closed-form nearest compatible reference.** We define $D_{\text{comp}}(\phi) = \|D_{1,\phi} D_{0,\phi}\|_F$ and provide a closed-form orthogonal projection producing $\widetilde{D}_{0,\phi} \in \ker(D_{1,\phi})$ with $D_{1,\phi}$ held fixed (Definitions 3.8–4.1). We show that the resulting residual equals the Frobenius distance to the nearest chain-compatible complex under fixed $D_{1,\phi}$ (Theorem 4.2)

- **Defect-to-Laplacian perturbation bounds and solver stability.** We derive explicit linear-quadratic operator-norm perturbation bounds for the induced Laplacian relative to the projected reference Laplacian, and propagate them to Symmetric Positive Definite (SPD) elliptic solve sensitivity (Theorems 4.4 and C.15).

- **Propagation to PDE error and low-frequency multiplicity stability with an operational criterion.** We establish an elliptic error decomposition that explicitly isolates a defect term, and we provide sufficient conditions to ensure the stability of low-frequency eigenmodes. This framework enables reliable, fixed-resolution computation of Betti numbers even in the presence of a spectral gap (Theorems 5.6 and 4.10) (Isufi et al., 2025b; Davies et al., 2023). We then combine the defect-to-drift budget with a once-estimated spectral margin to formulate a concise stability certificate, $\rho(\phi) < 1$, which can be used both as a training monitor and as a practical check of sufficient conditions (Paragraph 4.4).

- **Shift-focused evaluation.** Finally, we demonstrate through various numerical experiments that controlling the defect prevents conditioning instabilities induced by changes in resolution and enhances robustness under unstructured-mesh variations and topological distribution shifts, all without compromising surrogate accuracy (Section 6).

## 2. Related Work

**Compatible discretisations and discrete differential complexes.** Finite element exterior calculus (FEEC) provides a principled framework for stable mixed discretisations by viewing PDE operators as maps in a Hilbert complex and enforcing *commuting* discretisation/projection operators; in this setting, identities such as $D_{k+1} D_k = 0$ (the discrete analogue of $d \circ d = 0$) are structural constraints tied to stability and consistency rather than incidental algebraic coincidences (Arnold et al., 2006; 2010; Isufi et al., 2025b). Discrete exterior calculus (DEC) offers a complementary algebraic–geometric viewpoint built from incidence operators (coboundaries) and discrete Hodge stars, yielding constructions that preserve the cochain-complex condition by design and connect naturally to discrete differential forms on meshes and cell complexes (Desbrun et al., 2005; Isufi et al., 2025b). Discrete Hodge theory further clarifies the

role of the cochain-complex condition in shaping kernel structure and harmonic representatives, and hence in controlling nullspaces and low-energy modes of Hodge Laplacians (Dodziuk, 1976; Isufi et al., 2025b). This paper adopts the cochain-complex viewpoint but targets the *learning* regime: the incidence-like operators are learned (or modified by optimisation), the identity $D_1 D_0 = 0$ can be violated, and a *quantitative, computable* compatibility certificate is needed to detect and control downstream instability even when the predictive loss is small (McGreivy & Hakim, 2024; Kovachki et al., 2023).

**Topological signal processing and learning on complexes.** Work in topological signal processing and learning on combinatorial domains generalises graph signal processing beyond 1-skeletons by using Hodge Laplacians and higher-order operators to model diffusion, denoising, inference, and representation learning on simplicial and cellular complexes (Isufi et al., 2025a;b; Zia et al., 2024). In parallel, higher-order and sheaf-based neural architectures learn feature dynamics or restriction/transport maps over a fixed combinatorial scaffold to address heterophily, inductive bias, or distribution shift, typically assuming that the underlying incidence operators are *valid* and remain stable throughout training (Hernandez Caralt et al., 2024; Duţa et al., 2023; Zia et al., 2024). The present work complements these directions by focusing on *incidence-like operator learning* itself: when $D_1 D_0 \neq 0$, nullspaces and low-frequency structure can drift silently, corrupting spectral diagnostics and downstream tasks despite good data-fit; we therefore provide a certificate-to-stability pipeline tailored to this failure mode (McGreivy & Hakim, 2024; Isufi et al., 2025b).

**Learning discretisations and operator surrogates.** Learning discrete operators to compensate for modelling error, heterogeneity, or coarse resolution is central to modern data-driven PDE surrogates and neural operator methods (Bar-Sinai et al., 2019; Li et al., 2021; Lu et al., 2021; Pfaff et al., 2021; Kovachki et al., 2023; Li et al., 2023; Wang et al., 2024). These approaches often enforce desirable *matrix-level* properties (e.g., stability/PSD constraints, equivariances, conservation/energy structure) or rely on data-fit objectives that do not directly control complex identities and can therefore leave compatibility violations unconstrained (McGreivy & Hakim, 2024; Kovachki et al., 2023). This contribution is orthogonal: it targets the cochain identity $D_1 D_0 = 0$ as the object that governs discrete nullspaces and low-frequency behaviour, and converts a directly computable *cochain-compatibility defect* into operator-norm perturbation bounds for the induced Hodge Laplacian, making the stability implications of incompatibility explicit and testable (Isufi et al., 2025b).

**Learning Laplacians and graph structure.** Graph structure learning methods recover an adjacency or Laplacian from data under smoothness, sparsity, or probabilistic priors, typically enforcing positive semidefiniteness and other structural constraints directly at the matrix level (Kalofolias, 2016; Ju et al., 2024). While such constraints can improve numerical behaviour, they do not address *higher-order* chain identities that arise when operators come from learned differential complexes rather than from a single learned graph Laplacian (Isufi et al., 2025b; Zia et al., 2024). This work instead certifies compatibility *before* forming the Laplacian (at the level of incidence-like operators), then quantifies how incompatibility propagates into Laplacian drift and, ultimately, into solve sensitivity and spectral reliability (Kovachki et al., 2023).

**Spectral stability foundations.** Eigenvalue and invariant-subspace stability under perturbations is classical, with sharp tools for eigenvalue variation and subspace rotation bounds (Kato, 1995; Stewart & Sun, 1990; Davis & Kahan, 1970). No new perturbation theory is proposed; instead, the missing *bridge* needed in operator-learning settings is supplied: a route from a discrete-complex cochain-compatibility defect to explicit operator-norm budgets for learned Laplacians, enabling standard perturbation results to certify when low-frequency multiplicities, eigenspaces, and associated diagnostics remain reliable under learning-induced drift (Isufi et al., 2025b; Davies et al., 2023).

## 3. Background: Cochain Complexes and Defect Certificate

This section fixes notation and introduces the cochain-compatibility defect used throughout the paper. This section fixes finite-dimensional cochain notation with Euclidean inner products; $\| \cdot \|_{\mathrm{op}}$ and $\| \cdot \|_F$ denote operator/Frobenius norms and $\preceq$ the Loewner order on symmetric matrices.

### 3.1. FEEC/DEC Preliminaries

#### 3.1.1. COCHAIN SPACES AND COBOUNDARY OPERATORS

**Definition 3.1** (Cochain complex). A (linear) cochain complex on $(C^0, C^1, C^2)$ consists of linear maps

$$C^0 \xrightarrow{D_0} C^1 \xrightarrow{D_1} C^2$$

such that $D_1 D_0 = 0$.

*Remark* 3.2 (Terminology: cochain-complex condition vs. image–kernel equality). Throughout, *compatibility* denotes the cochain-complex (nilpotency) condition $D_{k+1} D_k = 0$. The stronger condition $\mathrm{im}(D_k) = \ker(D_{k+1})$—equivalently, vanishing cohomology in the corresponding degree—is *not* assumed unless stated explicitly.

In DEC, $D_0$ and $D_1$ are coboundary operators (discrete exterior derivatives), i.e. the transposes of the boundary operators on chains (incidence matrices) corresponding to oriented edges and faces in a cell complex (Desbrun et al., 2005). Only the algebraic property $D_1 D_0 = 0$ is used in the subsequent analysis.

### 3.1.2. DISCRETE HODGE STAR AND LAPLACIAN

**Assumption 3.3** (Discrete Hodge star on 1-cochains). Let $W_1 \in \mathbb{R}^{n_1 \times n_1}$ be symmetric positive definite.

**Definition 3.4** (0-cochain Laplacian). Under Assumption 3.3, the 0-cochain Laplacian is

$$L_0 := D_0^\mathsf{T} W_1 D_0 \in \mathbb{R}^{n_0 \times n_0}.$$

**Definition 3.5** (Harmonic 0-cochains). The harmonic subspace (in the chosen boundary-condition regime) at degree zero is

$$\mathcal{H}^0 := \ker(L_0).$$

**Lemma 3.6** (Kernel identity). *If $W_1 \succ 0$ then $L_0 = D_0^\mathsf{T} W_1 D_0$ is symmetric positive semidefinite and $\ker(L_0) = \ker(D_0)$.*

*Proof.* See Appendix C.

In DEC on a fixed complex $K$ and under boundary conditions for which Hodge theory identifies cohomology with Laplacian kernels (e.g. absolute/Neumann, or relative variants), $\dim \mathcal{H}^0 = \beta_0(K)$, the number of connected components (Dodziuk, 1976; Arnold et al., 2006). For strongly imposed Dirichlet conditions on a connected domain, the restricted solve operator is typically SPD and has trivial kernel; Section 4.4 makes the boundary-condition regime explicit for Betti estimators.

### 3.2. Learned Complexes and Cochain-Compatibility Defect

This section introduces learned cochain complexes depending on parameters $\phi$ and defines the cochain-compatibility defect that controls the main stability bounds. The construction is described in cochain form; an abstract factorised variant is deferred to the appendix.

### 3.2.1. LEARNED COBOUNDARY OPERATORS AND COCHAIN-COMPATIBILITY DEFECT

**Definition 3.7** (Learned coboundary operators). For each parameter $\phi$, let

$$D_{0,\phi} : C^0 \to C^1, \qquad D_{1,\phi} : C^1 \to C^2$$

be linear maps. No image–kernel equality assumption is imposed a priori.

**Definition 3.8** (Complex defect). Define the composite

$$\mathcal{C}(\phi) := D_{1,\phi} D_{0,\phi} \colon C^0 \to C^2,$$
$$D_{\mathrm{comp}}(\phi) := \|\mathcal{C}(\phi)\|_\mathrm{F} = \|D_{1,\phi} D_{0,\phi}\|_\mathrm{F}.$$

Theorem 4.2 makes $D_{\mathrm{comp}}$ quantitative by relating it to the Frobenius distance to the nearest complex with $D_{1,\phi} D_{0,\phi} = 0$.

### 3.2.2. LEARNED DISCRETE HODGE STAR AND LAPLACIAN

**Assumption 3.9** (Learned discrete Hodge star). For each $\phi$, let $W_{1,\phi} \in \mathbb{R}^{n_1 \times n_1}$ be symmetric positive definite, with uniform bounds

$$0 < w_{\min} I \preceq W_{1,\phi} \preceq w_{\max} I$$

for fixed constants $0 < w_{\min} \leq w_{\max} < \infty$.

**Definition 3.10** (degree-0 discrete Hodge Laplacian). Under Assumption 3.9, define

$$L_{0,\phi} := D_{0,\phi}^\mathsf{T} W_{1,\phi} D_{0,\phi}.$$

### 3.3. Structural Assumptions and Bounds

**Assumption 3.11** (Uniform operator bounds). There exists $B_D > 0$ such that, for all $\phi$,

$$\|D_{0,\phi}\|_\mathrm{op} \leq B_D, \qquad \|D_{1,\phi}\|_\mathrm{op} \leq B_D.$$

**Assumption 3.12** (Uniform nondegeneracy of $D_{1,\phi}$ off its kernel). Let $\sigma_{\min}^+(A)$ denote the smallest nonzero singular value of a matrix $A$. Assume there exists $\sigma_* > 0$ such that for all $\phi$,

$$\sigma_{\min}^+(D_{1,\phi}) \geq \sigma_*.$$

Equivalently, $\|D_{1,\phi}^\dagger\|_\mathrm{op} \leq \sigma_*^{-1}$.

*Remark* 3.13 (Enforcing nondegeneracy). When $D_{1,\phi} \equiv D_1^\mathrm{ref}$ is fixed per mesh, $\sigma_*$ is a mesh-dependent constant checked once. If $D_{1,\phi}$ is learned, $\sigma_{\min}^+(D_{1,\phi}) \geq \sigma_*$ is explicitly enforced and monitored; see Appendix J.

**Checkability.** Appendix A reports how each assumption parameter is computed from the learned operators and meshes used to generate the reported results and the observed ranges in the reported runs.

## 4. Certified Stability via Projection to a Chain-Compatible Complex

This section turns the defect into a nearby compatible reference and then into explicit stability budgets. The practical output is a scalar sufficient-condition criterion, $\rho(\phi) < 1$, that summarizes when the learned operator remains in a certified regime.

## 4.1. Projection to a Chain-Compatible Learned Complex

**Definition 4.1** (Projection onto $\ker(D_{1,\phi})$)**.** Let $P_{\ker(D_{1,\phi})} : C^1 \to C^1$ be the Euclidean-orthogonal projector onto $\ker(D_{1,\phi})$ with respect to the Euclidean inner product on $C^1$. Define

$$\widetilde{D}_{0,\phi} := P_{\ker(D_{1,\phi})} D_{0,\phi} : C^0 \to C^1, \qquad \widetilde{D}_{1,\phi} := D_{1,\phi}.$$

Equivalently, the projection onto the kernel of $D_{1,\phi}$ can be written as

$$P_{\ker(D_{1,\phi})} = I - D_{1,\phi}^{\dagger} D_{1,\phi},$$

where $D_{1,\phi}^{\dagger}$ denotes the Moore–Penrose pseudoinverse. In practice, the implementation relies on Hutchinson probe estimators together with sparse linear solves; additional computational details are provided in Appendix H.

## 4.2. Distance-to-Compatibility and Defect Equivalence

**Theorem 4.2** (Distance-to-compatibility and defect–projection control)**.** *Under Assumptions 3.11 and 3.12, fix $\phi$ and let $\mathcal{E}_{\phi} := \{B : C^0 \to C^1 \, : \, D_{1,\phi}B = 0\}$. Define $D_{\mathrm{proj}}(\phi) := \min_{B \in \mathcal{E}_{\phi}} \|B - D_{0,\phi}\|_F$. Then:*

*(i)* **Projection optimality.**

$$\widetilde{D}_{0,\phi} \quad = \quad \arg\min_{B : D_{1,\phi}B=0} \|B - D_{0,\phi}\|_{\mathrm{F}},$$

*and the minimizer is unique.*

*(ii)* **Defect–projection equivalence.**

$$D_{\mathrm{proj}}(\phi) \geq \frac{1}{\|D_{1,\phi}\|_{\mathrm{op}}} D_{\mathrm{comp}}(\phi),$$

$$D_{\mathrm{proj}}(\phi) \leq \frac{1}{\sigma_*} D_{\mathrm{comp}}(\phi).$$

*Proof.* See Appendix C.

## 4.3. Laplacian of the Projected Complex and Stability

**Definition 4.3** (Projected Laplacian)**.** Under Assumption 3.9, define

$$\widetilde{L}_{0,\phi} := \widetilde{D}_{0,\phi}^{\mathsf{T}} W_{1,\phi} \widetilde{D}_{0,\phi}.$$

The next result bounds $\left\|L_{0,\phi} - \widetilde{L}_{0,\phi}\right\|_{\mathrm{op}}$ as a linear–quadratic function of $D_{\mathrm{comp}}(\phi)$.

**Theorem 4.4** (Operator-norm stability of the Laplacian)**.** *Under Assumptions 3.11, 3.12, and 3.9, for every $\phi$,*

$$\left\|L_{0,\phi} - \widetilde{L}_{0,\phi}\right\|_{\mathrm{op}} \leq A\, D_{\mathrm{comp}}(\phi) + B\, D_{\mathrm{comp}}(\phi)^2,$$

$$A := \frac{2w_{\max}B_D}{\sigma_*}, \qquad B := \frac{w_{\max}}{\sigma_*^2}.$$

*Proof.* See Appendix C.

*Remark* 4.5 (Dependence of constants and mesh scaling)**.** The constants $A, B$ depend on $B_D$, $\sigma_*$ and the Hodge bounds $w_{\min}, w_{\max}$. Across refinement these quantities can be mesh-dependent under the chosen normalisation; Appendix I reports their measured scaling and the resulting margin of the budget.

## 4.4. Approximate Harmonic-Space and Matrix-Level Betti Stability

**Boundary conditions for Betti estimators.** All spectral-count and Betti statements use a Betti-estimation operator whose kernel represents $\beta_0$ (unpinned/relative/Neumann-type). PDE solves use a separate Dirichlet-restricted SPD operator; no Betti claim is made for the SPD solve kernel. Specific boundary choices are fixed in Appendix L. Appendix B defines the two regimes explicitly and specifies how $\lambda_*$ is computed in each.

Let the eigenvalues of $\widetilde{L}_{0,\phi}$ be ordered as

$$0 = \mu_1(\phi) \leq \cdots \leq \mu_{n_0}(\phi),$$

and define $m_0(\phi) := \dim \ker(\widetilde{L}_{0,\phi})$.

**Assumption 4.6** (Uniform gap above the harmonic subspace in the chosen boundary-condition regime)**.** There exists $\lambda_* > 0$ such that, for all $\phi$ in the parameter range of interest, the smallest *positive* eigenvalue of $\widetilde{L}_{0,\phi}$ satisfies

$$\mu_{m_0(\phi)+1}(\phi) \geq \lambda_*, \quad \text{where } m_0(\phi) = \dim \ker(\widetilde{L}_{0,\phi}).$$

*Remark* 4.7 (Sufficient condition via $\widetilde{D}_{0,\phi}$)**.** Since $\widetilde{L}_{0,\phi} = \widetilde{D}_{0,\phi}^{\mathsf{T}} W_{1,\phi} \widetilde{D}_{0,\phi}$ and $W_{1,\phi} \succeq w_{\min}I$, a sufficient condition for Assumption 4.6 is a uniform lower bound $\sigma_{\min}^+(\widetilde{D}_{0,\phi}) \geq \underline{\sigma} > 0$, which implies $\mu_{m_0+1}(\phi) \geq w_{\min}\underline{\sigma}^2$.

**Definition 4.8** (Approximate harmonic subspace (in the chosen boundary-condition regime))**.** Let $L_{0,\phi}$ be symmetric positive semidefinite with eigenvalues $0 \leq \lambda_1(\phi) \leq \cdots \leq \lambda_{n_0}(\phi)$. For $\tau > 0$, the approximate harmonic subspace (in the chosen boundary-condition regime) at threshold $\tau$ is

$$\mathcal{H}_{<\tau}^0(\phi) := \mathrm{span}\Big\{ v_i(\phi) \, : \, L_{0,\phi}v_i(\phi) = \lambda_i(\phi)v_i(\phi),$$
$$\lambda_i(\phi) < \tau \Big\}.$$

with dimension $m_{<\tau}(\phi) := \dim \mathcal{H}_{<\tau}^0(\phi)$.

The next result establishes a sufficient defect threshold for invariance of the low-frequency eigenvalue count.

**Theorem 4.9** (Low-frequency spectral-count stability)**.** *Suppose Assumptions 3.11, 3.12, 3.9, and 4.6 hold. Let $A, B$ be the constants from Theorem 4.4, and set*

$$\delta(\phi) := A\, D_{\mathrm{comp}}(\phi) + B\, D_{\mathrm{comp}}(\phi)^2.$$

If $\delta(\phi) < \lambda_*/2$, then $L_{0,\phi}$ and $\widetilde{L}_{0,\phi}$ have the same number of eigenvalues in $[0, \lambda_*/2)$. Equivalently,

$$m_{<\lambda_*/2}(\phi) = \dim \ker(\widetilde{L}_{0,\phi}).$$

*Proof.* See Appendix C.

**Corollary 4.10** (Fixed-resolution spectral $\beta_0$ correctness). *Assume the Betti-estimation setting in which* $\dim \ker(\widetilde{L}_{0,\phi}) = \beta_0(K_h)$ *(e.g. closed/Neumann/relative), and Assumption 4.6. If* $\delta(\phi) < \lambda_*/2$, *then the spectral count* $\#\{i : \lambda_i(\phi, h) < \lambda_*/2\}$ *equals* $\beta_0(K_h)$.

*Proof.* See Appendix C.

**Operational certificate and stability criterion (scalar sufficient-condition criterion).** Given any learned triple $(D_{0,\phi}, D_{1,\phi}, W_{1,\phi})$, estimate the cochain-compatibility defect $D_{\mathrm{comp}}(\phi) = \|D_{1,\phi}D_{0,\phi}\|_F$ (Appendix H), set the Laplacian-drift budget

$$\delta(\phi) := A\,D_{\mathrm{comp}}(\phi) + B\,D_{\mathrm{comp}}(\phi)^2 \qquad \text{(Theorem 4.4)},$$

and let $\lambda_*$ be the relevant spectral gap (Betti-estimation and/or SPD solve regime; Appendix L). Define the certificate ratio

$$\rho(\phi) := \frac{\delta(\phi)}{\lambda_*/2}.$$

If $\rho(\phi) < 1$, then small-eigenvalue count below threshold and SPD solves remain stable (Theorems 4.9–4.10, C.15). The ratio $\rho$ is used as a sufficient-condition check and training monitor (regularise or early-stop as $\rho \uparrow 1$).

**Higher-degree extensions.** All arguments extend degreewise to $k$-cochains and factorised operators $L_{k,\phi} = D_{k,\phi}^{\mathsf{T}} W_{k+1,\phi} D_{k,\phi}$ with analogous linear–quadratic stability in the corresponding defect $D_{k,\mathrm{comp}}(\phi)$. This aspect is further discussed in Appendix G.

## 5. PDE Error Bounds and Spectral Topology

This section connects the algebraic certificate to the two downstream objects that matter in practice: PDE solves and topology-sensitive spectral counts. The resulting bounds separate ordinary discretization error from the additional error introduced by cochain incompatibility.

The algebraic construction is embedded into a FEEC setting for an elliptic partial differential equation.

Let $V$ be a Hilbert space continuously embedded in $H^1(\Omega)$, for example $V = H_0^1(\Omega)$, and consider the variational problem: find $u \in V$ such that

$$a(u, v) = \ell(v) \quad \forall v \in V, \tag{2}$$

where $a : V \times V \to \mathbb{R}$ is a continuous coercive bilinear form and $\ell : V \to \mathbb{R}$ is continuous.

Let $(V_h^0)_{h>0}$ be a standard FEEC family of discrete spaces approximating $V$ (Arnold et al., 2006). On each mesh $K_h$, we identify $V_h^0$ with a cochain space $C^0(K_h) \cong \mathbb{R}^{n_0}$ via a fixed coefficient representation with Euclidean discrete inner product. Let $L_{0,h}^{\mathrm{ref}}$ denote a reference discrete 0-cochain Laplacian. The reference FEEC bilinear form is

$$a_h(u_h, v_h) := \mathbf{u}^{\mathsf{T}} L_{0,h}^{\mathrm{ref}} \mathbf{v},$$
$$u_h = \sum_i u_i \varphi_i, \qquad v_h = \sum_i v_i \varphi_i, \tag{3}$$

with coefficient vectors $\mathbf{u}, \mathbf{v} \in \mathbb{R}^{n_0}$.

On the learned side, a parameterisation of $(D_{0,\phi}, D_{1,\phi}, W_{1,\phi})$ on $K_h$ induces a learned Laplacian $L_{0,\phi,h}$. Similarly, $C^1(K_h) \cong \mathbb{R}^{n_1}$ denotes the chosen 1-cochain space (edge cochains/Whitney 1-forms), on which $D_{0,\phi,h} : C^0(K_h) \to C^1(K_h)$ acts. The corresponding learned bilinear form on $V_h^0 \times V_h^0$ is

$$a_{\phi,h}(u_h, v_h) := \mathbf{u}^{\mathsf{T}} L_{0,\phi,h}\, \mathbf{v},$$
$$L_{0,\phi,h} := D_{0,\phi,h}^{\mathsf{T}} W_{1,\phi,h} D_{0,\phi,h}. \tag{4}$$

*Remark* 5.1 (Mesh dependence of the defect). On a mesh family $(K_h)_h$ the defect is mesh-dependent, $D_{\mathrm{comp}}(\phi, h) := \|D_{1,\phi,h} D_{0,\phi,h}\|_{\mathrm{F}}$. To lighten notation, the dependence on $h$ is suppressed; all statements are understood with uniform control over the meshes and parameters of interest.

**Assumption 5.2** (Discrete setting and anchoring). For each $h$ and $\phi$:

- the learned operators satisfy the structural bounds of Sections 3.2–4.4 and Appendix G;

- the projected operator $\widetilde{D}_{0,\phi,h} = P_{\ker(D_{1,\phi,h})} D_{0,\phi,h}$ is used only as a *theoretical reference operator* to define distances/perturbation bounds; it is not used for deployment, and it need not preserve sparsity/locality; $\widetilde{L}_{0,\phi,h} := \widetilde{D}_{0,\phi,h}^{\mathsf{T}} W_{1,\phi,h} \widetilde{D}_{0,\phi,h}$ is well-defined on the same finite-dimensional space;

- the deviation from the reference family is quantified by

$$\mathcal{R}_E(\phi) := \left\| \widetilde{L}_{0,\phi,h} - L_{0,h}^{\mathrm{ref}} \right\|_{\mathrm{op}},$$

and training may optionally regularise this term to prevent operator-identification degeneracies.

### 5.1. Bilinear-Form Deviation Bound

**Lemma 5.3** (Bilinear-form deviation). *Under Assumptions 3.9 and 5.2, and the standard norm equivalence between the $V$-norm and the energy norm induced by the*

*FEEC discretization, there exists $C_{\mathrm{op}} > 0$ such that, for all $\phi$, $h$ and $u_h, v_h \in V_h^0$,*

$$
\begin{aligned}
|a_{\phi,h}(u_h, v_h) - a_h(u_h, v_h)| &\leq C_{\mathrm{op}} \left\| L_{0,\phi,h} - \widetilde{L}_{0,\phi,h} \right\|_{\mathrm{op}} \\
&\quad \cdot \|u_h\|_V \, \|v_h\|_V \\
&\quad + C_{\mathrm{op}} \left\| \widetilde{L}_{0,\phi,h} - L_{0,h}^{\mathrm{ref}} \right\|_{\mathrm{op}} \\
&\quad \cdot \|u_h\|_V \, \|v_h\|_V \, .
\end{aligned}
$$

*Combining with Theorem 4.4 yields a defect-controlled bound with the same linear–quadratic perturbation bound $AD_{\mathrm{comp}}(\phi) + BD_{\mathrm{comp}}(\phi)^2$ plus the deviation term $\mathcal{R}_E(\phi)$.*

*Proof.* See Appendix C. ∎

### 5.2. PDE Error Bound with a Complex-Defect Term

Let $u \in V$ be the solution of (2). The reference FEEC discrete problem is: find $u_h \in V_h^0$ such that

$$
a_h(u_h, v_h) = \ell(v_h) \quad \forall v_h \in V_h^0. \tag{5}
$$

Under standard FEEC assumptions, there exists a constant $C_{\mathrm{FEEC}}$ independent of $h$ such that

$$
\|u - u_h\|_V \leq C_{\mathrm{FEEC}} \Big( \inf_{v_h \in V_h^0} \|u - v_h\|_V \\
+ \text{(mesh/quadrature terms)} \Big).
$$

see e.g. (Arnold et al., 2006).

The learned-discrete problem uses $a_{\phi,h}$ instead of $a_h$: find $u_{\phi,h} \in V_h^0$ such that

$$
a_{\phi,h}(u_{\phi,h}, v_h) = \ell(v_h) \quad \forall v_h \in V_h^0. \tag{6}
$$

**Assumption 5.4** (Discrete coercivity). The learned bilinear forms are uniformly coercive on $V_h^0$, i.e. there exists $\alpha_0 > 0$, independent of $h$ and $\phi$, such that $a_{\phi,h}(v_h, v_h) \geq \alpha_0 \|v_h\|_V^2$ for all $v_h \in V_h^0$.

*Remark* 5.5. In practice this holds automatically for the SPD solve operator obtained after standard Dirichlet elimination/restriction, assuming $W_{1,\phi} \succ 0$ and full-rank restricted $D_{0,\phi}$ on the solve subspace.

Fix $u \in V$ and $w_h \in V_h^0$, and let $u_h^\star \in \arg\min_{v_h \in V_h^0} \|u - v_h\|_V$. A standard decomposition isolates the learning-induced term $a_h(u_h^\star, w_h) - a_{\phi,h}(u_h^\star, w_h)$, which can be controlled by Lemma 5.3.

The next result gives a Strang-type error decomposition with an explicit defect term (Strang & Fix, 1973; Arnold et al., 2006).

**Theorem 5.6** (PDE error with cochain-compatibility defect). *Let $u \in V$ solve (2) and $u_{\phi,h} \in V_h^0$ solve (6). Assume Assumptions 5.2 and 5.4, and standard FEEC regularity and interpolation estimates. Then there exist constants $C_1, C_2 > 0$, independent of $h$ and $\phi$, such that*

$$
\begin{aligned}
\|u - u_{\phi,h}\|_V &\leq C_1 \inf_{v_h \in V_h^0} \|u - v_h\|_V \\
&\quad + C_2 \Big( AD_{\mathrm{comp}}(\phi) + BD_{\mathrm{comp}}(\phi)^2 \Big) \\
&\quad + C_2 \, \mathcal{R}_E(\phi) \\
&\quad + \text{(mesh/quadrature terms)}.
\end{aligned}
$$

*Proof.* See Appendix C. ∎

Theorem 5.6 separates standard FEEC approximation and quadrature error from two learning-induced terms: $\mathcal{R}_E(\phi)$, which measures drift of the projected chain-compatible anchor from the reference family, and the linear–quadratic defect term, which measures the additional price of violating the cochain-complex condition. Consequently, if $D_{\mathrm{comp}}(\phi, h)$ and $\mathcal{R}_E(\phi, h)$ decay at least as fast as the reference discretisation error, the learned method preserves the asymptotic FEEC regime; otherwise the defect creates a non-vanishing perturbation floor under refinement.

## 6. Experiments

This section tests three questions: whether defect control stabilizes refinement, whether it improves robustness under discretization shift, and whether the certificate remains informative under topology shift. Full protocols, hyperparameters, additional plots, and run-by-run tables are deferred to Appendices E–M.

### 6.1. Training Regimes and Objectives

Across experiments, the learned operators are trained under three regimes:

- **Unconstrained.** Minimise a PDE loss $\mathcal{L}_{\mathrm{PDE}}(\phi)$ with respect to the trainable operators in the configuration (default: $D_{0,\phi}$; other variants train $D_{1,\phi}$ and/or $W_{1,\phi}$ when stated).

- **Soft defect regularisation.** Minimise

$$
\mathcal{L}_{\mathrm{PDE}}(\phi) + \lambda_{\mathrm{comp}} \|D_{1,\phi} D_{0,\phi}\|_{\mathrm{F}}^2 + \lambda_E \, \mathcal{R}_E(\phi)^2,
$$

where $\mathcal{R}_E(\phi) = \left\| \widetilde{L}_{0,\phi,h} - L_{0,h}^{\mathrm{ref}} \right\|_{\mathrm{op}}$ is an operator-deviation term defined by the projected reference operator.

- **Chain-compatible parameterisation (hard constraint).** Fix $D_1^{\mathrm{ref}}$ (mesh incidence) and parameterise $D_{0,\phi} \in \ker(D_1^{\mathrm{ref}})$ so that $D_1^{\mathrm{ref}} D_{0,\phi} \equiv 0$ by construction. In this regime $D_{\mathrm{comp}}(\phi) \equiv 0$ and training targets $\mathcal{L}_{\mathrm{PDE}}(\phi)$ and $\mathcal{R}_E(\phi)$.

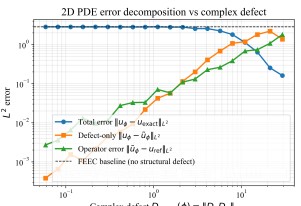
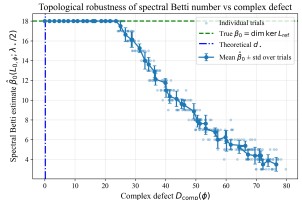

*(a)* PDE error versus defect    *(b)* Spectral $\widehat{\beta}_0^{\mathrm{spec}}$ versus defect

*Figure 1.* Trade-off induced by defect control. Left: PDE error scaling. Right: Fixed-resolution spectral Betti estimator.

**Baselines and logging.** Baselines are defined in Appendix K. We report $D_{\mathrm{comp}}(\phi)$ (and a cross-resolution normalisation) and log $\rho(\phi)$ as an instability monitor; full protocols are in the appendix. Unless stated otherwise, all certificate quantities are computed directly from the realized sparse operators on the evaluation mesh.

### 6.2. Structural Evidence on Elliptic Benchmarks

On a fixed mesh, random perturbations of $(D_0, D_1)$ (or learned updates under $\mathcal{L}_{\mathrm{PDE}}$) yield:

- **Operator and PDE error scaling.** Operator deviation follows the predicted linear–quadratic regime and $L^2$ solution error separates into a FEEC baseline plus a defect-dependent contribution that scales like $AD_{\mathrm{comp}}(\phi) + BD_{\mathrm{comp}}(\phi)^2$, consistent with Theorems 4.4 and 5.6.

- **Resolution scaling and solver conditioning.** At coarse resolution $N = 8$, all regimes achieve comparable test accuracy; at higher resolution $N = 32$, unconstrained training enters a severely ill-conditioned regime ($\mathrm{Cond}(L_0) \approx 1.7 \times 10^4$, Table 1), while defect-controlled training maintains stable conditioning ($\mathrm{Cond}(L_0) \approx 2.4 \times 10^2$).

- **Topology shift and Betti robustness.** The spectral estimator $\widehat{\beta}_0^{\mathrm{spec}}$ remains correct throughout a small-defect region (Theorem 4.10), and defect-based certificate ratios predict numerical-solver and topological failures under shift substantially better than validation PDE loss (AUC $\approx 0.85$ vs. near chance).

### 6.3. Out-of-Distribution (OOD) Discretisation-Shift Evaluation on Unstructured Meshes

Discretisation shift is evaluated in two regimes. The first trains on structured triangulations and tests on unstructured triangulations with the same PDE; defect control improves relative $L^2$ error on both pressure and flux targets while keeping structure diagnostics small (Appendix M, Table 9). The second is stricter: both $D_0$ and $D_1$ are learned jointly under OOD mesh shift, so the comparison is neither a fixed-$D_1$ case nor a projection-only ablation. In that regime the

methods are nearly matched on ID error, but the defect-aware model achieves the best OOD nRMSE, the smallest raw compatibility defect, the lowest realized drift relative to the certified budget, the lowest failure rates across the certificate threshold, and the highest OOD spectral Betti accuracy (Appendix M, Table 10).

### 6.4. Topology-Shift Evaluation

Trained only on connected instances ($\beta_0 = 1$), models are tested on meshes where $\beta_0 \in \{1, 2, 3\}$ by edge removal (Table 2). Each checkpoint is scored by the certificate ratio $\rho(\phi)$ from Paragraph 4.4, i.e. $\rho = \delta(\phi)/(\lambda_*/2)$ with $\delta(\phi) = AD_{\mathrm{comp}}(\phi) + BD_{\mathrm{comp}}(\phi)^2$. Failure rate is low throughout $\rho < 1$ and increases rapidly once $\rho$ crosses the threshold (Fig. 2a), and $D_{\mathrm{comp}}$ predicts solver/topology failures substantially better than validation PDE loss (AUC $\approx 0.85$ vs. near chance, Fig. 2b).

## 7. Discussion

**Applicability.** The certificate is aimed at settings where incidence-like operators are learned or adapted and compatibility by construction is not guaranteed. In that regime, low PDE loss alone does not rule out spectral drift, poor conditioning, or unstable topological diagnostics. The certificate is computed from the realized sparse operators on the evaluation mesh, so it is deployment-neutral in the sense that it audits the discrete operator that the solver actually receives after assembly under the chosen boundary regime. It is therefore orthogonal to architecture-level descriptions and aggregate prediction metrics. Two models with similar validation error can induce materially different nullspaces, spectral gaps, and conditioning once realised on a shifted mesh family or topology. In that setting, $D_{\mathrm{comp}}(\phi)$ and $\rho(\phi)$ report whether the learned operator remains within the perturbative regime controlled by the projected reference, rather than whether the training target was fit well on average. This is why the certificate remains informative precisely where loss values are least diagnostic, namely under refinement, remeshing, and topology-sensitive deployment.

**Hard versus soft structural control.** When compatibility can be imposed by construction without excluding the operator family of interest, hard compatibility is the preferred option. The experiments support a clear hierarchy: hard constraints give the strongest structural guarantee, soft defect control captures most of the stability benefit while remaining competitive in PDE accuracy, and unconstrained training is the regime most prone to conditioning and spectral failures. Soft control is therefore not a substitute for a compatibility-preserving parameterization; its role is to handle settings in which compatibility by construction is unavailable, too restrictive, mesh-specific, or mismatched

*Table 1.* **Resolution scaling and stability breakdown.** Unconstrained training exhibits severe conditioning deterioration at higher resolutions, while defect-controlled methods remain in a certified regime. Results show mean $\pm$ std over 5 random seeds for PDE1 (constant coefficient Poisson, $\kappa \equiv 1$, analytic solution $u = \sin(\pi x)\sin(\pi y)$). Additional PDEs and diagnostics in Appendix L, Table 7.

| Resolution | Training regime | Test MSE ↓ | Cond($L_0$) ↓ | Norm. Defect ↓ | Spectral Gap ↑ | Certificate status |
|---|---|---|---|---|---|---|
| $N = 16$ | Unconstrained | $(8.73 \pm 1.24) \times 10^{-4}$ | $(1.89 \pm 0.31) \times 10^{2}$ | $1.47 \pm 0.08$ | $(1.83 \pm 0.21) \times 10^{-2}$ | Near threshold |
| | Soft defect penalty | $(3.42 \pm 0.18) \times 10^{-6}$ | $(6.12 \pm 0.09) \times 10^{1}$ | $(2.31 \pm 0.14) \times 10^{-3}$ | $(1.07 \pm 0.02) \times 10^{-2}$ | Certificate satisfied |
| | Chain-compatible | $(8.74 \pm 0.31) \times 10^{-7}$ | $(6.08 \pm 0.06) \times 10^{1}$ | $0$ | $(1.07 \pm 0.01) \times 10^{-2}$ | Certificate satisfied |
| $N = 32$ | Unconstrained | $(2.18 \pm 0.62) \times 10^{2}$ | $(1.73 \pm 0.29) \times 10^{4}$ | $3.82 \pm 0.51$ | $(3.47 \pm 1.82) \times 10^{-4}$ | Certificate violated |
| | Soft defect penalty | $(2.91 \pm 0.14) \times 10^{-7}$ | $(2.47 \pm 0.12) \times 10^{2}$ | $(6.18 \pm 0.83) \times 10^{-4}$ | $(1.34 \pm 0.03) \times 10^{-3}$ | Certificate satisfied |
| | Chain-compatible | $(1.47 \pm 0.09) \times 10^{-8}$ | $(2.45 \pm 0.08) \times 10^{2}$ | $0$ | $(1.34 \pm 0.02) \times 10^{-3}$ | Certificate satisfied |
| $N = 64$ | Unconstrained | $(1.94 \pm 0.81) \times 10^{4}$ | $(3.58 \pm 1.47) \times 10^{6}$ | $7.23 \pm 1.19$ | $(8.91 \pm 6.34) \times 10^{-6}$ | Solver failure |
| | Soft defect penalty | $(4.83 \pm 0.27) \times 10^{-8}$ | $(9.63 \pm 0.28) \times 10^{2}$ | $(1.84 \pm 0.22) \times 10^{-4}$ | $(1.68 \pm 0.04) \times 10^{-4}$ | Certificate satisfied |
| | Chain-compatible | $(7.21 \pm 0.48) \times 10^{-10}$ | $(9.58 \pm 0.19) \times 10^{2}$ | $0$ | $(1.68 \pm 0.03) \times 10^{-4}$ | Certificate satisfied |

Mean $\pm$ std over 5 seeds; metrics defined in Appendix L; training details in Appendix L.

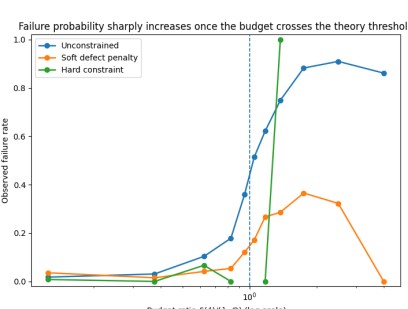 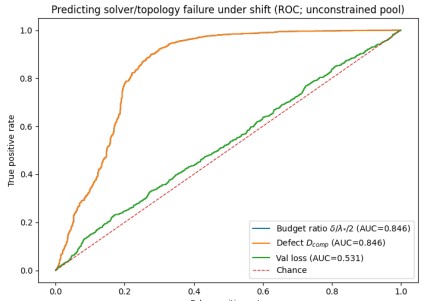 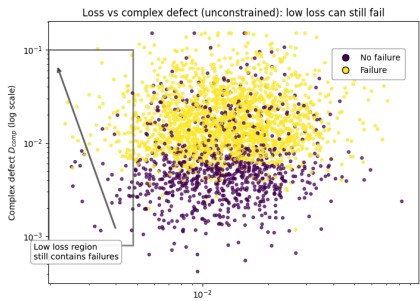

*(a)* Failure rate vs. $\rho = \delta/(\lambda_*/2)$ (log-x); dashed line: $\rho = 1$.

*(b)* Predicting failure under shift (ROC; unconstrained pool).

*(c)* Loss vs. defect (unconstrained): low loss can still fail.

*Figure 2.* **Certificate-based budgets predict deployment failures under topological distribution shift.** Results aggregated over 5 seeds. (a) Failure probability increases sharply once the predicted budget ratio $\rho$ crosses the $\lambda_*/2$ threshold from Theorem 4.9. (b) $D_{\text{comp}}$ (or $\rho$) predicts failures well (AUC $\approx 0.85$), while validation PDE loss is near chance; the $D_{\text{comp}}$ and $\rho$ ROC curves overlap. (c) Loss and defect are weakly coupled: failures can occur at low loss when defect is large. Failure denotes solver breakdown and/or incorrect $\widehat{\beta}_0^{\text{spec}}$ (details in Appendix E).

*Table 2.* **Topology-shift evaluation.** Training on $\beta_0 = 1$; testing on $\beta_0 \in \{1, 2, 3\}$. Reported: mean nRMSE, max error, Betti accuracy, and defect.

| Training regime | nRMSE ↓ | max err. ↓ | Acc. ↑ | $D_{\text{comp}}$ ↓ |
|---|---|---|---|---|
| Unconstrained | $4.6 \times 10^{-2}$ | $1.18 \times 10^{-1}$ | $0.71$ | $2.1 \times 10^{-2}$ |
| Soft defect reg. | $4.1 \times 10^{-2}$ | $6.9 \times 10^{-2}$ | $0.93$ | $6.0 \times 10^{-3}$ |
| Chain-compatible (hard) | $4.3 \times 10^{-2}$ | $6.1 \times 10^{-2}$ | $0.98$ | $0$ |

to the intended sparse or local operator family.

**Interpretation and limits.** The certificate is a sufficient-condition diagnostic for structural safety, not a surrogate for best approximation accuracy. Some accurate models can remain uncertified, while certified models are consistently better behaved. The projected reference operator is global and may destroy sparsity, so it is used for analysis rather than deployment. The nondegeneracy of $D_{1,\phi}$ must be monitored or enforced when $D_1$ is learned. The raw Frobenius defect is basis- and resolution-dependent, so normalized variants are needed across meshes. Topology claims also require the appropriate boundary regime and a spectral gap above the

relevant near-zero cluster. Degree-wise extensions follow the same proof pattern and are collected in Appendix G.1.

## 8. Conclusion

A scalar cochain-compatibility defect provides a computable certificate for chain-condition violation in learned cochain Laplacians. An explicit orthogonal Frobenius-orthogonal projection onto the constraint subspace yields a distance-to-compatibility anchor and a linear–quadratic operator perturbation regime for the induced Laplacian. In FEEC settings, the defect appears as an explicit additive contribution in Strang-type error bounds, and under a spectral gap in a compatible boundary-condition regime it yields sufficient conditions for correct low-frequency multiplicities and fixed-resolution spectral Betti-number estimators. Experiments are consistent with the predicted regimes and indicate that explicit defect control improves robustness under deployment shifts that alter topology, and under discretisation out-of-distribution shifts on unstructured meshes where error reductions coincide with small structure diagnostics.

## Impact Statement

This paper introduces a computable certificate for whether learned discrete operators on meshes/complexes respect key algebraic identities that underpin stable PDE solves and low-frequency spectral diagnostics. By making such structure violations detectable and monitorable, it can improve reliability and reporting in operator-learning pipelines, especially under refinement and distribution shift.

A key risk is over-interpretation: the certificate provides sufficient-condition guarantees under explicit assumptions and should not be treated as a universal safety guarantee or a substitute for problem-specific validation.

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

# A. Implementation Details and Verification Protocols

This appendix makes each structural assumption operational: it records (i) what is assumed, (ii) how the quantity is computed from the learned operators and meshes used to generate the reported results, (iii) the observed ranges in the reported runs, and (iv) the characteristic failure signature when the assumption is violated.

## A.1. Practical Considerations

**Rationale for the Frobenius Defect Norm.** It is an unbiased trace-estimable quantity via Hutchinson probes (Appendix H), cheap enough to monitor during training, and it upper-bounds the operator-norm defect $D_{\mathrm{op}}$ (Appendix C) used for tighter but costlier certification.

**Basis and Resolution Dependence.** Yes. We therefore report a normalised defect for cross-resolution comparison (Appendix E) and provide a metric-aware variant $D_{\mathrm{comp}}^{(W)}$ with explicit equivalence constants (Definition C.2, Lemma C.3).

**Sparsity Preservation and Acceptability of Projection.** $P_{\ker(D_1)}$ is generally global and may densify $\widetilde{D}_0$. The projected operator is used only as an analysis anchor and diagnostic reference, not for deployment (Assumption 5.2); all deployed operators remain in the original sparse parametrisation family.

**Selection of $\lambda_*$** This parameter is chosen conservatively from the projected-anchor spectra on the training distribution and validated under shift (Appendix C).

**Conditions for Certificate Validity.** When either (i) $\sigma_{\min}^+(D_{1,\phi})$ becomes small so that $A, B$ grow unbounded (Appendix J), or (ii) the relevant spectral gap collapses so that $\delta(\phi) \geq \lambda_*/2$ (Appendix C). In both cases the criterion

*Table 3.* **Assumption verification (I): operator/metric bounds and projection nondegeneracy.** Observed values report min/median/max over the reported seeds, meshes, and checkpoints.

| Assumed quantity | Where used | How it is estimated (check-able) | Failure signature |
|---|---|---|---|
| $B_D$ (Assump. 3.11) | Thms. 4.2, 4.4, downstream budgets | Power iteration on $D_{0,\phi}^\top D_{0,\phi}$ and $D_{1,\phi}^\top D_{1,\phi}$ (matrix-free); report $\max\{\|D_{0,\phi}\|_{\mathrm{op}}, \|D_{1,\phi}\|_{\mathrm{op}}\}$ | Budget margin grows; $\delta(\phi) = AD_{\mathrm{comp}} + BD_{\mathrm{comp}}^2$ becomes overly conservative |
| $(w_{\min}, w_{\max})$ (Assump. 3.9) | All Laplacian stability and PDE bounds | If $W_{1,\phi}$ is diagonal/elementwise positive: $w_{\min} = \min_i(W_{1,\phi})_{ii}$, $w_{\max} = \max_i(W_{1,\phi})_{ii}$; otherwise estimate extreme eigenvalues by Lanczos/power iteration | Loss of SPD in $W_{1,\phi}$; unstable solves; drift constants $A, B$ grow unbounded |
| $\sigma_*$ (Assump. 3.12) | Projection equivalence; constants $A, B$; projector stability | Estimate $\sigma_{\min}^+(D_{1,\phi})$ via Lanczos on $D_{1,\phi}D_{1,\phi}^\top$ restricted to $\mathrm{range}(D_{1,\phi})$; equivalently estimate $\|D_{1,\phi}^\dagger\|_{\mathrm{op}}$ by inverse iteration with projection to $\mathrm{range}(D_{1,\phi})$ | $\|D_{1,\phi}^\dagger\|_{\mathrm{op}} \uparrow$; projection solve becomes ill-conditioned; $D_{\mathrm{proj}}$ no longer tracks $D_{\mathrm{comp}}$ tightly |

$\rho(\phi) \geq 1$ is expected to trigger, signalling unreliable Betti estimates or unstable solves.

## A.2. Notation Audit: Operators, Regimes, and Corresponding Results

The notations used and corresponding applications are given in Table 5.

## B. Boundary Condition Regimes

This appendix fixes the two boundary-condition regimes used in the paper and makes the role of $\lambda_*$ explicit in each: (i) the Dirichlet-restricted SPD solve operator used for PDE solves and conditioning, and (ii) the Betti-estimation operator whose kernel represents $\beta_0$ (Neumann/relative/unpinned regimes).

### B.1. Dirichlet SPD Solve Operator (PDE Solves)

Let $L_{0,\phi} \in \mathbb{R}^{n_0 \times n_0}$ be the learned degree-0 Laplacian. Impose Dirichlet conditions by elimination/restriction to interior degrees of freedom.

**Restriction/elimination operator.** Let $\mathcal{I} \subset \{1, \ldots, n_0\}$ index interior DOFs and define the restriction matrix $R \in \mathbb{R}^{|\mathcal{I}| \times n_0}$ that selects interior components, i.e. $(Ru)_j = u_{\mathcal{I}_j}$ and $R^\top$ injects by zero-filling boundary entries. Define the

SPD operator

$$L_{0,\phi}^{\mathrm{SPD}} := R\,L_{0,\phi}\,R^\top \in \mathbb{R}^{|\mathcal{I}| \times |\mathcal{I}|}. \tag{7}$$

When Dirichlet conditions are strongly imposed on a connected domain and $W_{1,\phi} \succ 0$, the restricted operator is SPD under full-rank of the restricted gradient map.

**Condition number in the SPD regime.** All reported $\mathrm{Cond}(L_0)$ values in tables correspond to $\mathrm{Cond}(L_{0,\phi}^{\mathrm{SPD}})$:

$$\mathrm{Cond}(L_{0,\phi}^{\mathrm{SPD}}) := \frac{\lambda_{\max}(L_{0,\phi}^{\mathrm{SPD}})}{\lambda_{\min}(L_{0,\phi}^{\mathrm{SPD}})}.$$

**$\lambda_*$ in the SPD regime.** For Theorem C.15, the relevant spectral margin is

$$\lambda_*^{\mathrm{SPD}} := \lambda_{\min}(\widetilde{L}_{0,\phi}^{\mathrm{SPD}}), \qquad \widetilde{L}_{0,\phi}^{\mathrm{SPD}} := R\,\widetilde{L}_{0,\phi}\,R^\top.$$

The sufficient condition used in Corollary C.16 is $\|L_{0,\phi}^{\mathrm{SPD}} - \widetilde{L}_{0,\phi}^{\mathrm{SPD}}\|_{\mathrm{op}} < \lambda_*^{\mathrm{SPD}}/2$.

### B.2. Betti-Estimation Operator (Neumann/Relative/Unpinned)

Betti estimation uses an operator whose kernel represents $\beta_0$; this is *not* the Dirichlet SPD operator.

**Unpinned/relative regime.** Let $\widetilde{L}_{0,\phi}$ denote the projected reference Laplacian in the Betti-estimation regime. The

*Table 4.* **Assumption verification (II): spectral gap, SPD coercivity, and FEEC mesh constants.** All entries are computed from the learned operators and meshes used to generate the reported results.

| Assumed quantity | Where used | How it is estimated (check-able) | Failure signature |
|---|---|---|---|
| $\lambda_*$ (Assump. 4.6) | Thm. 4.9, Cor. 4.10, criterion $\rho$ | Compute the smallest positive eigenvalue of the projected reference $\widetilde{L}_{0,\phi}$ in the Betti-estimation regime (Appendix B); use Lanczos for a few extremal eigenpairs; set $\lambda_*$ as conservative inf/low-quantile over the training pool | Gap collapse: $\tilde{\mu}_{m_0+1} \downarrow 0$; thresholding becomes unstable; $\rho(\phi) \uparrow 1$ predicts failure |
| $\alpha_0$ (Assump. 5.4) | PDE well-posedness and Strang-type bounds | In Dirichlet SPD regime, $\alpha_0$ is the smallest eigenvalue of $L^{\mathrm{SPD}}$ (Appendix B); estimate via Lanczos; report per mesh $\alpha_0(h) := \lambda_{\min}(L^{\mathrm{SPD}})$ | SPD loss: near-zero eigenvalues; solver breakdown; condition number explosion |
| Shape-regularity / norm-equivalence constants | Lemma 5.3 and norm maps in Appendix C.1 | Use mesh generator's shape metrics: minimal angle / aspect ratio bounds; or report standard FEEC constants implied by mesh family. If computed: report $\max_K h_K/\rho_K$ or min angle on each mesh | Non-uniform norm equivalence; FEEC constants inflate; error bounds become loose |

*Table 5.* Operator roles and boundary regimes. Betti stability results apply to the Betti-estimation operator, not to the Dirichlet SPD solve operator.

| Symbol | Meaning | Application |
|---|---|---|
| $L_{0,\phi} = D_{0,\phi}^\top W_{1,\phi} D_{0,\phi}$ | learned Laplacian | Production solver / Betti-estimation operator |
| $\widetilde{D}_{0,\phi} = P_{\ker(D_{1,\phi})} D_{0,\phi}$ | projected (anchor) coboundary | certification + analysis only |
| $\widetilde{L}_{0,\phi} = \widetilde{D}_{0,\phi}^\top W_{1,\phi} \widetilde{D}_{0,\phi}$ | projected (anchor) Laplacian | gap definition / perturbation reference |
| $L_{0,h}^{\mathrm{ref}}$ | reference FEEC/DEC Laplacian | PDE baseline / anchoring |
| $L^{\mathrm{solve}}$ | Dirichlet-restricted SPD operator | elliptic solvers (Theorem C.15) |
| $L^{\mathrm{top}}$ | unpinned/relative topology operator | Betti estimation (Cor. 4.10) |

spectral Betti stability analysis assumes a regime in which

$$\dim \ker(\widetilde{L}_{0,\phi}) = \beta_0(K_h)$$

(e.g. Neumann/absolute on closed components, or relative variants where the cohomology is represented by Laplacian kernels; see (Dodziuk, 1976; Arnold et al., 2006)).

$\lambda_*$ **in the Betti-estimation regime.** Let $\tilde{\mu}_1(\phi) \leq \cdots \leq \tilde{\mu}_{n_0}(\phi)$ be eigenvalues of $\widetilde{L}_{0,\phi}$ in the Betti-estimation regime with $m_0(\phi) = \dim \ker(\widetilde{L}_{0,\phi})$. Define

$$\lambda_*^{\mathrm{top}} := \inf_{\phi \in \Phi_{\mathrm{train}}} \tilde{\mu}_{m_0(\phi)+1}(\phi),$$

optionally using a conservative low-quantile across seeds/checkpoints to reduce numerical noise while remaining conservative for the reported distribution.

**Thresholding and numerical tolerance.** Theorems 4.9–4.10 use the threshold interval $[0, \lambda_*^{\mathrm{top}}/2)$. In computations, eigenvalues are treated as "below threshold" if $\lambda_i < \lambda_*^{\mathrm{top}}/2 - \tau_{\mathrm{eig}}$, where $\tau_{\mathrm{eig}}$ matches the Lanczos/solver tolerance used to compute extremal eigenpairs and the projection solves. This prevents classification inconsistencies due to numerical noise in cases where the spectral gap is small.

**Which $\lambda_*$ enters the operational criterion.** The certificate ratio in Paragraph 4.4 uses $\lambda_*^{\mathrm{top}}$ for spectral-count/Betti claims and $\lambda_*^{\mathrm{SPD}}$ for SPD solve sensitivity. When a single ratio is plotted, the relevant $\lambda_*$ is reported in the caption (Betti-estimation regime by default in the shift tests).

## C. Proofs and Technical Lemmas

### Metric-Aware Defect and Basis Robustness

**Assumption C.1** (Metrics on $C^0$ and $C^2$ (optional)). Let $W_{0,\phi} \in \mathbb{R}^{n_0 \times n_0}$ and $W_{2,\phi} \in \mathbb{R}^{n_2 \times n_2}$ be SPD with uniform bounds

$$0 < w_{0,\min} I \preceq W_{0,\phi} \preceq w_{0,\max} I,$$
$$0 < w_{2,\min} I \preceq W_{2,\phi} \preceq w_{2,\max} I.$$

**Definition C.2** (Weighted Frobenius defect). Under Assumption C.1, define the metric-aware defect

$$D_{\mathrm{comp}}^{(W)}(\phi) := \big\| W_{2,\phi}^{1/2} (D_{1,\phi} D_{0,\phi}) W_{0,\phi}^{-1/2} \big\|_F.$$

**Lemma C.3** (Equivalence of raw and metric-aware defects).

*Under Assumption C.1,*

$$\sqrt{\frac{w_{2,\min}}{w_{0,\max}}}\, D_{\mathrm{comp}}(\phi) \;\le\; D_{\mathrm{comp}}^{(W)}(\phi) \;\le\;$$

$$\sqrt{\frac{w_{2,\max}}{w_{0,\min}}}\, D_{\mathrm{comp}}(\phi).$$

*Proof.* Using $\|MAN\|_F \le \|M\|_{\mathrm{op}}\|A\|_F\|N\|_{\mathrm{op}}$ and the SPD bounds,

$$\|W_{2,\phi}^{1/2}\|_{\mathrm{op}} \le \sqrt{w_{2,\max}}, \qquad \|W_{0,\phi}^{-1/2}\|_{\mathrm{op}} \le \frac{1}{\sqrt{w_{0,\min}}}.$$

Hence $D_{\mathrm{comp}}^{(W)}(\phi) \le \sqrt{w_{2,\max}/w_{0,\min}}\,\|D_{1,\phi}D_{0,\phi}\|_F$. For the lower bound, apply the same inequality to $D_{\mathrm{comp}}(\phi) = \|W_{2,\phi}^{-1/2}\,(W_{2,\phi}^{1/2}D_{1,\phi}D_{0,\phi}W_{0,\phi}^{-1/2})\,W_{0,\phi}^{1/2}\|_F$ and use $\|W_{2,\phi}^{-1/2}\|_{\mathrm{op}} \le 1/\sqrt{w_{2,\min}}$, $\|W_{0,\phi}^{1/2}\|_{\mathrm{op}} \le \sqrt{w_{0,\max}}$. $\qquad\square$

*Consequence.* All bounds stated in terms of $D_{\mathrm{comp}}(\phi)$ remain valid (with constants rescaled by the factors in Lemma C.3) when $D_{\mathrm{comp}}^{(W)}(\phi)$ is used instead. In FEEC/DEC, $W_{0,\phi}, W_{2,\phi}$ can be taken as the discrete Hodge stars (mass matrices) on 0- and 2-cochains.

**Remark on the Projection Inner Product.** The projection $P_{\ker(D_1)}$ is taken with respect to the Euclidean inner product on $C^1$ to admit a closed-form pseudoinverse representation and stable solver-based application. Metric-aware variants can be defined analogously by replacing $D_1^\dagger$ with the weighted pseudoinverse under a chosen $W_2$-inner product; the present work fixes the Euclidean projector for reproducibility and computational simplicity.

*Defect Selection for Specific Claims.* The algebraic results (Theorem 4.2) and Laplacian drift bounds (Theorem 4.4) hold for the raw Frobenius defect $D_{\mathrm{comp}}$. Chain-condition distance certificates scale with the smallest singular value $\sigma_*$; Laplacian spectral stability scales with operator bounds $B_D$ and weights. The metric-aware variant $D_{\mathrm{comp}}^{(W)}$ is strictly tighter for specific choices of weights but equivalent up to constants. All experiments in the main text report $D_{\mathrm{comp}}$ to avoid ambiguity.

**Lemma C.4** (Basic properties of $L_0$). *Under Assumption 3.3, $L_0$ is symmetric positive semidefinite. Moreover, for any $c^0 \in C^0$,*

$$(c^0)^\mathsf{T} L_0 c^0 = (D_0 c^0)^\mathsf{T} W_1 (D_0 c^0) \ge 0.$$

**Lemma C.5** (Kernel identity $\ker(L_0) = \ker(D_0)$). *Under Assumption 3.3, $\ker(L_0) = \ker(D_0)$.*

*Proof.* Since $L_0 = D_0^\mathsf{T} W_1 D_0$ and $W_1 \succ 0$, $L_0 c^0 = 0$ if and only if $D_0 c^0 = 0$. $\qquad\square$

**Theorem C.6** (Low-frequency eigenspace stability (Davis–Kahan)). *Assume $\widetilde{L}_{0,\phi} \succeq 0$ has eigenvalues $0 = \tilde{\mu}_1 = \cdots = \tilde{\mu}_{m_0} < \tilde{\mu}_{m_0+1}$ with $\tilde{\mu}_{m_0+1} \ge \lambda_* > 0$. Let $E := L_{0,\phi} - \widetilde{L}_{0,\phi}$ and assume $\|E\|_{\mathrm{op}} \le \delta(\phi)$ with $\delta(\phi) < \lambda_*/2$. Let $\widetilde{\Pi}(\phi)$ be the orthogonal projector onto $\ker(\widetilde{L}_{0,\phi})$ and let $\Pi_{<\lambda_*/2}(\phi)$ be the spectral projector of $L_{0,\phi}$ onto eigenvalues in $[0, \lambda_*/2)$. Then*

$$\|\Pi_{<\lambda_*/2}(\phi) - \widetilde{\Pi}(\phi)\|_{\mathrm{op}} \le \frac{\delta(\phi)}{\lambda_* - \delta(\phi)} \le \frac{2\,\delta(\phi)}{\lambda_*}.$$

*Proof.* By Weyl, the spectrum of $L_{0,\phi}$ lies within $\delta(\phi)$ of that of $\widetilde{L}_{0,\phi}$, so the cluster near 0 remains separated from the rest when $\delta(\phi) < \lambda_*/2$. Apply the Davis–Kahan $\sin\Theta$ theorem to the pair $(\widetilde{L}_{0,\phi}, L_{0,\phi})$ with the invariant subspace $\ker(\widetilde{L}_{0,\phi})$ and the complementary subspace associated with eigenvalues $\ge \lambda_*$, whose separation from 0 is at least $\lambda_* - \delta(\phi)$ under perturbation. This yields $\|\Pi - \widetilde{\Pi}\|_{\mathrm{op}} \le \|E\|_{\mathrm{op}}/(\lambda_* - \|E\|_{\mathrm{op}})$, and the final inequality uses $\delta(\phi) < \lambda_*/2$. $\qquad\square$

### Choosing and Validating the Gap Parameter $\lambda_*$

Assumption 4.6 requires a margin $\lambda_*$ separating the harmonic subspace (in the chosen boundary-condition regime) of the projected reference operator $\widetilde{L}_{0,\phi}$ from the rest of its spectrum. In practice $\lambda_*$ is chosen conservatively from the projected-reference spectra computed on the training distribution and then validated under the shift regimes in Section 6.

**Conservative choice.** Let $\tilde{\mu}_{m_0(\phi)+1}(\phi)$ denote the smallest positive eigenvalue of $\widetilde{L}_{0,\phi}$ in the Betti-estimation regime. Set

$$\lambda_* := \inf_{\phi \in \Phi_{\mathrm{train}}} \tilde{\mu}_{m_0(\phi)+1}(\phi),$$

optionally with a small safety factor (e.g. using a low quantile over seeds/checkpoints rather than the minimum) to reduce sensitivity to numerical noise while remaining conservative for the regimes of interest.

**Gap inheritance mechanism.** Since $\widetilde{L}_{0,\phi}$ is the Laplacian of the *projected* complex, its kernel dimension and gap are determined by the cohomology of the nearest chain-compatible reference. When the defect is small, Theorem C.6 ensures that the learned operator inherits this gap structure up to a perturbation of order $\delta(\phi)/\lambda_*$. The parameter $\lambda_*$ is thus a property of the *target topology* anchored by the projection, not an arbitrary hyperparameter.

**Validation under shift.** On each shift family (mesh OOD, topological distribution shift), $\tilde{\mu}_{m_0(\phi)+1}(\phi)$ is computed for

the projected reference operator, and the following are confirmed: (i) $\tilde{\mu}_{m_0+1}(\phi) \geq \lambda_*$ throughout the region where the criterion $\rho(\phi) < 1$ predicts stability; (ii) when topology changes induce gap collapse, the measured $\tilde{\mu}_{m_0+1}(\phi)$ decreases and the sufficient condition $\delta(\phi) < \lambda_*/2$ is violated, as expected. This directly distinguishes "certificate conservative but correct" from "certificate vacuous" cases.

**Behavior Under Spectral Gap Collapse.** If topological distribution shift collapses the gap so that $\tilde{\mu}_{m_0+1}(\phi)$ approaches 0 (for example, if a hole closes or a new component disconnects in the projected reference), then no fixed threshold can guarantee spectral-count stability. In this regime verify that the criterion triggers (via $\rho(\phi) \geq 1$), correctly signalling that Betti estimates from low-frequency counts are not reliable. This differentiates between confirmed failure, which alerts the user, and undetected failure, which produces a definite but incorrect Betti number.

**Lemma C.7** (Chain condition for the projected complex).
*For every $\phi$,*

$$\widetilde{D}_{1,\phi}\,\widetilde{D}_{0,\phi} = D_{1,\phi}\widetilde{D}_{0,\phi} = 0.$$

*Thus the sequence $C^0 \xrightarrow{\widetilde{D}_{0,\phi}} C^1 \xrightarrow{\widetilde{D}_{1,\phi}} C^2$ is a cochain complex.*

**Lemma C.8** (Control of the projection perturbation). *Under Assumptions 3.11 and 3.12, for every $\phi$,*

$$\left\| D_{0,\phi} - \widetilde{D}_{0,\phi} \right\|_{\mathrm{op}} \leq \frac{1}{\sigma_*}\, D_{\mathrm{comp}}(\phi),$$

$$\left\| D_{0,\phi} - \widetilde{D}_{0,\phi} \right\|_{\mathrm{F}} \leq \frac{1}{\sigma_*}\, D_{\mathrm{comp}}(\phi).$$

**Lemma C.9** (Regularized projector error bound). *Let $D_1$ be any matrix with $\sigma^+_{\min}(D_1) \geq \sigma_* > 0$ and define*

$$P := I - D_1^\dagger D_1,$$
$$P_\eta := I - D_1^\top (D_1 D_1^\top + \eta I)^{-1} D_1, \qquad \eta > 0.$$

*Then*
$$\|P_\eta - P\|_{\mathrm{op}} \leq \frac{\eta}{\sigma_*^2 + \eta} \leq \frac{\eta}{\sigma_*^2}.$$

*Proof.* On $\mathrm{range}(D_1^\top)$, the orthogonal projector $P$ annihilates, while $P_\eta$ acts with shrinkage factors $\eta/(\sigma_i^2 + \eta)$ on singular directions with singular value $\sigma_i > 0$. The operator norm is therefore maximized at the smallest nonzero singular value, giving $\|P_\eta - P\|_{\mathrm{op}} \leq \eta/(\sigma^+_{\min}(D_1)^2 + \eta) \leq \eta/(\sigma_*^2 + \eta)$. □

*Remark on Regularization.* Regularising the projector with $\eta > 0$ smooths the optimization landscape and ensures robustness against rank changes in $D_1$. The bound above

shows this comes at the cost of a controlled bias in the certificate. In the main experiments we use the unregularized orthogonal projector (via sparse solves on the range) unless otherwise noted.

**Proof of Lemma C.4**

**Notation.** In purely algebraic proofs below, the parameter $\phi$ is fixed and subscripts $\phi$ are dropped.

Symmetry follows from $W_1 = W_1^\top$:

$$(L_0)^\top = (D_0^\top W_1 D_0)^\top = D_0^\top W_1^\top D_0 = L_0.$$

For positive semidefiniteness, let $u \in C^0$. Then

$$u^\top L_0 u = u^\top D_0^\top W_1 D_0 u = (D_0 u)^\top W_1 (D_0 u).$$

Since $W_1$ is SPD (Assumption 3.3), $v^\top W_1 v \geq \lambda_{\min}(W_1) \|v\|^2 \geq 0$ for any vector $v$. Setting $v = D_0 u$ yields the result.

**Proof of Lemma 3.6**

By Lemma C.4, $u^\top L_0 u = (D_0 u)^\top W_1 (D_0 u)$. Since $W_1$ is SPD, the quadratic form is zero if and only if $D_0 u = 0$. Thus $\ker(L_0) = \ker(D_0)$.

**Proof of Lemma C.7**

Recall $\widetilde{D}_{1,\phi} := D_{1,\phi}$ and $\widetilde{D}_{0,\phi} := P_{\ker(D_{1,\phi})} D_{0,\phi}$. Observe that $D_{1,\phi}\widetilde{D}_{0,\phi} = D_{1,\phi} P_{\ker(D_{1,\phi})} D_{0,\phi}$. By definition, the image of $P_{\ker(D_{1,\phi})}$ lies in $\ker(D_{1,\phi})$, so $D_{1,\phi}$ annihilates it. Thus $D_{1,\phi}\widetilde{D}_{0,\phi} = 0$.

**Proof of Lemma C.8**

Since $\widetilde{D}_{0,\phi}$ is the orthogonal projection of $D_{0,\phi}$ onto the subspace $\mathcal{K}_\phi = \ker(D_{1,\phi})$, the distance $\left\| D_{0,\phi} - \widetilde{D}_{0,\phi} \right\|_{\mathrm{F}}$ is the distance from $D_{0,\phi}$ to $\mathcal{K}_\phi$. Let $E = D_{0,\phi} - \widetilde{D}_{0,\phi}$. Then $E \perp \ker(D_{1,\phi})$, i.e., $E \in \mathrm{im}(D_{1,\phi}^\top)$ (in finite dimensions). Consider the quantity $D_{\mathrm{comp}}(\phi) = \|D_{1,\phi} D_{0,\phi}\|_{\mathrm{F}}$. Since $\widetilde{D}_{0,\phi} \in \ker(D_{1,\phi})$, we have $D_{1,\phi} D_{0,\phi} = D_{1,\phi}(D_{0,\phi} - \widetilde{D}_{0,\phi}) = D_{1,\phi} E$. It remains to bound $\|E\|_{\mathrm{F}}$ in terms of $\|D_{1,\phi} E\|_{\mathrm{F}}$. Since $E \in (\ker D_{1,\phi})^\perp$, restrict $D_{1,\phi}$ to $(\ker D_{1,\phi})^\perp$; on this subspace the map is injective. Its minimum singular value on this complement is $\sigma_{\min}(D_{1,\phi}|_{\ker(D_{1,\phi})^\perp})$. Let this be $\sigma_*(\phi)$. Then $\|D_{1,\phi} E\|_{\mathrm{F}} \geq \sigma_*(\phi) \|E\|_{\mathrm{F}}$. Assumption 3.12 posits a uniform lower bound $\sigma_* \leq \sigma_*(\phi)$. Thus $D_{\mathrm{comp}}(\phi) \geq \sigma_* \|E\|_{\mathrm{F}}$, or $\|E\|_{\mathrm{F}} \leq \frac{1}{\sigma_*} D_{\mathrm{comp}}(\phi)$. The same argument applies to the operator norm: $\|E\|_{\mathrm{op}} \leq \frac{1}{\sigma_*} \|D_{1,\phi} E\|_{\mathrm{op}} = \frac{1}{\sigma_*} \|D_{1,\phi} D_{0,\phi}\|_{\mathrm{op}}$. Note that $\|\cdot\|_{\mathrm{op}} \leq \|\cdot\|_{\mathrm{F}}$, so the Frobenius bound is strictly stronger (and sufficient).

**Lemma C.10** (Structured Laplacian drift decomposition).

*Let $\Delta D_{0,\phi} := D_{0,\phi} - \widetilde{D}_{0,\phi}$. Then*

$$\begin{aligned}
L_{0,\phi} - \widetilde{L}_{0,\phi} &= \widetilde{D}_{0,\phi}^\top W_{1,\phi}\,\Delta D_{0,\phi} \\
&\quad + \Delta D_{0,\phi}^\top W_{1,\phi}\,\widetilde{D}_{0,\phi} \\
&\quad + \Delta D_{0,\phi}^\top W_{1,\phi}\,\Delta D_{0,\phi}.
\end{aligned}$$

**Proof of Lemma C.10**

Let $\Delta := D_{0,\phi} - \widetilde{D}_{0,\phi}$. Direct calculation:

$$\begin{aligned}
L_{0,\phi} &= (\widetilde{D}_{0,\phi} + \Delta)^\top W_{1,\phi}(\widetilde{D}_{0,\phi} + \Delta) \\
&= \widetilde{D}_{0,\phi}^\top W_{1,\phi}\widetilde{D}_{0,\phi} \\
&\quad + \widetilde{D}_{0,\phi}^\top W_{1,\phi}\Delta + \Delta^\top W_{1,\phi}\widetilde{D}_{0,\phi} \\
&\quad + \Delta^\top W_{1,\phi}\Delta \\
&= \widetilde{L}_{0,\phi} + \text{cross terms} + \text{quadratic term}.
\end{aligned}$$

**Proof of Theorem 4.2**

Fix $\phi$ and write $D_0 := D_{0,\phi}$, $D_1 := D_{1,\phi}$, $\widetilde{D}_0 := \widetilde{D}_{0,\phi}$. Let $\mathcal{E}_\phi := \{B : D_1 B = 0\}$ and note that $\mathcal{E}_\phi$ is a closed linear subspace of the Frobenius inner-product space of matrices $B : C^0 \to C^1$. By definition, $\widetilde{D}_0 = P_{\ker(D_1)}D_0$ applies the orthogonal projector columnwise, hence $\widetilde{D}_0 \in \mathcal{E}_\phi$ and $D_0 - \widetilde{D}_0 \perp \mathcal{E}_\phi$ in the Frobenius inner product. Therefore $\widetilde{D}_0$ is the unique Frobenius-orthogonal projection of $D_0$ onto $\mathcal{E}_\phi$. This establishes projection optimality and identifies $D_{\mathrm{proj}}(\phi) = \|D_0 - \widetilde{D}_0\|_F$.

For the inequalities in (ii), since $D_1 \widetilde{D}_0 = 0$,

$$D_1 D_0 = D_1(D_0 - \widetilde{D}_0).$$

The upper bound follows by submultiplicativity: $\|D_1 D_0\|_F \leq \|D_1\|_{\mathrm{op}}\|D_0 - \widetilde{D}_0\|_F$. For the lower bound, note that $D_0 - \widetilde{D}_0 \in (\ker D_1)^\perp$, and on $(\ker D_1)^\perp$ the smallest singular value is $\sigma^+_{\min}(D_1) \geq \sigma_*$ (Assumption 3.12), hence $\|D_1(D_0 - \widetilde{D}_0)\|_F \geq \sigma_*\|D_0 - \widetilde{D}_0\|_F$. Rearranging gives $\frac{1}{\|D_1\|_{\mathrm{op}}}D_{\mathrm{comp}}(\phi) \leq D_{\mathrm{proj}}(\phi) \leq \frac{1}{\sigma_*}D_{\mathrm{comp}}(\phi)$. $\square$

**Proof of Theorem 4.4**

Using Lemma C.10 with $\Delta := D_{0,\phi} - \widetilde{D}_{0,\phi}$,

$$\begin{aligned}
\|L_{0,\phi} - \widetilde{L}_{0,\phi}\|_{\mathrm{op}} &\leq 2\|\widetilde{D}_{0,\phi}\|_{\mathrm{op}}\,\|W_{1,\phi}\|_{\mathrm{op}}\,\|\Delta\|_{\mathrm{op}} \\
&\quad + \|W_{1,\phi}\|_{\mathrm{op}}\,\|\Delta\|^2_{\mathrm{op}}.
\end{aligned}$$

Since $\widetilde{D}_{0,\phi} = P_{\ker(D_{1,\phi})}D_{0,\phi}$ and $P_{\ker(D_{1,\phi})}$ is an orthogonal projector, $\|\widetilde{D}_{0,\phi}\|_{\mathrm{op}} \leq \|D_{0,\phi}\|_{\mathrm{op}} \leq B_D$ (Assumption 3.11). Also $\|W_{1,\phi}\|_{\mathrm{op}} \leq w_{\max}$ (Assumption 3.9) and

Lemma C.8 gives $\|\Delta\|_{\mathrm{op}} \leq \sigma_*^{-1}D_{\mathrm{comp}}(\phi)$. Thus

$$\begin{aligned}
\|L_{0,\phi} - \widetilde{L}_{0,\phi}\|_{\mathrm{op}} &\leq \frac{2w_{\max}B_D}{\sigma_*}\,D_{\mathrm{comp}}(\phi) \\
&\quad + \frac{w_{\max}}{\sigma_*^2}\,D_{\mathrm{comp}}(\phi)^2,
\end{aligned}$$

i.e. $\delta(\phi) = A D_{\mathrm{comp}}(\phi) + B D_{\mathrm{comp}}(\phi)^2$ with $A = 2w_{\max}B_D/\sigma_*$ and $B = w_{\max}/\sigma_*^2$ under Assumptions 3.11, 3.9, and 3.12.

**Operator-Norm Defect Variant (Tighter Certificate)**

Define the operator-norm defect

$$D_{\mathrm{op}}(\phi) := \|D_{1,\phi}D_{0,\phi}\|_{\mathrm{op}}.$$

Since $\|\cdot\|_{\mathrm{op}} \leq \|\cdot\|_F$, one always has $D_{\mathrm{op}}(\phi) \leq D_{\mathrm{comp}}(\phi)$.

**Tighter projection perturbation bound.** The proof of Lemma C.8 yields

$$\|D_{0,\phi} - \widetilde{D}_{0,\phi}\|_{\mathrm{op}} \leq \frac{1}{\sigma_*}\,D_{\mathrm{op}}(\phi),$$

and the Laplacian drift proof gives the sharper budget

$$\|L_{0,\phi} - \widetilde{L}_{0,\phi}\|_{\mathrm{op}} \leq A\,D_{\mathrm{op}}(\phi) + B\,D_{\mathrm{op}}(\phi)^2$$

with the same constants $A, B$ as Theorem 4.4. All downstream results (Weyl, Davis–Kahan, SPD forward error) hold verbatim with $\delta(\phi)$ defined using $D_{\mathrm{op}}$.

**Estimation.** $D_{\mathrm{op}}(\phi)$ is estimated by power iteration on $(D_{1,\phi}D_{0,\phi})^\top(D_{1,\phi}D_{0,\phi})$ using matrix-free products, at a cost comparable to a few Hutchinson probes.

**Practical Recommendation.** While $D_{\mathrm{op}}$ provides the tightest theoretical certificate, $D_{\mathrm{comp}}$ is cheaper to estimate via trace estimators and provides an upper bound. A practical recommendation is to monitor $D_{\mathrm{comp}}$ during training (for cheap gradients) and checking $D_{\mathrm{op}}$ for final validation or criterion certification.

**Lemma C.11** (Weyl's inequality). *Let $A, B \in \mathbb{R}^{n \times n}$ be symmetric with eigenvalues $\alpha_1 \leq \cdots \leq \alpha_n$ and $\beta_1 \leq \cdots \leq \beta_n$, respectively. If $\|A - B\|_{\mathrm{op}} \leq \delta$, then*

$$|\beta_i - \alpha_i| \leq \delta \quad \text{for all } i = 1, \ldots, n.$$

**Proof of Lemma C.11**

For symmetric matrices $A$ and $B$, Weyl's inequalities state $\alpha_i + \lambda_{\min}(B - A) \leq \beta_i \leq \alpha_i + \lambda_{\max}(B - A)$. Since $B - A$ is symmetric, $|\lambda(B - A)| \leq \|B - A\|_{\mathrm{op}}$, giving the result.

**Lemma C.12** (Low-frequency multiplicity stability under a gapped reference). *Let $L^{\mathrm{ref}} \succeq 0$ be symmetric with ordered*

eigenvalues $0 = \nu_1 = \cdots = \nu_{\beta_0} < \nu_{\beta_0+1}$ and suppose $\nu_{\beta_0+1} \geq \lambda_{\mathrm{ref}} > 0$. If $\|\widetilde{L}_{0,\phi} - L^{\mathrm{ref}}\|_{\mathrm{op}} \leq \varepsilon_E$ with $\varepsilon_E < \lambda_{\mathrm{ref}}/2$, then

$$\#\{\, i : \mu_i(\phi) < \lambda_{\mathrm{ref}}/2 \,\} = \beta_0,$$
$$\mu_{\beta_0+1}(\phi) \geq \lambda_{\mathrm{ref}} - \varepsilon_E \; \geq \; \tfrac{1}{2}\lambda_{\mathrm{ref}}.$$

*Proof.* By Weyl's inequality (Lemma C.11), each eigenvalue of $\widetilde{L}_{0,\phi}$ deviates from the corresponding eigenvalue of $L^{\mathrm{ref}}$ by at most $\varepsilon_E$. In particular, $\mu_i(\phi) \leq \varepsilon_E$ for $i \leq \beta_0$ and $\mu_{\beta_0+1}(\phi) \geq \lambda_{\mathrm{ref}} - \varepsilon_E$. Since $\varepsilon_E < \lambda_{\mathrm{ref}}/2$, exactly $\beta_0$ eigenvalues lie in $[0, \lambda_{\mathrm{ref}}/2)$, hence the thresholded multiplicity equals $\beta_0$ and the stated gap follows. $\square$

## Proof of Theorem 4.9

Let $\mu_i(\phi)$ and $\lambda_i(\phi)$ denote the eigenvalues of $\widetilde{L}_{0,\phi}$ and $L_{0,\phi}$. By Weyl (Lemma C.11), $|\lambda_i - \mu_i| \leq \left\|L_{0,\phi} - \widetilde{L}_{0,\phi}\right\|_{\mathrm{op}} \leq \delta$. Given the spectral gap $\mu_{k+1} \geq \lambda_*$, the perturbed eigenvalues satisfy $\lambda_{k+1} \geq \mu_{k+1} - \delta \geq \lambda_* - \lambda_*/2 = \lambda_*/2$. The zero eigenvalues $\mu_1 = \cdots = \mu_k = 0$ become $\lambda_i \leq \delta < \lambda_*/2$. Thus the count below $\lambda_*/2$ is preserved.

**Corollary C.13** (Complex-defect threshold). *Under the assumptions of Theorem 4.9, there exists $\varepsilon_* > 0$ such that, if $D_{\mathrm{comp}}(\phi) < \varepsilon_*$, then*

$$m_{<\lambda_*/2}(\phi) = \dim \ker(\widetilde{L}_{0,\phi}).$$

*One may take $\varepsilon_*$ to be the smallest positive solution of $A\varepsilon + B\varepsilon^2 = \lambda_*/2$, i.e. if $B > 0$, one may take $\varepsilon_* = \frac{-A+\sqrt{A^2+2B\lambda_*}}{2B}$; if $B = 0$, take $\varepsilon_* = \lambda_*/(2A)$.*

### C.1. Forward Error Analysis Statements

**Assumption C.14** (Positive-definite setting). Let $\widetilde{L}, L \in \mathbb{R}^{n \times n}$ be symmetric positive definite with eigenvalues

$$0 < \tilde{\lambda}_1 \leq \cdots \leq \tilde{\lambda}_n, \qquad 0 < \lambda_1 \leq \cdots \leq \lambda_n.$$

Assume:

(i) $\tilde{\lambda}_1 \geq \lambda_* > 0$;

(ii) $\left\|L - \widetilde{L}\right\|_{\mathrm{op}} \leq \varepsilon$ with $\varepsilon < \lambda_*/2$.

**Theorem C.15** (Forward error bound in the SPD case). *Under Assumption C.14, for any $f \in \mathbb{R}^n$, let $\tilde{u} = \widetilde{L}^{-1}f$ and $u = L^{-1}f$. Then*

$$\frac{\|u - \tilde{u}\|}{\|\tilde{u}\|} \leq \frac{\varepsilon}{\lambda_* - \varepsilon} \leq \frac{2\varepsilon}{\lambda_*}.$$

**Corollary C.16** (Solution error bound via cochain-compatibility defect). *Let $L_{0,\phi}^{\mathrm{SPD}}$ and $\widetilde{L}_{0,\phi}^{\mathrm{SPD}}$ denote the Dirichlet-restricted operators from Appendix B. Suppose that, after*

Dirichlet restriction to interior DOFs, the smallest eigenvalue of the resulting $\widetilde{L}_{0,\phi}^{\mathrm{SPD}}$ is at least $\lambda_*^{\mathrm{SPD}} > 0$. Define

$$\varepsilon(\phi) := A\, D_{\mathrm{comp}}(\phi) + B\, D_{\mathrm{comp}}(\phi)^2,$$

*where $A = 2w_{\max}B_D/\sigma_*$ and $B = w_{\max}/\sigma_*^2$ are the constants from Theorem 4.4. Because restriction is a contraction, $\|R(L_{0,\phi} - \widetilde{L}_{0,\phi})R^\top\|_{\mathrm{op}} \leq \|L_{0,\phi} - \widetilde{L}_{0,\phi}\|_{\mathrm{op}}$, so the same $\varepsilon(\phi)$ upper-bounds the restricted perturbation. If $\varepsilon(\phi) < \lambda_*^{\mathrm{SPD}}/2$, then for any right-hand side $f$, the solutions $u_\phi$ and $\tilde{u}_\phi$ of*

$$L_{0,\phi}^{\mathrm{SPD}} u_\phi = f, \qquad \widetilde{L}_{0,\phi}^{\mathrm{SPD}} \tilde{u}_\phi = f,$$

*satisfy*

$$\frac{\|u_\phi - \tilde{u}_\phi\|}{\|\tilde{u}_\phi\|} \leq \frac{\varepsilon(\phi)}{\lambda_*^{\mathrm{SPD}} - \varepsilon(\phi)} \leq \frac{2\,\varepsilon(\phi)}{\lambda_*^{\mathrm{SPD}}}.$$

### Proof of Theorem C.15

Let $E := L - \widetilde{L}$. Then

$$u - \tilde{u} = L^{-1}f - \widetilde{L}^{-1}f = L^{-1}(\widetilde{L}-L)\widetilde{L}^{-1}f = -L^{-1}E\,\tilde{u}.$$

Hence

$$\|u - \tilde{u}\| \leq \|L^{-1}\|_{\mathrm{op}}\|E\|_{\mathrm{op}}\|\tilde{u}\|.$$

By Weyl's inequality applied to $(L, \widetilde{L})$ and Assumption C.14(i)–(ii), $\lambda_{\min}(L) \geq \tilde{\lambda}_1 - \|E\|_{\mathrm{op}} \geq \lambda_* - \varepsilon > 0$, so $\|L^{-1}\|_{\mathrm{op}} = 1/\lambda_{\min}(L) \leq 1/(\lambda_* - \varepsilon)$. Therefore

$$\frac{\|u - \tilde{u}\|}{\|\tilde{u}\|} \leq \frac{\varepsilon}{\lambda_* - \varepsilon} \leq \frac{2\varepsilon}{\lambda_*},$$

using $\varepsilon < \lambda_*/2$ for the final inequality. $\square$

**Corollary C.17** (Conditioning bound). *Under the same assumptions, the condition number of the learned SPD operator satisfies*

$$\kappa(L) \leq \frac{\lambda_{\max}(\widetilde{L}) + \varepsilon}{\lambda_* - \varepsilon}.$$

*Since $\varepsilon(\phi) \approx AD_{\mathrm{comp}}$, minimizing the defect explicitly controls the worst-case conditioning drift.*

### Proof of Corollary 4.10

Apply Theorem 4.9 with the assumption that $\dim \ker(\widetilde{L}_{0,\phi}) = \beta_0$.

### Proof of Theorem 5.6

Uses Strang's first lemma (Lemma C.18). The consistency term is bounded by Lemma C.19, which splits the error into Defect part + Manifold deviation part.

## Coefficient-to-Function Norm Map and Strang Instantiation

The norm equivalences used to pass between coefficient vectors and FEEC energy norms in Lemma 5.3 and Theorem 5.6 are made explicit here.

**Discrete energy norms.** Let $u_h = \sum_i u_i \varphi_i \in V_h^0$ and identify $\mathbf{u} = (u_i)_i \in \mathbb{R}^{n_0}$. Let $M_{0,h}$ denote the degree-0 mass matrix and $L_{0,h}^{\text{ref}}$ the reference stiffness/Laplacian matrix. Define

$$\|u_h\|_{L^2(\Omega)}^2 := \mathbf{u}^\top M_{0,h} \mathbf{u}, \qquad \|u_h\|_{E,h}^2 := \mathbf{u}^\top L_{0,h}^{\text{ref}} \mathbf{u}.$$

**Uniform norm equivalence on shape-regular families.** On a shape-regular mesh family, there exist constants $c_0^\pm, c_1^\pm > 0$ independent of $h$ such that

$$c_0^- \|\mathbf{u}\|_2 \le \|u_h\|_{L^2(\Omega)} \le c_0^+ \|\mathbf{u}\|_2,$$
$$c_1^- \|\mathbf{u}\|_{L_{0,h}^{\text{ref}}} \le \|u_h\|_{H^1(\Omega)} \le c_1^+ \|\mathbf{u}\|_{L_{0,h}^{\text{ref}}}.$$

where $\|\mathbf{u}\|_{L_{0,h}^{\text{ref}}}^2 := \mathbf{u}^\top L_{0,h}^{\text{ref}} \mathbf{u}$.

**Bilinear-form perturbation bound in coefficient form.** For any $u_h, v_h \in V_h^0$ with coefficient vectors $\mathbf{u}, \mathbf{v}$,

$$\begin{aligned}
\left| a_{\phi,h}(u_h, v_h) - a_h(u_h, v_h) \right| &= \left| \mathbf{u}^\top (L_{0,\phi,h} - L_{0,h}^{\text{ref}}) \mathbf{v} \right| \\
&\le \|L_{0,\phi,h} - L_{0,h}^{\text{ref}}\|_{\text{op}} \|\mathbf{u}\|_2 \\
&\quad \times \|\mathbf{v}\|_2.
\end{aligned}$$

Using the $L^2$-equivalence $\|\mathbf{u}\|_2 \le (c_0^-)^{-1} \|u_h\|_{L^2(\Omega)}$ and the continuous embedding $V \hookrightarrow L^2(\Omega)$ yields

$$\begin{aligned}
\left| a_{\phi,h}(u_h, v_h) - a_h(u_h, v_h) \right| &\le C_{\text{op}} \|L_{0,\phi,h} - L_{0,h}^{\text{ref}}\|_{\text{op}} \\
&\quad \times \|u_h\|_V \|v_h\|_V.
\end{aligned}$$

with $C_{\text{op}}$ depending only on the mesh-shape regularity and the chosen FEEC basis (not on $h$ or $\phi$). Splitting $L_{0,\phi,h} - L_{0,h}^{\text{ref}} = (L_{0,\phi,h} - \widetilde{L}_{0,\phi,h}) + (\widetilde{L}_{0,\phi,h} - L_{0,h}^{\text{ref}})$ and applying Theorem 4.4 gives the defect-controlled term plus $\mathcal{R}_E(\phi)$ used in Lemma 5.3.

**Strang lemma used.** Lemma C.18 is Strang's first lemma applied with the discrete coercivity constant $\alpha_0$ from Assumption 5.4, and with the consistency term bounded by the inequality above together with Theorem 4.4. The remaining "mesh/quadrature terms" are precisely the FEEC interpolation and quadrature consistency terms appearing in the reference bound for $a_h$ (e.g. (Arnold et al., 2006)).

**Lemma C.18** (Strang lemma for learned bilinear forms).
*Assume $a$ is continuous and coercive on $V$ with constants $\gamma, \alpha > 0$, and assume Assumption 5.4 and standard FEEC interpolation estimates. Then there exists $C > 0$, independent of $h$ and $\phi$, such that*

$$\|u - u_{\phi,h}\|_V \le C \Bigg( \|u - I_h u\|_V$$
$$+ \sup_{w_h \in V_h^0 \setminus \{0\}} \frac{|a(I_h u, w_h) - a_{\phi,h}(I_h u, w_h)|}{\|w_h\|_V} \Bigg)$$

*(mesh/quadrature terms).*

### Proof of Lemma C.18

The proof follows the FEEC consistency framework (Arnold et al., 2006).

**Lemma C.19** (Operator-learning consistency term). *Under the assumptions of Lemma 5.3, there exist constants $C_1, C_2 > 0$ independent of $h$ and $\phi$ such that*

$$\left\| \left( a_h - a_{\phi,h} \right)\left( u_h^\star, \cdot \right) \right\|_{(V_h^0)^*} \le \left( C_1\, \delta(\phi) + C_2\, \mathcal{R}_E(\phi) \right)$$
$$\times \|u_h^\star\|_V .$$

*where $\delta(\phi) := A D_{\text{comp}}(\phi) + B D_{\text{comp}}(\phi)^2$ and $\mathcal{R}_E(\phi) := \left\| \widetilde{L}_{0,\phi,h} - L_{0,h}^{\text{ref}} \right\|_{\text{op}}$.*

### Proof of Lemma C.19

Follows from bilinear form deviation bounds.

### Proof of Lemma G.7

(i) Since the columns of $K$ lie in $\ker(D_1^{\text{ref}})$, $D_1^{\text{ref}} K = 0$, so $D_1^{\text{ref}}(KR) = (D_1^{\text{ref}} K)R = 0 \cdot R = 0$ for all $R$ and hence $B = KR \in \mathcal{M}_{\text{cc}}$.

(ii) Let $B \in \mathcal{M}_{\text{cc}}$. Each column $b_j$ of $B$ satisfies $D_1^{\text{ref}} b_j = 0$, so $b_j \in \ker(D_1^{\text{ref}})$ and hence $b_j = K r_j$ for some unique $r_j \in \mathbb{R}^r$ (since $K$ has orthonormal columns spanning the kernel). Collecting the $r_j$ as columns yields a unique matrix $R \in \mathbb{R}^{r \times n_0}$ with $B = KR$.

### Proof of Proposition G.9

Using $D_{1,\phi} = D_1^{\text{ref}}$ and $D_{0,\phi} = KR(\phi)$,

$$D_{1,\phi} D_{0,\phi} = D_1^{\text{ref}} K R(\phi) = 0 \cdot R(\phi) = 0,$$

since $K$ spans $\ker(D_1^{\text{ref}})$. Thus the sequence is a cochain complex

(satisfies $D_1 D_0 = 0$) and $D_{\text{comp}}(\phi) = \|D_{1,\phi} D_{0,\phi}\|_{\text{F}} = 0$. The projector $P_{\ker(D_1^{\text{ref}})}$ acts as $KK^\top$, and $D_{0,\phi}$ already has range contained in $\ker(D_1^{\text{ref}})$, so $P_{\ker(D_1^{\text{ref}})} D_{0,\phi} = D_{0,\phi}$ and $\widetilde{D}_{0,\phi} = D_{0,\phi}$. Hence $D_{\text{proj}}(\phi) = \left\| D_{0,\phi} - \widetilde{D}_{0,\phi} \right\|_{\text{F}} = 0$.

**Proof of Proposition G.4**

Write

$$\Delta_{k,\phi,h} - \widetilde{\Delta}_{k,\phi,h} = U_\phi + D_\phi,$$

where

$$U_\phi := D_{k,\phi}^\mathsf{T} W_{k+1,\phi} D_{k,\phi} - \widetilde{D}_{k,\phi}^\mathsf{T} W_{k+1,\phi} \widetilde{D}_{k,\phi},$$

and

$$\begin{aligned} D_\phi := {} & W_{k,\phi}^{-1} D_{k-1,\phi} W_{k-1,\phi} D_{k-1,\phi}^\mathsf{T} \\ & - W_{k,\phi}^{-1} \widetilde{D}_{k-1,\phi} W_{k-1,\phi} \widetilde{D}_{k-1,\phi}^\mathsf{T}. \end{aligned}$$

The upward part $U_\phi$ is bounded by the degree-$k$ Laplacian stability result:

$$\|U_\phi\|_{\mathrm{op}} \le A_k D_{k,\mathrm{comp}}(\phi) + B_k D_{k,\mathrm{comp}}(\phi)^2.$$

For the downward part, set

$$\begin{aligned} B_{k-1,\phi} &:= W_{k-1,\phi}^{1/2} D_{k-1,\phi}^\mathsf{T} W_{k,\phi}^{-1/2}, \\ \widetilde{B}_{k-1,\phi} &:= W_{k-1,\phi}^{1/2} \widetilde{D}_{k-1,\phi}^\mathsf{T} W_{k,\phi}^{-1/2}. \end{aligned}$$

Then

$$D_\phi = W_{k,\phi}^{-1/2} \big( B_{k-1,\phi}^\mathsf{T} B_{k-1,\phi} - \widetilde{B}_{k-1,\phi}^\mathsf{T} \widetilde{B}_{k-1,\phi} \big) W_{k,\phi}^{1/2}.$$

Hence

$$\begin{aligned} \|D_\phi\|_{\mathrm{op}} &= \left\| W_{k,\phi}^{-1/2} \big( B_{k-1,\phi}^\mathsf{T} B_{k-1,\phi} - \widetilde{B}_{k-1,\phi}^\mathsf{T} \widetilde{B}_{k-1,\phi} \big) W_{k,\phi}^{1/2} \right\|_{\mathrm{op}} \\ &\le \left\| W_{k,\phi}^{-1/2} \right\|_{\mathrm{op}} \left\| B_{k-1,\phi}^\mathsf{T} B_{k-1,\phi} - \widetilde{B}_{k-1,\phi}^\mathsf{T} \widetilde{B}_{k-1,\phi} \right\|_{\mathrm{op}} \\ &\quad \times \left\| W_{k,\phi}^{1/2} \right\|_{\mathrm{op}}. \end{aligned}$$

Since $W_{k,\phi}$ is SPD with eigenvalues between $w_{\min,k}$ and $w_{\max,k}$,

$$\left\| W_{k,\phi}^{\pm 1/2} \right\|_{\mathrm{op}} \le \max\{w_{\min,k}^{-1/2}, w_{\max,k}^{1/2}\} =: C_W.$$

Applying degree-wise Laplacian stability to the factorised pair $(B_{k-1,\phi}, D_{k,\phi})$ and its projection $(\widetilde{B}_{k-1,\phi}, D_{k,\phi})$ produces constants $\hat{A}_{k-1}, \hat{B}_{k-1} > 0$ and a corresponding degree-$(k-1)$ defect $D_{k-1,\mathrm{comp}}(\phi)$ such that

$$\begin{aligned} \left\| B_{k-1,\phi}^\mathsf{T} B_{k-1,\phi} - \widetilde{B}_{k-1,\phi}^\mathsf{T} \widetilde{B}_{k-1,\phi} \right\|_{\mathrm{op}} &\le \hat{A}_{k-1} D_{k-1,\mathrm{comp}}(\phi) \\ &\quad + \hat{B}_{k-1} D_{k-1,\mathrm{comp}}(\phi)^2. \end{aligned}$$

Thus

$$\|D_\phi\|_{\mathrm{op}} \le C_W^2 \big( \hat{A}_{k-1} D_{k-1,\mathrm{comp}}(\phi) + \hat{B}_{k-1} D_{k-1,\mathrm{comp}}(\phi)^2 \big).$$

Combining the bounds on $U_\phi$ and $D_\phi$ and setting

$$C_k^{(1)} := A_k + C_W^2 \hat{A}_{k-1}, \qquad C_k^{(2)} := B_k + C_W^2 \hat{B}_{k-1}$$

yields

$$\left\| \Delta_{k,\phi,h} - \widetilde{\Delta}_{k,\phi,h} \right\|_{\mathrm{op}} \le 2C_k^{(1)} D_{\max}(\phi) + 4C_k^{(2)} D_{\max}(\phi)^2,$$
$$D_{\max}(\phi) := \max\{D_{k-1,\mathrm{comp}}(\phi), D_{k,\mathrm{comp}}(\phi)\}.$$

as claimed.

**Proof of Theorem G.6**

Let $\mu_i^{(1)}(\phi)$ and $\lambda_i^{(1)}(\phi)$ denote the eigenvalues of $\widetilde{\Delta}_{1,\phi,h}$ and $\Delta_{1,\phi,h}$, respectively, ordered nondecreasingly. Proposition G.4 with $k = 1$ gives

$$\left\| \Delta_{1,\phi,h} - \widetilde{\Delta}_{1,\phi,h} \right\|_{\mathrm{op}} \le \Xi_1(\phi).$$

Under Assumption G.5 the first $\beta_1(K_h)$ eigenvalues of $\widetilde{\Delta}_{1,\phi,h}$ are zero, and the $(\beta_1(K_h) + 1)$-th eigenvalue is at least $\lambda_1^*$. Lemma C.11 implies

$$|\lambda_i^{(1)}(\phi) - \mu_i^{(1)}(\phi)| \le \Xi_1(\phi) \quad \text{for all } i = 1, \dots, n.$$

If $\Xi_1(\phi) < \lambda_1^*/2$, then for $1 \le i \le \beta_1(K_h), 0 \le \lambda_i^{(1)}(\phi) \le \Xi_1(\phi) < \lambda_1^*/2$, while for $i > \beta_1(K_h)$,

$$\lambda_i^{(1)}(\phi) \ge \mu_i^{(1)}(\phi) - \Xi_1(\phi) \ge \lambda_1^* - \lambda_1^*/2 = \lambda_1^*/2.$$

Hence exactly $\beta_1(K_h)$ eigenvalues of $\Delta_{1,\phi,h}$ lie in $[0, \lambda_1^*/2)$, i.e. $\widehat{\beta}_1^{\mathrm{spec}}(\Delta_{1,\phi,h}; \lambda_1^*/2) = \beta_1(K_h)$.

## D. Abstract Factorised Formulation (Operator-Level View)

Let $V, W, Z$ be finite-dimensional real Hilbert spaces. For each parameter $\phi$, let

$$B_\phi : V \to W, \qquad C_\phi : W \to Z$$

be linear maps and let $W_\phi \in \mathbb{R}^{\dim W \times \dim W}$ be SPD. Define a *factorised operator*

$$A_\phi := B_\phi^\mathsf{T} W_\phi B_\phi : V \to V$$

and a *cochain-compatibility defect*

$$D_{\mathrm{comp}}(\phi) := \|C_\phi B_\phi\|_{\mathrm{F}}.$$

All algebraic results in Section 4 and Theorem C.15 apply to the triple $(B_\phi, C_\phi, W_\phi)$ and the induced operator $A_\phi$. The cochain interpretation arises by taking

$$\begin{aligned} V = C^0, \qquad W &= C^1, \qquad Z = C^2, \\ B_\phi = D_{0,\phi}, \qquad C_\phi &= D_{1,\phi}, \qquad W_\phi = W_{1,\phi}, \\ A_\phi &= L_{0,\phi}. \end{aligned}$$

## E. Additional Experimental Details and Diagnostics

This appendix presents the numerical experiments in full detail. All implementation details, the specific PDE families used, and complete numerical tables for all meshes, regimes, and diagnostics are collected in Appendix L. All core structural experiments are carried out for a scalar elliptic model problem on a unit square with homogeneous

Dirichlet conditions and a smooth forcing term, so that the corresponding analytic solution is available. The focus is on how structural perturbations of incidence operators and their learned variants influence operator-level deviations, PDE error, and spectral Betti estimates.

### E.1. Model Problem and Implementation Details

The domain is $\Omega = [0, 1]^2$, discretised by a family of structured triangular meshes obtained by splitting each rectangle in a $N \times N$ Cartesian grid into two triangles. Unless stated otherwise, $N \in \{8, 16, 32\}$ is used; mesh-size dependence is studied explicitly in Section E.4. Cochain spaces $C^0(K_h)$ and $C^1(K_h)$ correspond to vertices and edges, and the reference incidence operators $D_0^{\mathrm{ref}}$, $D_1^{\mathrm{ref}}$ and weights $W_1^{\mathrm{ref}}$ define a DEC/FEEC Laplacian $L_{0,h}^{\mathrm{ref}}$ on 0-cochains.

Two concrete elliptic model problems are used throughout. Both are Dirichlet problems for $-\nabla \cdot (\kappa \nabla u) = f$ on $\Omega$, with the right-hand side $f$ chosen so that a smooth closed-form solution is available. PDE1 is the constant-coefficient baseline

$$\kappa_1(x, y) \equiv 1, \qquad u_1(x, y) = \sin(\pi x)\,\sin(\pi y),$$

corresponding to PDE1. PDE2 is a heterogeneous-coefficient variant

$$\kappa_2(x, y) = 1 + 0.5\,\sin(2\pi x)\,\sin(2\pi y),$$
$$u_2(x, y) = \sin(2\pi x)\,\sin(\pi y).$$

Unless otherwise stated, results are reported for both PDE1 and PDE2 on the same mesh family and with identical training hyperparameters, so differences isolate the effect of heterogeneous coefficients. In the generic loss notation below, the corresponding analytic solution is denoted by $u_{\mathrm{ana}}$.

In the perturbation tests, learned operators $(D_{0,\phi,h}, D_{1,\phi,h}, W_{1,\phi,h})$ are constructed by applying controlled random perturbations to $(D_0^{\mathrm{ref}}, D_1^{\mathrm{ref}}, W_1^{\mathrm{ref}})$ and then rescaling to maintain the uniform operator and SPD bounds of Section 3.2. Unless explicitly stated, the numerical run that produces the tables in Appendix L keeps $D_1$ and $W_1$ fixed and learns only $D_0$, using $D_1$ solely to measure $D_{\mathrm{comp}}(\phi) = \|D_1 D_0(\phi)\|_F$.

**Normalized defect (mesh-comparable).** Since $\| \cdot \|_F$ scales with problem size and discretization conventions, cross-resolution plots additionally report the dimensionless normalization

$$D_{\mathrm{comp}}^{\mathrm{norm}}(\phi) := \frac{\|D_1 D_0(\phi)\|_F}{\|D_1\|_F\,\|D_0(\phi)\|_F + 10^{-12}},$$

while the tables in Appendix L reproduce the raw $D_{\mathrm{comp}}$ values logged by the code for traceability.

In the learning-dynamics experiments, the parameters are updated by gradient descent on a PDE loss:

$$\mathcal{L}_{\mathrm{PDE}}(\phi) := \frac{1}{2}\,\|u_{\phi,h} - u_{\mathrm{ana}}\|_{L_h^2}^2,$$

where $u_{\phi,h}$ solves the discrete problem (6) on a fixed mesh. Three regimes are compared:

(a) *Unconstrained*: $D_{0,\phi,h}$ (and, when enabled, $D_{1,\phi,h}$) are unconstrained apart from structural bounds; the loss is purely $\mathcal{L}_{\mathrm{PDE}}$.

(b) *Soft-defect-regularised*: the loss is $\mathcal{L}_{\mathrm{PDE}} + \lambda_{\mathrm{comp}} D_{\mathrm{comp}}(\phi)^2$ with a fixed regularisation weight $\lambda_{\mathrm{comp}}$.

(c) *Chain-compatible manifold (hard constraint)*: $D_{1,\phi,h}$ is fixed at $D_1^{\mathrm{ref}}$ and $D_{0,\phi,h} = KR(\phi)$ is parameterised as in Section G.7, so $D_{\mathrm{comp}}(\phi) \equiv 0$ by construction. The loss is again $\mathcal{L}_{\mathrm{PDE}}$ (optionally with additional anchoring terms in some runs; see Appendix L).

All operator norms are estimated using power iterations, and $L^2$ errors are computed using the standard mass matrix on $C^0(K_h)$.

### E.2. Operator and PDE Sensitivity to Cochain-Compatibility Defect

The first group of experiments tests the Laplacian stability Theorem 4.4 and the PDE error Theorem 5.6 on a fixed mesh with $N = 32$. Starting from the reference operators, random perturbations are added and the cochain-compatibility defect $D_{\mathrm{comp}}(\phi) = \|D_{1,\phi,h} D_{0,\phi,h}\|_F$, the operator deviation $\left\|L_{0,\phi,h} - \widetilde{L}_{0,\phi,h}\right\|_{\mathrm{op}}$, and three solution errors are recorded: the total $L^2$ error with respect to the analytic solution $u_{\mathrm{ana}}$, the error between the learned solution $u_{\phi,h}$ and the solution $\widetilde{u}_{\phi,h}$ using the projected Laplacian $\widetilde{L}_{0,\phi,h}$, and the error between $\widetilde{u}_{\phi,h}$ and a reference FEEC solution $u_{\mathrm{ref},h}$ obtained from $L_{0,h}^{\mathrm{ref}}$.

Figure 5 plots the operator deviation $\left\|L_{0,\phi,h} - \widetilde{L}_{0,\phi,h}\right\|_{\mathrm{op}}$ against the cochain-compatibility defect. For small values of $D_{\mathrm{comp}}(\phi)$ the curve is approximately linear in log–log coordinates; as the defect grows, a mild upward curvature appears, consistent with the quadratic $BD_{\mathrm{comp}}(\phi)^2$ term in Theorem 4.4. In the regime where $D_{\mathrm{comp}}(\phi) \lesssim 1$, the observed deviation is bounded by a constant multiple of $AD_{\mathrm{comp}}(\phi) + BD_{\mathrm{comp}}(\phi)^2$, as predicted.

Figure 1a displays the three $L^2$ error quantities as functions of $D_{\mathrm{comp}}(\phi)$. Several features are notable.

- For very small defects the defect-only error $\|u_{\phi,h} - \widetilde{u}_{\phi,h}\|_{L_h^2}$ lies close to the theoretical bound

$AD_{\mathrm{comp}}(\phi) + BD_{\mathrm{comp}}(\phi)^2$, indicating that in this regime the defect-induced perturbation behaves as a controlled additive term in the PDE error.

- In the same regime the total error $\|u_{\phi,h} - u_{\mathrm{ana}}\|_{L^2_h}$ is nearly constant and essentially equal to the FEEC baseline with no structural defect. This indicates that discretization error dominates the total error, and that sufficiently small structural defects are negligible compared with standard FEEC approximation error.

- As the defect increases beyond the small-defect regime, the sufficient conditions underlying Theorem 5.6 no longer apply. The error curves can become non-monotone: the defect-only term grows, and eventually the structural perturbation dominates the discretization error, but the total error need not increase monotonically because the discretization is no longer close to any FEEC-like scheme.

Figure 3 illustrates a representative solution at a large defect and the corresponding error field. The observed mesh-aligned artefacts support the interpretation that for large structural defects the discretization can exit the range of validity of the theory. In the rest of this section, the discussion of quantitative agreement with the theory is restricted to the small-defect regime.

### E.3. Learning Dynamics with and Without Structural Control

The second group of experiments compares the three training regimes described in Section E.1 on a fixed mesh. For each regime the operator error $\left\| L_{0,\phi,h} - L_{0,h}^{\mathrm{ref}} \right\|_{\mathrm{op}}$, the cochain-compatibility defect $D_{\mathrm{comp}}(\phi)$, and the PDE loss $\mathcal{L}_{\mathrm{PDE}}(\phi)$ are recorded as functions of the gradient step.

Figures 6–7 show the three curves. Several conclusions can be drawn.

- Unconstrained training decreases the operator error and PDE loss rapidly, but the cochain-compatibility defect remains large and essentially unchanged. This indicates that PDE-loss optimization does not, by itself, enforce the cochain-complex condition, even when the target operator pair is chain-compatible.

- Adding a soft defect regulariser strongly reduces the cochain-compatibility defect and yields an operator whose defect is several orders of magnitude smaller than in the unconstrained case. The operator error and PDE loss remain competitive, but the final loss is slightly worse than the unconstrained baseline, reflecting the additional regularisation constraint.

- Training on the chain-compatible subspace keeps the defect at numerical zero throughout, as expected from the

analysis in Section G.7. The operator error and PDE loss converge slightly above the unconstrained baseline.

These experiments highlight a trade-off: enforcing the chain-compatibility constraint produces discretizations whose chain structure is controlled, at a modest but visible increase in PDE loss relative to an unconstrained model. The theory in Sections 5.2 and 4.4 explains how the defect and operator deviation terms enter the error bounds; the learning-dynamics experiments show that these quantities are not automatically small under unconstrained optimization and that defect regularisation or manifold constraints are effective mechanisms for controlling them.

### E.4. Mesh Convergence and Defect Scaling

The third experiment examines how the cochain-compatibility defect interacts with mesh refinement. A family of meshes with $N \in \{8, 16, 32, 64\}$, corresponding to mesh sizes $h \propto 1/N$, is considered, and three structural regimes are compared:

(i) *Baseline*: the reference DEC/FEEC operators with $D_{\mathrm{comp}}(\phi) \equiv 0$.

(ii) *Fixed structural defect*: a perturbation that introduces a defect of comparable magnitude on all meshes, so that $D_{\mathrm{comp}}(\phi, h)$ does not decay with $h$.

(iii) *Defect scaled with $h^2$*: perturbations constructed so that $\|\Delta D_0(h)\|_{\mathrm{F}} \sim h^2$, yielding a defect that decays with the same order as the standard $H^1$ FEEC error for the model problem.

For each regime the $L^2$ error $\|u - u_{\phi,h}\|_{L^2_h}$ is computed as a function of $h$.

Figure 9 plots the error versus mesh size on a log–log scale. The baseline and $h^2$-scaled-defect curves lie close and follow an $\mathcal{O}(h^2)$ slope, in line with standard FEEC theory and the defect-dependent error bound in Theorem 5.6. The fixed-defect curve has a higher error level but a similar slope across the mesh range tested. This is consistent with the interpretation that a defect decaying at least as quickly as the FEEC approximation error preserves the overall convergence order, while a fixed defect acts as an additive floor that would become dominant only at finer meshes than those shown.

### E.5. Spectral and Topological Robustness

The final group of experiments tests the spectral Betti stability Theorem 4.10. A mesh with a known number of connected components and reference Betti number $\beta_0$ is fixed, and the defect is varied by applying random perturbations

## Perturbed solution $u_\phi$ at $\sigma = 5.0e - 02$

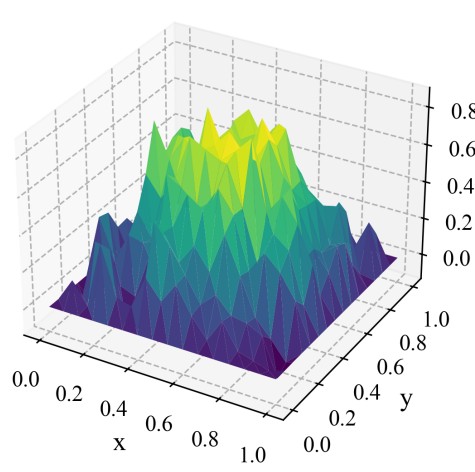

## Absolute error field at largest defect

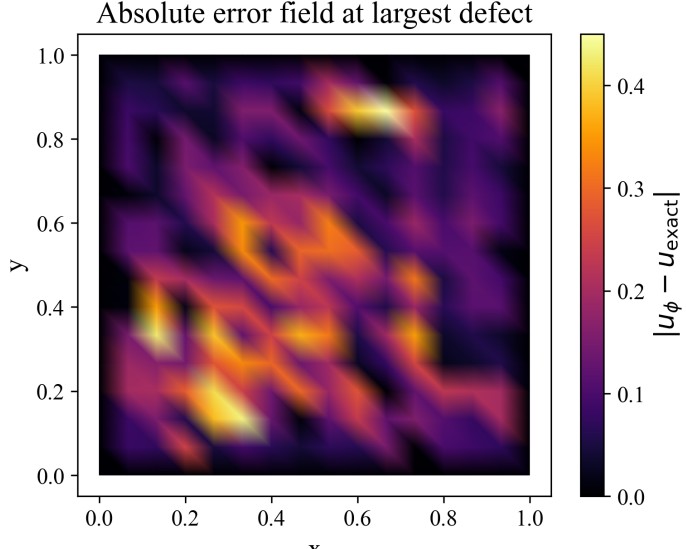

*Figure 3.* Left: perturbed solution $u_{\phi,h}$ at a representative large-defect point. Right: absolute error field $|u_{\phi,h} - u_{\text{ana}}|$. The error exhibits mesh-aligned high-frequency artefacts, illustrating that the large-defect regime lies outside the assumptions of the small-defect analysis.

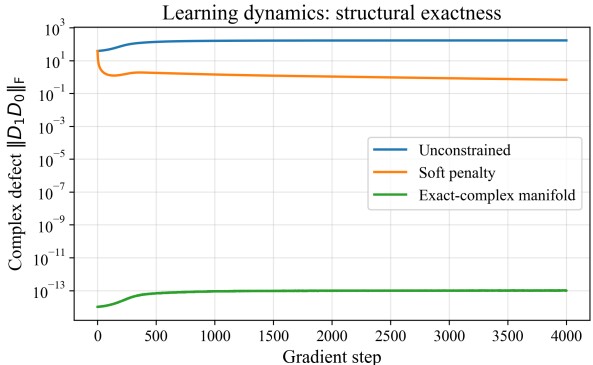

*Figure 4.* Evolution of the cochain-compatibility defect $D_{\text{comp}}(\phi)$ during training for the three regimes. Unconstrained training converges to a solution with a large structural defect, soft regularisation reduces the defect substantially but not to zero, and the manifold-constrained regime enforces $D_{\text{comp}}(\phi) \approx 0$ throughout.

as in Section E.2. For each defect magnitude multiple realisations of $(D_{0,\phi,h}, D_{1,\phi,h})$ are generated, the eigenvalues of $L_{0,\phi,h}$ are computed, and the spectral Betti estimate

$$\widehat{\beta}_0^{\text{spec}}(L_{0,\phi,h}; \lambda_*/2) = \#\{\, i : \lambda_i(\phi, h) < \lambda_*/2 \,\}$$

is evaluated, where $\lambda_*$ is the reference spectral gap for the projected Laplacian $\widetilde{L}_{0,\phi,h}$.

Figure 1b shows the resulting spectral Betti estimate as a function of $D_{\text{comp}}(\phi)$. The estimator matches the true Betti number across all realisations up to a defect range larger than the sufficient threshold $d_*$ from Corollary C.13, indicating that the theoretical bound is conservative but

still identifies a regime where the estimator is stable. As the defect increases further, the mean estimate decreases and the variance grows, reflecting the loss of a clean low-frequency spectral cluster.

Figure 8 plots the low-frequency spectrum as the defect varies. For defect values within the region where $AD_{\text{comp}}(\phi) + BD_{\text{comp}}(\phi)^2 < \lambda_*/2$, the group of eigenvalues forming the near-zero cluster remains separated from the rest of the spectrum and lies below $\lambda_*/2$, as required for spectral Betti stability. As the defect increases beyond this region, eigenvalues from the cluster drift upward and begin to collide with higher modes, at which point the spectral Betti estimator no longer coincides with the topological Betti number.

### E.6. Topology-Shift Failure Diagnostics

This subsection documents the failure-prediction analysis presented in Figure 2.

**Failure criteria.** For topology-shift evaluation, a test instance is marked as a *failure* if any of the following conditions occur:

(i) **Solver failure**: the SPD solve yields `NaN` or `inf` values, or the conjugate-gradient iteration fails to converge within $T_{\max} = 10{,}000$ iterations at a relative residual tolerance of $10^{-8}$;

(ii) **Ill-conditioning**: the condition number of the degree-0 discrete Hodge Laplacian $L_{0,\phi}$ (computed via power iteration and the inverse-iteration estimate of

the smallest nonzero eigenvalue) exceeds $10^5$;

(iii) **Betti-estimation failure**: the spectral Betti estimate $\widehat{\beta}_0^{\mathrm{spec}}(L_{0,\phi,h}; \lambda_*/2)$ computed in the unpinned Betti-estimation setting does not equal the ground-truth $\beta_0(K_h)$.

In practice, criterion (iii) is the most common trigger in the soft-defect and chain-compatible regimes, while criterion (i) or (ii) occasionally manifests in the unconstrained regime under extreme topological distribution shift.

**Estimating the spectral gap $\lambda_*$.** For each mesh family, $\lambda_*$ is estimated as follows. On the training distribution (meshes with $\beta_0 = 1$), the smallest positive eigenvalue of the projected Laplacian $\widetilde{L}_{0,\phi,h}$ (equivalently, the smallest positive eigenvalue of the reference Laplacian $L_{0,h}^{\mathrm{ref}}$ on the relative/unpinned operator) is computed across multiple training configurations, and $\lambda_*$ is taken to be the minimum of these values reduced by a safety margin of $10\%$. This ensures that Assumption 4.6 holds uniformly across the training set. For shifted test instances with $\beta_0 \in \{2, 3\}$, the same $\lambda_*$ value is used for consistency; the kernel-dimension increase under shift corresponds to the emergence of additional near-zero eigenvalues, and the gap above the cluster is expected to remain comparable under moderate topological changes that preserve local geometry.

**Budget ratio and binning.** Figure 2a bins test instances using the dimensionless ratio $\rho = \delta(\phi)/(\lambda_*/2)$, where $\delta(\phi) = A D_{\mathrm{comp}}(\phi) + B D_{\mathrm{comp}}(\phi)^2$. The constants $A$ and $B$ are computed from the uniform bounds defined in Theorem 4.4, using $\sigma_*$ (the minimum singular value of $D_1$ on its image) and the Hodge weight bounds $w_{\min}, w_{\max}$ as measured on the training set. Bin widths are logarithmically spaced over the range $[10^{-2}, 10^1]$ in $\rho$-space. For each bin, the failure rate is the fraction of pooled test instances (over all 5 seeds) whose $\rho$ value falls in the bin and that meet at least one failure criterion. Bins with fewer than 5 instances are not plotted.

**ROC analysis and per-seed variability.** Figure 2b shows receiver operating characteristic (ROC) curves treating failure prediction as binary classification. Each scalar score— $\rho = \delta(\phi)/(\lambda_*/2)$, raw defect $D_{\mathrm{comp}}(\phi)$, or validation PDE loss—is treated as a one-dimensional classifier. The ROC is computed by sweeping a threshold over the pooled unconstrained test set (aggregated across 5 seeds). Per-seed AUCs are computed independently for each seed and reported in Table 6.

The near-identical AUC for $\rho$ and $D_{\mathrm{comp}}$ arises because $\delta(\phi) = A D_{\mathrm{comp}}(\phi) + B D_{\mathrm{comp}}(\phi)^2$ is a monotone function of $D_{\mathrm{comp}}$ in the observed range, so their ranking of test instances is the same.

*Table 6.* **Per-seed AUC for failure prediction under topological distribution shift.** Mean $\pm$ standard deviation over 5 seeds. Validation PDE loss is near chance ($\approx 0.53$); budget ratio and defect are strong predictors.

| Predictor | AUC (mean $\pm$ std) |
|---|---|
| $\rho = \delta(\phi)/(\lambda_*/2)$ | $0.846 \pm 0.027$ |
| $D_{\mathrm{comp}}(\phi)$ | $0.846 \pm 0.027$ |
| Validation PDE loss | $0.531 \pm 0.041$ |

**Loss versus defect scatter.** Figure 2c plots validation PDE loss against $D_{\mathrm{comp}}$ for the unconstrained test pool. Points in the failure class (red) and non-failure class (blue) are overlaid. The scatter demonstrates that low PDE loss does not imply structural safety: numerous failures occur at low loss when defect is large, confirming that PDE-loss minimization alone does not prevent chain-condition violations that destabilise Betti estimates and solvers.

### E.7. Summary of Numerical Evidence

Table 30 summarises the core structural experiments and their relationship to the theoretical results.

Overall, the experiments support the view that the cochain-compatibility defect $D_{\mathrm{comp}}(\phi)$ behaves as a meaningful scalar order parameter for structural deviation from the chain-compatibility condition. In regimes where the defect is kept below a moderate threshold, operator deviations, PDE errors, and spectral Betti estimates behave in line with the theoretical bounds. When the defect becomes large, the discretization leaves the range of validity of the theory and the numerical behaviour indicates that PDE and topological information can become unreliable.

## F. Training Dynamics and Additional Diagnostics

This section gathers the learning curves and diagnostic figures referenced in the main text that have been moved here to improve the readability. The curves are presented in Figures 5-8.

## G. Additional Theoretical Extensions

### G.1. Degree-Wise Extension to Higher-Order Complexes

The analysis above has been presented for degree 0. This section records the direct extension to higher-degree factorised operators of the form

$$L_{k,\phi} := D_{k,\phi}^{\mathsf{T}} W_{k+1,\phi} D_{k,\phi}, \qquad k = 0, \ldots, K-1.$$

Let $(C^k)_{k=0}^K$ be finite-dimensional Hilbert spaces and, for each $\phi$, let $D_{k,\phi} : C^k \to C^{k+1}$ be linear maps.

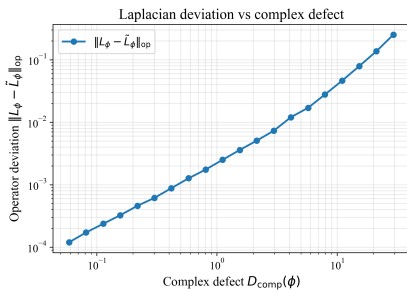

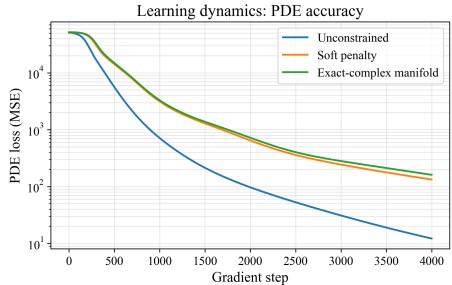

*Figure 5.* Laplacian operator deviation $\left\| L_{0,\phi,h} - \widetilde{L}_{0,\phi,h} \right\|_{\mathrm{op}}$ versus cochain-compatibility defect $D_{\mathrm{comp}}(\phi)$ on a log–log scale. Each point corresponds to a perturbation of $(D_0, D_1)$ on a fixed mesh. The relationship follows the predicted linear–quadratic regime.

*Figure 7.* PDE loss $\mathcal{L}_{\mathrm{PDE}}(\phi)$ versus optimization step. Soft regularisation and chain-compatible parameterisation converge to similar loss scales while maintaining improved structural consistency.

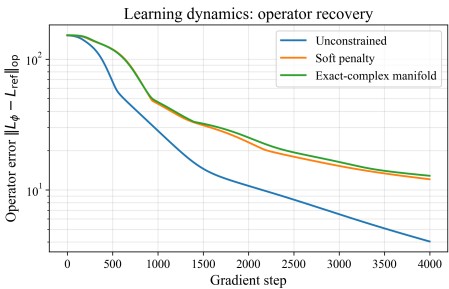

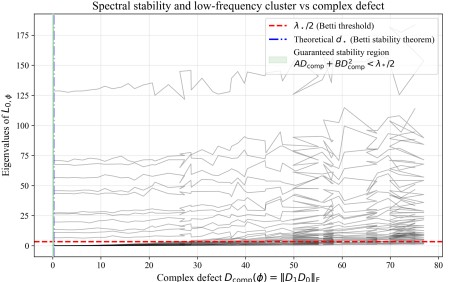

*Figure 6.* Learning dynamics of the operator error $\left\| L_{0,\phi,h} - L_{0,h}^{\mathrm{ref}} \right\|_{\mathrm{op}}$ for three regimes: unconstrained, soft defect regularisation, and chain-compatible parameterisation.

*Figure 8.* Spectral flow of eigenvalues of $L_{0,\phi,h}$ as the cochain-compatibility defect increases. The dashed line marks $\lambda_*/2$ and the shaded region indicates the sufficient stability condition $AD_{\mathrm{comp}}(\phi) + BD_{\mathrm{comp}}(\phi)^2 < \lambda_*/2$.

**Definition G.1** (Degree-wise cochain-compatibility defects). For $k = 0, \dots, K - 2$ define

$$\mathcal{C}_k(\phi) := D_{k+1,\phi} D_{k,\phi} : C^k \to C^{k+2},$$
$$D_{k,\mathrm{comp}}(\phi) := \left\| \mathcal{C}_k(\phi) \right\|_{\mathrm{F}}.$$

For each $k$ and $\phi$, define

$$\widetilde{D}_{k,\phi} := P_{\ker(D_{k+1,\phi})} D_{k,\phi} : C^k \to C^{k+1}.$$

**Assumption G.2** (Degree-wise structural bounds). For each $k = 0, \dots, K - 2$, assume:

- there exists $B_{D,k} > 0$ such that $\left\| D_{k,\phi} \right\|_{\mathrm{op}} \le B_{D,k}$ and $\left\| D_{k+1,\phi} \right\|_{\mathrm{op}} \le B_{D,k}$ for all $\phi$;

- there exists $\sigma_{*,k} > 0$ such that every nonzero singular value of $D_{k+1,\phi}$ is at least $\sigma_{*,k}$ for all $\phi$;

- the weight matrices $W_{k+1,\phi}$ are SPD with $w_{\min,k} I \preceq W_{k+1,\phi} \preceq w_{\max,k} I$ uniformly in $\phi$.

**Theorem G.3** (Degree-$k$ Laplacian stability). *For each* $k = 0, \dots, K - 2$ *define*

$$L_{k,\phi} := D_{k,\phi}^{\mathsf{T}} W_{k+1,\phi} D_{k,\phi},$$
$$\widetilde{L}_{k,\phi} := \widetilde{D}_{k,\phi}^{\mathsf{T}} W_{k+1,\phi} \widetilde{D}_{k,\phi}.$$

*Under the degree-$k$ structural bounds, there exist constants* $A_k, B_k > 0$ *independent of* $\phi$ *such that*

$$\left\| L_{k,\phi} - \widetilde{L}_{k,\phi} \right\|_{\mathrm{op}} \le A_k D_{k,\mathrm{comp}}(\phi) + B_k D_{k,\mathrm{comp}}(\phi)^2.$$

*Proof.* See Appendix C.

This appendix collects extensions that are useful for completeness and implementation, but are not required to follow the degree-zero development in the main text.

### G.2. Higher-Degree FEEC Complexes and Full Hodge Laplacians

The defect machinery is extended to the full discrete Hodge Laplacian on $k$-cochains in a FEEC/DEC setting. The numerical experiments in Section 6 focus on degree zero; the analysis below shows how analogous bounds arise at higher degrees.

Let $(C^k(K_h))_{k=0}^K$ be the FEEC cochain spaces on a fixed cell complex $K_h$, with incidence operators $D_k : C^k \to C^{k+1}$ and SPD Hodge weights $W_k$.

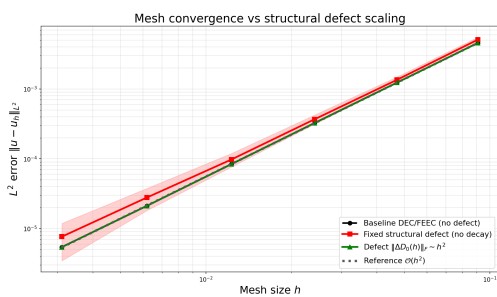

*Figure 9.* Mesh convergence of the $L^2$ error $\|u - u_{\phi,h}\|_{L^2_h}$ for different structural regimes. When the defect decays with $h$, the observed slope matches baseline FEEC convergence behaviour; a fixed defect increases the error level.

### G.3. Full Hodge Laplacian and Learned Analogue

The reference discrete Hodge Laplacian at degree $k$ is

$$\Delta_{k,h}^{\mathrm{ref}} := D_k^\mathsf{T} W_{k+1} D_k + W_k^{-1} D_{k-1} W_{k-1} D_{k-1}^\mathsf{T}, \quad (8)$$

with the convention that the "downward" term is absent for $k = 0$ and the "upward" term is absent for $k = K$.

On the learned side, degree-wise learned operators $(D_{k,\phi})_{k=0}^{K-1}$ and weights $(W_{k,\phi})_{k=0}^{K}$ satisfying uniform SPD and operator bounds are considered. The learned Hodge Laplacian is

$$\Delta_{k,\phi,h} := D_{k,\phi}^\mathsf{T} W_{k+1,\phi} D_{k,\phi} \\ + W_{k,\phi}^{-1} D_{k-1,\phi} W_{k-1,\phi} D_{k-1,\phi}^\mathsf{T}. \quad (9)$$

The "upward" part is of the factorised form treated earlier. The "downward" part can be reduced to a factorised form by conjugation with $W_{k,\phi}$ as follows.

### G.4. Degree-Wise Defects and Projected Complexes

For each $k = 0, \dots, K - 1$ and $\phi$, recall

$$D_{k,\mathrm{comp}}(\phi) = \|D_{k+1,\phi} D_{k,\phi}\|_\mathrm{F},$$

and define

$$\widetilde{D}_{k,\phi} := P_{\ker(D_{k+1,\phi})} D_{k,\phi}.$$

For $k \geq 1$ define $\widetilde{D}_{k-1,\phi} := P_{\ker(D_{k,\phi})} D_{k-1,\phi}$. The projected full Hodge Laplacian is

$$\widetilde{\Delta}_{k,\phi,h} := \widetilde{D}_{k,\phi}^\mathsf{T} W_{k+1,\phi} \widetilde{D}_{k,\phi} \\ + W_{k,\phi}^{-1} \widetilde{D}_{k-1,\phi} W_{k-1,\phi} \widetilde{D}_{k-1,\phi}^\mathsf{T}. \quad (10)$$

An operator-norm bound for $\Delta_{k,\phi,h} - \widetilde{\Delta}_{k,\phi,h}$ is now derived.

### G.5. Operator-Norm Stability for the Full Hodge Laplacian

For the upward part,

$$\left\| D_{k,\phi}^\mathsf{T} W_{k+1,\phi} D_{k,\phi} \\ - \widetilde{D}_{k,\phi}^\mathsf{T} W_{k+1,\phi} \widetilde{D}_{k,\phi} \right\|_\mathrm{op} \leq A_k D_{k,\mathrm{comp}}(\phi) \\ + B_k D_{k,\mathrm{comp}}(\phi)^2.$$

for some constants $A_k, B_k > 0$.

For the downward part define

$$B_{k-1,\phi} := W_{k-1,\phi}^{1/2} D_{k-1,\phi}^\mathsf{T} W_{k,\phi}^{-1/2},$$

so that

$$B_{k-1,\phi}^\mathsf{T} B_{k-1,\phi} = W_{k,\phi}^{-1/2} D_{k-1,\phi} W_{k-1,\phi} D_{k-1,\phi}^\mathsf{T} W_{k,\phi}^{-1/2}.$$

The projected analogue is $\widetilde{B}_{k-1,\phi} := W_{k-1,\phi}^{1/2} \widetilde{D}_{k-1,\phi}^\mathsf{T} W_{k,\phi}^{-1/2}$, with

$$\widetilde{B}_{k-1,\phi}^\mathsf{T} \widetilde{B}_{k-1,\phi} = W_{k,\phi}^{-1/2} \widetilde{D}_{k-1,\phi} W_{k-1,\phi} \widetilde{D}_{k-1,\phi}^\mathsf{T} W_{k,\phi}^{-1/2}.$$

Applying degree-wise Laplacian stability to the factorised pair $(B_{k-1,\phi}, D_{k,\phi})$ yields constants $\hat{A}_{k-1}, \hat{B}_{k-1} > 0$ such that

$$\left\| B_{k-1,\phi}^\mathsf{T} B_{k-1,\phi} \right\|_\mathrm{op} - \left\| \widetilde{B}_{k-1,\phi}^\mathsf{T} \widetilde{B}_{k-1,\phi} \right\|_\mathrm{op} \\ \leq \hat{A}_{k-1} D_{k-1,\mathrm{comp}}(\phi) \\ + \hat{B}_{k-1} D_{k-1,\mathrm{comp}}(\phi)^2.$$

where $D_{k-1,\mathrm{comp}}$ is understood here as a degree-$(k-1)$ defect controlling the chain-condition violation for the pair $(B_{k-1,\phi}, D_{k,\phi})$; any choice equivalent to $\|D_{k,\phi} B_{k-1,\phi}\|_\mathrm{F}$ up to constants depending only on the weight bounds is admissible.

Conjugating back with $W_{k,\phi}^{1/2}$ and using boundedness of $W_{k,\phi}^{\pm 1/2}$ yields a bound for the downward part in the original variables. Collecting these pieces gives:

**Proposition G.4** (Full Hodge Laplacian stability at degree $k$)**.** *Under the degree-wise structural bounds on $(D_{k-1,\phi}, D_{k,\phi}, D_{k+1,\phi})$ and the Hodge weights $(W_{k-1,\phi}, W_{k,\phi}, W_{k+1,\phi})$, there exist constants $C_k^{(1)}, C_k^{(2)} > 0$ independent of $\phi$ such that*

$$\left\| \Delta_{k,\phi,h} - \widetilde{\Delta}_{k,\phi,h} \right\|_\mathrm{op} \leq C_k^{(1)} D_{k-1,\mathrm{comp}}(\phi) \\ + C_k^{(1)} D_{k,\mathrm{comp}}(\phi) \\ + C_k^{(2)} D_{k-1,\mathrm{comp}}(\phi)^2 \\ + C_k^{(2)} D_{k,\mathrm{comp}}(\phi)^2.$$

where $D_{k,\mathrm{comp}}$ is interpreted degree-wise as a structural defect controlling the violation of $D_{k+1,\phi}D_{k,\phi} = 0$ for the upward part and of the corresponding weighted cochain-complex condition for the downward part.

*Proof.* See Appendix C. □

### G.6. Example: $k = 1$ and Spectral $\beta_1$ Stability

At $k = 1$ the reference Hodge Laplacian on edges is

$$\Delta_{1,h}^{\mathrm{ref}} = D_1^\mathsf{T} W_2 D_1 + W_1^{-1} D_0 W_0 D_0^\mathsf{T},$$

with $\dim \ker(\Delta_{1,h}^{\mathrm{ref}}) = \beta_1(K_h)$. On the learned side, $\Delta_{1,\phi,h}$ is built from $(D_{0,\phi}, D_{1,\phi}, D_{2,\phi})$ and $(W_{0,\phi}, W_{1,\phi}, W_{2,\phi})$, and the relevant defects are

$$D_{0,\mathrm{comp}}(\phi) = \|D_{1,\phi}D_{0,\phi}\|_\mathrm{F},$$
$$D_{1,\mathrm{comp}}(\phi) = \|D_{2,\phi}D_{1,\phi}\|_\mathrm{F}.$$

for the upward part, together with the analogous degree-0 defect for the downward term.

Assume a degree-1 analogue of Assumption 5.2, interpreted as an *analysis anchor*: the projected operators define a chain-compatible complex and the projected Hodge Laplacian $\widetilde{\Delta}_{1,\phi,h}$ has the same kernel dimension $\beta_1(K_h)$ and admits a uniform spectral gap. (The projection $P_{\ker(D_{k+1,\phi})}$ is generally global and may destroy sparsity/locality; $\widetilde{D}_{k,\phi}$ is therefore used here as a theoretical reference operator rather than asserted to be a valid coboundary in a fixed FEEC/DEC discretization class without additional locality constraints.)

**Assumption G.5** (Spectral gap for projected $\Delta_1$)**.** There exists $\lambda_1^* > 0$ such that for all $\phi$ in the parameter range of interest,

$$0 = \mu_1^{(1)}(\phi) = \cdots = \mu_{\beta_1(K_h)}^{(1)}(\phi),$$
$$\mu_{\beta_1(K_h)+1}^{(1)}(\phi) > 0, \qquad \mu_{\beta_1(K_h)+1}^{(1)}(\phi) \geq \lambda_1^*.$$

where $\mu_i^{(1)}(\phi)$ are the ordered eigenvalues of $\widetilde{\Delta}_{1,\phi,h}$.

Let $\lambda_i^{(1)}(\phi)$ denote the eigenvalues of $\Delta_{1,\phi,h}$ and define the spectral 1st Betti number at resolution $\tau > 0$ as

$$\widehat{\beta}_1^{\mathrm{spec}}(\Delta_{1,\phi,h}; \tau) := \#\{\, i : \lambda_i^{(1)}(\phi) < \tau \,\}.$$

**Theorem G.6** (Edge-level spectral Betti-number stability)**.** *Assume the degree-wise structural bounds, the degree-1 analogue of Assumption 5.2 for $(D_{0,\phi}, D_{1,\phi}, D_{2,\phi})$ and $(W_{0,\phi}, W_{1,\phi}, W_{2,\phi})$, and Assumption G.5. Let*

$$\Xi_1(\phi) := C_1^{(1)}\big(D_{0,\mathrm{comp}}(\phi) + D_{1,\mathrm{comp}}(\phi)\big)$$
$$+ C_1^{(2)}\big(D_{0,\mathrm{comp}}(\phi)^2 + D_{1,\mathrm{comp}}(\phi)^2\big).$$

*where $C_1^{(1)}, C_1^{(2)} > 0$ are the constants from Proposition G.4 with $k = 1$. If $\Xi_1(\phi) < \lambda_1^*/2$, then*

$$\widehat{\beta}_1^{\mathrm{spec}}(\Delta_{1,\phi,h}; \lambda_1^*/2) = \beta_1(K_h).$$

*Proof.* See Appendix C. □

### G.7. Manifold-Constrained Training on Chain-Compatible Complexes

The projection formula $\widetilde{D}_{0,\phi} = P_{\ker(D_{1,\phi})}D_{0,\phi}$ suggests a constructive way to parameterise the manifold of chain-compatible complexes: instead of regularising $D_{\mathrm{comp}}(\phi)$ towards zero, training can take place directly on the manifold where $D_{\mathrm{comp}}(\phi) \equiv 0$.

### G.8. Kernel-Based Parametrisation of Chain-Compatible Complexes

Fix an incidence operator $D_1^{\mathrm{ref}} : C^1 \to C^2$ with $\ker(D_1^{\mathrm{ref}})$ of dimension $r$. Let $K \in \mathbb{R}^{n_1 \times r}$ have orthonormal columns forming a basis of $\ker(D_1^{\mathrm{ref}})$, so that

$$P_{\ker(D_1^{\mathrm{ref}})} = KK^\mathsf{T}.$$

**Lemma G.7** (Parametrisation of chain-compatible complexes with fixed $D_1$)**.** *Let*

$$\mathcal{M}_{\mathrm{cc}} := \{\, B \in \mathbb{R}^{n_1 \times n_0} : D_1^{\mathrm{ref}} B = 0 \,\}$$

*be the manifold of coboundary operators with fixed higher coboundary $D_1^{\mathrm{ref}}$ that satisfy the cochain-complex condition. Then:*

(i) *For every $R \in \mathbb{R}^{r \times n_0}$, the matrix $B = KR$ belongs to $\mathcal{M}_{\mathrm{cc}}$.*

(ii) *Conversely, every $B \in \mathcal{M}_{\mathrm{cc}}$ can be written uniquely as $B = KR$ for some $R \in \mathbb{R}^{r \times n_0}$.*

*Proof.* See Appendix C. □

*Remark* G.8 (Dimension and gradients)*.* The manifold $\mathcal{M}_{\mathrm{cc}}$ has dimension $rn_0$, and the map $R \mapsto KR$ is an isometric embedding with respect to the Euclidean/Frobenius metric. Training in unconstrained coordinates $R(\theta)$ therefore corresponds to gradient-based optimization on $\mathcal{M}_{\mathrm{cc}}$ without requiring specialised optimization algorithms.

### G.9. Manifold-Constrained Training Regime

Suppose that in a learned-operator family

$$D_{1,\phi} \equiv D_1^{\mathrm{ref}}$$

holds for all parameters $\phi$, and

$$D_{0,\phi}(\theta) := KR(\theta)$$

is parameterised, with $\theta$ describing the entries of $R \in \mathbb{R}^{r \times n_0}$. Then the cochain-complex condition is enforced as a hard constraint:

**Proposition G.9** (Chain condition and vanishing defect under kernel parametrisation). *Under the manifold-constrained parametrisation $D_{0,\phi} = KR(\phi)$ and $D_{1,\phi} \equiv D_1^{\mathrm{ref}}$,*

(i) $D_{1,\phi} D_{0,\phi} = 0$ *for all $\phi$, so the learned complex is chain-compatible at degrees $0$ and $1$;*

(ii) *the cochain-compatibility defect vanishes identically, $D_{\mathrm{comp}}(\phi) \equiv 0$;*

(iii) *the projection satisfies $\widetilde{D}_{0,\phi} = P_{\ker(D_1^{\mathrm{ref}})} D_{0,\phi} = D_{0,\phi}$, so $D_{\mathrm{proj}}(\phi) \equiv 0$.*

*Proof.* See Appendix C. $\square$

Combining Proposition G.9 with the main stability results yields an immediate corollary.

**Corollary G.10** (PDE and topology under hard chain-compatibility constraint). *In the manifold-constrained regime of Proposition G.9, the Laplacian stability Theorem 4.4 and the PDE error Theorem 5.6 reduce to bounds driven solely by the operator-deviation term $\mathcal{R}_E(\phi)$. In particular,*

$$D_{\mathrm{comp}}(\phi) \equiv 0 \quad \Longrightarrow \quad \|u - u_{\phi,h}\|_V \lesssim \mathcal{E}_{\mathrm{FEEC}}(h) + \varepsilon_E,$$

*and the spectral Betti-number identities of Theorems 4.10 and G.6 hold automatically, without any defect threshold condition.*

*Remark* G.11 (Soft versus hard structural control). The cochain-compatibility defect $D_{\mathrm{comp}}(\phi)$ supports two complementary training regimes:

- *Soft structural control* via defect regularisation, where $D_{0,\phi}$ and $D_{1,\phi}$ are learned unconstrained and $D_{\mathrm{comp}}(\phi)$ is added to the loss. The theory above quantifies how keeping $D_{\mathrm{comp}}(\phi)$ below an explicit threshold yields PDE and topology guarantees; the learning-dynamics experiment in Section 6 illustrates that this leads to a substantial reduction in defect at a moderate cost in PDE loss.

- *Hard structural control* via kernel parametrisation, where $D_{1,\phi}$ is fixed and $D_{0,\phi}$ is parameterised as $KR(\phi)$, so that $D_{\mathrm{comp}}(\phi) \equiv 0$ by design and the analysis collapses to the operator-deviation term $\mathcal{R}_E(\phi)$. This provides a minimal Euclidean parametrisation of chain-compatible complexes that can be combined with standard optimization tools.

# H. Computational Cost: Defect and Projection

This section details the computational procedures for evaluating the cochain-compatibility defect and applying the projection $P_{\ker(D_1)}$ without forming a pseudoinverse, addressing feasibility concerns for the analysis operators used in Sections 4–5.

**Defect estimation via Hutchinson.** The cochain-compatibility defect $D_{\mathrm{comp}}(\phi) = \|D_{1,\phi} D_{0,\phi}\|_F$ is never computed by forming the product $D_{1,\phi} D_{0,\phi}$ as a dense matrix. Using the Hutchinson trace estimator identity $\|A\|_F^2 = \mathrm{tr}(A^\top A) = \mathbb{E}_z \|Az\|_2^2$ for Rademacher vectors $z \in \{\pm 1\}^{n_0}$, we have

$$\|D_{1,\phi} D_{0,\phi}\|_F^2 = \mathrm{tr}\big((D_{1,\phi} D_{0,\phi})^\top (D_{1,\phi} D_{0,\phi})\big)$$
$$= \mathbb{E}_z \|D_{1,\phi} D_{0,\phi} z\|_2^2.$$

In practice, $s$ independent Rademacher vectors $\{z_j\}_{j=1}^s$ are drawn and the estimate

$$\|D_{1,\phi} D_{0,\phi}\|_F^2 \approx \frac{1}{s} \sum_{j=1}^s \|D_{1,\phi} D_{0,\phi} z_j\|_2^2$$

is used, where each evaluation requires two sparse matvecs: $v_j = D_{0,\phi} z_j$ followed by $w_j = D_{1,\phi} v_j$. The probe count $s$ is chosen so that the relative standard error of the trace estimate is below a fixed tolerance specified in the experimental settings, ensuring defect curves are not dominated by estimator noise. Computational cost is $O(s(\mathrm{nnz}(D_{0,\phi}) + \mathrm{nnz}(D_{1,\phi})))$, comparable to evaluating the Laplacian action $L_{0,\phi} = D_{0,\phi}^\top W_{1,\phi} D_{0,\phi}$ on $s$ vectors. Gradients with respect to $\phi$ are obtained via standard automatic differentiation through the matvec pipeline.

**Applying $P_{\ker(D_{1,\phi})}$ without forming $D_{1,\phi}^\dagger$.** The analysis uses the Euclidean orthogonal projector

$$P_{\ker(D_{1,\phi})} = I - D_{1,\phi}^\dagger D_{1,\phi}.$$

Given $v \in C^1$, set $b := D_{1,\phi} v \in C^2$. The vector $x := D_{1,\phi}^\dagger b$ is the minimum-norm solution of $D_{1,\phi} x = b$. Computationally, one recovers the minimum-norm solution up to solver tolerance by solving the normal equation on the range:

$$D_{1,\phi} D_{1,\phi}^\top y = b \quad \text{(with } b \in \mathrm{range}(D_{1,\phi}) \text{ automatically),}$$
$$x = D_{1,\phi}^\top y.$$

using MINRES/LSQR (or CG on the SPD restriction to $\mathrm{range}(D_{1,\phi})$). When $D_1 D_1^\top$ is singular, the solve is carried out in $\mathrm{range}(D_1)$ (e.g. MINRES with nullspace deflation, or CG on the SPD restriction after projecting the iterate orthogonally to $\ker(D_1 D_1^\top)$). In all cases $b = D_1 v$ lies in $\mathrm{range}(D_1)$, so the minimum-norm solution is well-defined. Then $P_{\ker(D_{1,\phi})} v = v - x$.

**Lemma H.1** (Projector application via normal equations). *If $b \in \mathrm{range}(A)$, then the solution $x^* = A^\top y$, where $AA^\top y = b$, satisfies $x^* = A^\dagger b$.*

*Proof.* Let $A = U\Sigma V^\top$. $A^\dagger b = V\Sigma^{-1}U^\top b$. $AA^\top = U\Sigma^2 U^\top$. $AA^\top y = b \implies y = U\Sigma^{-2}U^\top b + k$, $k \in \mathrm{ker}(AA^\top)$. $A^\top y = V\Sigma U^\top (U\Sigma^{-2}U^\top b) = V\Sigma^{-1}U^\top b = A^\dagger b$. $\square$

For numerical robustness, the Tikhonov-regularized map

$$P_\eta := I - D_{1,\phi}^\top (D_{1,\phi}D_{1,\phi}^\top + \eta I)^{-1}D_{1,\phi}, \qquad \eta > 0,$$

may be used, which is *not* the orthogonal projector but converges to $P_{\mathrm{ker}(D_{1,\phi})}$ as $\eta \downarrow 0$. Lemma C.9 quantifies the induced operator error; using $P_\eta$ makes the certificate conservative if the additional slack is accounted for.

**Pre-factorisation when $D_{1,\phi}$ is fixed.** In the experiments of Section 6, $D_{1,\phi} \equiv D_1^{\mathrm{ref}}$ is fixed per mesh and only $D_{0,\phi}$ is learned. In this regime, the matrix $A := D_1 D_1^\top + \eta I \in \mathbb{R}^{n_2 \times n_2}$ is constant and can be pre-factorised once using sparse Cholesky or LU (when $A$ is SPD, Cholesky is preferred). Each subsequent projection application then reduces to two sparse triangular solves (forward and backward substitution) plus one matvec with $D_1^\top$, making the projection cost negligible compared to a single Laplacian matvec. Factorisation is performed offline at mesh preprocessing; online cost per projection is $O(\mathrm{nnz}(L) + n_2)$ where $L$ is the Cholesky factor.

**Matvec with the projected Laplacian $\widetilde{L}_{0,\phi}$.** The projected Laplacian $\widetilde{L}_{0,\phi} = \widetilde{D}_{0,\phi}^\top W_{1,\phi} \widetilde{D}_{0,\phi}$ with $\widetilde{D}_{0,\phi} = P_{\mathrm{ker}(D_{1,\phi})}D_{0,\phi}$ is used only for analysis and diagnostics (operator-norm estimation via power iteration, spectral bounds). Since $P_{\mathrm{ker}(D_1)}$ is an orthogonal projector, $P_{\mathrm{ker}(D_1)}^\top = P_{\mathrm{ker}(D_1)}$, and

$$\begin{aligned}\widetilde{L}_{0,\phi} &= (PD_{0,\phi})^\top W_{1,\phi}(PD_{0,\phi}) \\ &= D_{0,\phi}^\top P W_{1,\phi} P D_{0,\phi}, \\ P &:= P_{\mathrm{ker}(D_{1,\phi})}.\end{aligned}$$

For any $x \in C^0$, compute $\widetilde{L}_{0,\phi}x$ via:

(i) $a = D_{0,\phi}x$ (matvec with $D_{0,\phi}$)

(ii) $b = Pa$ (projection solve)

(iii) $c = W_{1,\phi}b$ (diagonal scaling)

(iv) $d = Pc$ (projection solve)

(v) return $D_{0,\phi}^\top d$ (matvec with $D_{0,\phi}^\top$)

This is the correct algebra for $D_{0,\phi}^\top PW_{1,\phi}PD_{0,\phi}x$. Each matvec with $\widetilde{L}_{0,\phi}$ costs two matvecs with $D_{0,\phi}$ and two projection applications. When $D_1$ is fixed and pre-factorised, the total cost is dominated by $O(\mathrm{nnz}(D_0))$ sparse operations plus $O(n_2)$ triangular solves.

**Summary of computational claims.**

- The defect $D_{\mathrm{comp}}(\phi)$ and its gradient are computed via Hutchinson probes with $s \ll n_0$ samples; cost $O(s\,\mathrm{nnz}(D_0, D_1))$.

- The projection $P_{\mathrm{ker}(D_1)}$ is applied via sparse linear solves, not pseudoinverse formation; when $D_1$ is fixed, pre-factorisation makes each application $O(\mathrm{nnz}(L) + n_1)$.

- Matvecs with $\widetilde{L}_{0,\phi}$ (used for power iteration / operator-norm estimates) cost $O(\mathrm{nnz}(D_0))$ plus projection overhead.

- All procedures scale to moderately large meshes ($n_0 \sim 10^4$–$10^5$ vertices) without forming dense matrices or computing full spectral decompositions.

# I. Mesh-Scaling Audit of Certificate Constants

This appendix quantifies how the stability-budget constants in Theorem 4.4 behave under mesh refinement for the discretisation and normalisation used in the experiments. Recall

$$A = \frac{2w_{\max}B_D}{\sigma_*}, \qquad B = \frac{w_{\max}}{\sigma_*^2},$$

where $B_D$ upper bounds $\|D_{0,\phi}\|_{\mathrm{op}}, \|D_{1,\phi}\|_{\mathrm{op}}$ (Assumption 3.11), $w_{\max}$ upper bounds $\|W_{1,\phi}\|_{\mathrm{op}}$ (Assumption 3.9), and $\sigma_*$ lower bounds $\sigma_{\min}^+(D_{1,\phi})$ (Assumption 3.12).

**What is measured.** For each mesh $K_h$ used in Section 6 and each training regime, we report:

$$\begin{aligned}B_D(h) &:= \max\{\|D_{0,\phi,h}\|_{\mathrm{op}}, \|D_{1,\phi,h}\|_{\mathrm{op}}\}, \\ \sigma_*(h) &:= \inf_\phi \sigma_{\min}^+(D_{1,\phi,h}), \\ w_{\max}(h) &:= \sup_\phi \|W_{1,\phi,h}\|_{\mathrm{op}}.\end{aligned}$$

In the fixed-$D_1$ regimes, $\sigma_*(h) = \sigma_{\min}^+(D_{1,h}^{\mathrm{ref}})$ is mesh-dependent but independent of $\phi$.

**How the norms are estimated.** Operator norms are estimated by power iteration on $A^\top A$ with matrix-free products. For $\sigma_{\min}^+(D_{1,h})$, the smallest positive eigenvalue of $D_{1,h}D_{1,h}^\top$ restricted to $\mathrm{range}(D_{1,h})$ is estimated by Lanczos on the SPD restriction; equivalently, $\|(D_{1,h}^\dagger)\|_{\mathrm{op}}$ is estimated via inverse iteration on $D_{1,h}D_{1,h}^\top$ with a projection

to $\mathrm{range}(D_{1,h})$. All estimates use a tolerance matching the linear-solve tolerance in the projection implementation (Appendix H).

**Why this matters.** If $\sigma_*(h)$ decays with refinement under the chosen scaling, then $A(h), B(h)$ grow and the certificate becomes increasingly conservative. Conversely, if $\sigma_*(h)$ remains bounded away from zero (as in fixed-incidence settings under standard normalisations), then the budget constants remain controlled and the observed slack should remain stable across $h$.

**Budget margin diagnostic.** Define the observed drift

$$\Delta_L(\phi, h) := \|L_{0,\phi,h} - \widetilde{L}_{0,\phi,h}\|_{\mathrm{op}}$$

and the predicted budget

$$\delta(\phi, h) := A(h)\, D_{\mathrm{comp}}(\phi, h) + B(h)\, D_{\mathrm{comp}}(\phi, h)^2.$$

The margin ratio $\mathrm{margin}(\phi, h) := \Delta_L(\phi, h)/\delta(\phi, h)$ is reported across seeds and regimes to verify that the linear–quadratic form remains predictive under refinement.

## J. When $D_{1,\phi}$ Is Learned: Enforcing and Monitoring $\sigma_{\min}^+(D_{1,\phi})$

All constants in Theorem 4.2 and Theorem 4.4 depend on $\sigma_* \leq \sigma_{\min}^+(D_{1,\phi})$ (Assumption 3.12). If $\sigma_{\min}^+(D_{1,\phi})$ approaches 0, the certificate becomes conservative and can become vacuous.

**Monitoring.** The quantity $\sigma_{\min}^+(D_{1,\phi})$ is monitored during training by estimating the smallest positive eigenvalue of $D_{1,\phi}D_{1,\phi}^\top$ on $\mathrm{range}(D_{1,\phi})$ using a Lanczos run on the SPD restriction, consistent with the linear-solve tolerance used for projection (Appendix H). A rising $\|D_{1,\phi}^\dagger\|_{\mathrm{op}}$ is treated as an instability signal independent of $D_{\mathrm{comp}}(\phi)$.

**Enforcement mechanisms.** When $D_{1,\phi}$ is trainable, one of the following mechanisms is used:

- *Spectral floor penalty.* Add a penalty term

$$\mathcal{L}_\sigma(\phi) := \lambda_\sigma \left( \max\{0, \sigma_* - \widehat{\sigma}_{\min}^+(D_{1,\phi})\} \right)^2$$

  where $\widehat{\sigma}_{\min}^+$ is the monitored estimate.

- *Regularised projector.* Use the Tikhonov projector $P_\eta$ in Appendix H; Lemma C.9 bounds the induced conservatism. This prevents numerical instability in the projection step when $D_{1,\phi}$ becomes ill-conditioned.

- *Fixed-rank / reparameterised $D_1$.* Parameterise $D_{1,\phi} = U_\phi S_\phi V_\phi^\top$ with $S_\phi$ constrained to have singular values in $[\sigma_*, \sigma_{\max}]$ on the intended active subspace, so that $\sigma_{\min}^+(D_{1,\phi}) \geq \sigma_*$ by construction.

**Scope.** In the main experiments of Section 6, $D_{1,\phi} \equiv D_1^{\mathrm{ref}}$ is fixed per mesh, so $\sigma_*$ is a mesh constant and the learned-$D_1$ enforcement is not invoked.

## K. Baseline Objectives

This appendix records the full mathematical definitions for the three baselines that do not enforce the complex condition (B1–B3) introduced in Section 6.1. These baselines do not include $\|D_{1,\phi}D_{0,\phi}\|$ in the objective and do not use $P_{\mathrm{ker}(D_1)}$.

### K.1. SPD Conditioning Regularizer (Baseline B1)

Let $L_{0,\phi}^{\mathrm{SPD}}$ denote the Dirichlet-restricted (SPD) solve operator obtained after boundary elimination. Minimise

$$\mathcal{L}_{\mathrm{PDE}}(\phi) + \lambda_{\mathrm{cond}}\, \mathcal{R}_{\mathrm{cond}}(\phi),$$
$$\mathcal{R}_{\mathrm{cond}}(\phi) := \mathrm{tr}\left( \left(L_{0,\phi}^{\mathrm{SPD}} + \gamma I\right)^{-1} \right).$$

with $\gamma = 10^{-6}$, estimated by Hutchinson probes and conjugate-gradient solves (Appendix H). This baseline targets conditioning instability directly without referencing $D_1$, the cochain-complex condition, or topology.

### K.2. Laplacian Anchoring Only (Baseline B2)

Minimise

$$\mathcal{L}_{\mathrm{PDE}}(\phi) + \lambda_L \|L_{0,\phi} - L_{0,h}^{\mathrm{ref}}\|_{\mathrm{op}}^2,$$

with operator norm estimated via power iteration. Unlike $\mathcal{R}_E(\phi)$, this baseline uses the unprojected Laplacian directly and does not involve $P_{\mathrm{ker}(D_1)}$ or the projected reference operator $\widetilde{L}_{0,\phi}$.

### K.3. Low-Frequency Spectral Penalty (Baseline B3)

Let $U_k = [u_1^{\mathrm{ref}}, \ldots, u_k^{\mathrm{ref}}] \in \mathbb{R}^{n_0 \times k}$ contain the $k$ lowest eigenvectors of the reference unpinned Laplacian $L_{0,h}^{\mathrm{top,ref}}$ used for Betti estimation (Section 4.4). Minimise

$$\mathcal{L}_{\mathrm{PDE}}(\phi) + \lambda_{\mathrm{spec}} \|(L_{0,\phi}^{\mathrm{top}} - L_{0,\mathrm{ref}}^{\mathrm{top}})U_k\|_F^2,$$

which penalizes distortion of the low-frequency subspace used by spectral Betti estimation, without using $D_1$ or the chain-defect. The penalty is evaluated via $k$ sparse matvecs with $L_{0,\phi}$.

## L. Full Numerical Setup and Tables

This appendix records the precise numerical setup and the full tables of diagnostics for the experiments in Appendix E.

## L.1. Meshes, PDEs, and Discrete Objects

**Meshes.** Structured triangular meshes on $[0,1]^2$ obtained by splitting each cell of an $N \times N$ Cartesian grid into two triangles are used. The values

$$N \in \{8, 16, 32\}$$

are considered.

**PDE families.** Two scalar elliptic problems are considered as described in Section E.1:

- PDE1 (constant conductivity):

$$\kappa_1(x,y) = 1, \qquad u_1(x,y) = \sin(\pi x)\,\sin(\pi y).$$

- PDE2 (heterogeneous conductivity):

$$\kappa_2(x,y) = 1 + 0.5\,\sin(2\pi x)\,\sin(2\pi y),$$
$$u_2(x,y) = \sin(2\pi x)\,\sin(\pi y).$$

The right-hand sides are chosen so that the analytic solutions $u_i$ solve $-\nabla \cdot (\kappa_i \nabla u_i) = f_i$ with homogeneous Dirichlet conditions, and $L^2$-errors are measured against $u_i$.

**Discrete DEC/FEEC objects.** On each mesh $K_h$:

- $D_0 \colon C^0 \to C^1$ and $D_1 \colon C^1 \to C^2$ are incidence matrices, so that $D_1 D_0 = 0$ by construction.

- Lumped vertex areas and edge/face Hodges $W_0$, $W_1$, $W_2$ are constructed from the chosen $\kappa(x,y)$.

- The reference degree-0 discrete Hodge Laplacian is

$$L_{0,\mathrm{ref}} = D_0^\top W_1 D_0,$$

restricted to interior vertices by eliminating Dirichlet boundary nodes. The corresponding reference 1-cochain Hodge Laplacian is

$$L_{1,\mathrm{ref}} = D_1^\top W_2 D_1 + W_1^{-1} D_0 W_0 D_0^\top,$$

as standard for DEC.

## L.2. Training Regimes and Projection Baseline

In the run summarised in the tables below, the learned 0-cochain operator has the form

$$L_0(\phi) = D_0(\phi)^\top W_1 D_0(\phi),$$

with $D_1$ and $W_1$ fixed by the mesh and PDE, and parameters updated by gradient descent on variants of the PDE loss

$$\mathcal{L}_{\mathrm{PDE}}(\phi) := \tfrac{1}{2}\big\| u_{\phi,h} - u_{\mathrm{ana}} \big\|_{L^2(\Omega)}^2,$$

where $u_{\phi,h}$ solves the learned-discrete problem on $K_h$ and $u_{\mathrm{ana}}$ denotes the analytic solution for the selected PDE family. The cochain-compatibility defect is always measured as $D_{\mathrm{comp}}(\phi) = \|D_1 D_0(\phi)\|_F$ using the fixed $D_1$.

Three training regimes plus a projection baseline are instantiated.

**Unconstrained.** Here

$$D_0(\phi) = D_{0,\mathrm{base}} + UV^\top,$$

where $D_{0,\mathrm{base}}$ is the reference incidence restricted to interior vertices and $U, V$ collect trainable low-rank perturbations. The loss is purely PDE-based:

$$\mathrm{loss} = \mathcal{L}_{\mathrm{PDE}},$$

This regime trains purely on PDE loss.

**Soft defect penalty.** The parametrisation is the same, but the chain-condition violation is penalised:

$$D_{\mathrm{comp}}(D_0) := \|D_1 D_0\|_F,$$

and the loss is

$$\mathrm{loss} = \mathcal{L}_{\mathrm{PDE}} + \lambda_{\mathrm{defect}} \big\| D_1 D_0(\phi) \big\|_F^2,$$

with soft defect regularization.

**Chain-compatible subspace (hard constraint + anchoring).** In this regime $D_1$ is fixed at its reference value and $D_0(\phi)$ is parameterised on the chain-compatible subspace as in Section G.7: Let $K \in \mathbb{R}^{n_1 \times r}$ have orthonormal columns spanning $\ker(D_1)$. Define $R_{\mathrm{ref}} := K^\top D_{0,\mathrm{base}}$ and parameterise

$$D_0(\phi) := K\,(R_{\mathrm{ref}} + \Delta R(\phi)),$$

so that $D_1 D_0(\phi) \equiv 0$ for all $\phi$. The loss combines PDE accuracy, an operator-deviation penalty, and a small quadratic anchor on $\Delta R$:

$$\mathrm{loss} = w_{\mathrm{pde}} \cdot \mathcal{L}_{\mathrm{PDE}} + \lambda_E \big\| L_0(\phi) - L_{0,\mathrm{ref}} \big\|_F^2 + \lambda_{\Delta R} \|\Delta R\|_F^2.$$

with appropriate balancing of PDE loss and manifold regularization terms.

**Projection of the unconstrained solution.** For each unconstrained run a post-hoc projected incidence

$$\tilde{D}_0 = P_{\ker D_1}\, D_0^{(\mathrm{uncon})}$$

is formed, with corresponding Laplacian $\tilde{L}_0 = \tilde{D}_0^\top W_1 \tilde{D}_0$, and the PDE is solved once with this projected operator. No training is performed in this regime; it serves as a baseline for the effect of projection alone.

### L.3. Diagnostics

For any learned incidence operator $D_0$ the cochain-compatibility defect is

$$D_{\text{comp}}(D_0) := \|D_1 D_0\|_F,$$

Final cochain-compatibility defect is recorded. PDE errors are measured via train and test $L^2$-MSE against the reference discrete solution $u_{\text{ref},h}$ on held-out forcing instances (same mesh/PDE family unless stated).

For the degree-0 discrete Hodge Laplacian the following quantities are recorded:

$$\|L_0\|_{\text{op}}, \quad \|L_0 - L_{0,\text{ref}}\|_{\text{op}}, \quad \|L_0 - L_{0,\text{ref}}\|_F,$$
$$\lambda_{\min}(L_0), \quad \lambda_{\max}(L_0),$$
$$\text{cond}(L_0) := \lambda_{\max}(L_0)/\lambda_{\min}(L_0),$$

together with $\|L_1 - L_{1,\text{ref}}\|_{\text{op}}$ for the 1-cochain Hodge Laplacian.

Whenever $D_{\text{comp}}$ is not too small, an effective constant for the linear part of the defect law is also estimated,

$$A_{\text{eff}} := \frac{\|L_0 - \tilde{L}_0\|_{\text{op}}}{D_{\text{comp}}(D_0)},$$

where $\tilde{L}_0$ is the Laplacian of the projected complex $P_{\ker D_1} D_0$. This quantity is reported for analysis.

### L.4. Complex Defect and PDE Errors

Table 7 summarises the cochain-compatibility defect and train/test PDE mean-squared error across meshes, PDEs and regimes for the run.

### L.5. $L_0$ Diagnostics

Table 8 collects the main diagnostics for the degree-0 discrete Hodge Laplacian: operator norm, deviations from the reference operator, and condition numbers.

### L.6. $L_1$ Diagnostics and Effective Constants (Summary)

For completeness the operator-norm deviation of the 1-cochain Hodge Laplacian and the effective constant $A_{\text{eff}}$ defined above are briefly summarised. Full values are omitted for brevity; the reported ranges are sufficient for the claims referenced in the main text.

On the $8 \times 8$ and $16 \times 16$ meshes, the unconstrained regime yields $\|L_1 - L_{1,\text{ref}}\|_{\text{op}}$ in the range $4 \times 10^{-3}$–$2 \times 10^{-2}$ across PDE1 and PDE2, whereas the soft-penalty runs reduce this to $\sim 10^{-4}$ and the manifold regime to $O(10^{-8})$. The effective constant $A_{\text{eff}}$ lies around $10^{-2}$ in these moderate-defect settings, consistent with the linear part of the defect law.

On the $32 \times 32$ mesh, unconstrained training produces very large deviations, with $\|L_1 - L_{1,\text{ref}}\|_{\text{op}}$ of order $10^1$, while both regularised regimes remain essentially at machine precision ($\sim 10^{-18}$ for the soft penalty and $\sim 10^{-8}$ for the manifold).

These tables and ranges are the source of the qualitative statements in Section 6 about the relative behaviour of the three training regimes, the scaling of errors with the cochain-compatibility defect, and the onset of the large-defect regime where the theoretical assumptions break down.

## M. Extended Experimental Evaluation

This appendix collects additional hybrid operator-learning and PINN benchmarks, scaling and ablation studies, and explicit topological tests that complement the core structural experiments of Appendix E.

### M.1. Out-of-Distribution (OOD) Discretisation Shift Evaluation on Unstructured Meshes

This experiment evaluates robustness to *discretisation shift* on unstructured meshes. Models are trained on a train-like mesh distribution and evaluated on: (i) held-out in-distribution meshes (ID); (ii) OOD-1, a finer-mesh regime; and (iii) OOD-2, a pathological-mesh regime that stresses mesh quality and element irregularity. The primary targets include a scalar field (pressure/potential) and a 1-form/flux quantity, reported as relative $L^2$ errors. In each setting, the best baseline among methods {Unconstrained, Soft, Manifold} is reported as "Best baseline". Structure diagnostics include the (optionally normalised) cochain-compatibility defect (Remark E.1) and an anchoring consistency residual.

### M.2. Jointly Learned Operator Adaptation under OOD Mesh Shift

This study isolates the regime in which learned operator adaptation is genuinely required: both $D_0$ and $D_1$ are learned jointly under mesh shift. The comparison is therefore distinct from the fixed-$D_1$ setting and from post-hoc projection baselines. Table 10 shows that the three methods are nearly matched on ID error, so the separation is not explained by better in-distribution fitting. The difference appears under shift: the defect-aware model attains the best OOD nRMSE, the smallest raw compatibility defect, the strongest observed nondegeneracy of $D_1$, the smallest realized drift relative to the certified budget, the lowest failure rates across the certificate threshold, and the highest OOD spectral Betti accuracy. This is the intended use case for the certificate: performance is similar in-distribution and separates only when the learned operator must extrapolate structurally.

*Table 7.* Complex defect $D_{\mathrm{comp}} = \|D_1 D_0\|_F$ and PDE train/test MSE per mesh, PDE, and regime. Proj(Unconstr.) denotes post-hoc projection of the unconstrained $D_0$ onto $\ker(D_1)$ without retraining.

| Mesh | PDE | Regime | $D_{\mathrm{comp}}$ | Train MSE | Test MSE |
|---|---|---|---|---|---|
| $8 \times 8$ | PDE1 | Unconstrained | $1.70 \times 10^{-2}$ | $5.52 \times 10^{-10}$ | $8.59 \times 10^{-8}$ |
| | | Soft Penalty | $7.88 \times 10^{-7}$ | $2.79 \times 10^{-9}$ | $2.41 \times 10^{-9}$ |
| | | Manifold | $2.36 \times 10^{-14}$ | $1.51 \times 10^{-14}$ | $2.10 \times 10^{-14}$ |
| | | Proj(Unconstr.) | $2.36 \times 10^{-14}$ | — | $1.78 \times 10^{-7}$ |
| | PDE2 | Unconstrained | $2.47 \times 10^{-2}$ | $2.96 \times 10^{-10}$ | $1.86 \times 10^{-7}$ |
| | | Soft Penalty | $7.88 \times 10^{-7}$ | $3.13 \times 10^{-9}$ | $2.92 \times 10^{-9}$ |
| | | Manifold | $2.36 \times 10^{-14}$ | $1.55 \times 10^{-14}$ | $2.18 \times 10^{-14}$ |
| | | Proj(Unconstr.) | $2.35 \times 10^{-14}$ | — | $3.92 \times 10^{-7}$ |
| $16 \times 16$ | PDE1 | Unconstrained | $1.61 \times 10^{-1}$ | $9.71 \times 10^{-7}$ | $5.13 \times 10^{-4}$ |
| | | Soft Penalty | $1.48 \times 10^{-6}$ | $3.67 \times 10^{-10}$ | $4.80 \times 10^{-10}$ |
| | | Manifold | $7.43 \times 10^{-14}$ | $1.18 \times 10^{-12}$ | $1.41 \times 10^{-12}$ |
| | | Proj(Unconstr.) | $7.46 \times 10^{-14}$ | — | $5.39 \times 10^{-4}$ |
| | PDE2 | Unconstrained | $1.34 \times 10^{-1}$ | $3.55 \times 10^{-4}$ | $2.27 \times 10^{-3}$ |
| | | Soft Penalty | $1.45 \times 10^{-6}$ | $3.77 \times 10^{-10}$ | $5.13 \times 10^{-10}$ |
| | | Manifold | $7.44 \times 10^{-14}$ | $1.24 \times 10^{-12}$ | $1.47 \times 10^{-12}$ |
| | | Proj(Unconstr.) | $7.44 \times 10^{-14}$ | — | $2.13 \times 10^{-3}$ |
| $32 \times 32$ | PDE1 | Unconstrained | $3.69 \times 10^{1}$ | $1.18 \times 10^{-1}$ | $2.06 \times 10^{2}$ |
| | | Soft Penalty | $3.83 \times 10^{-16}$ | $7.02 \times 10^{-28}$ | $7.92 \times 10^{-28}$ |
| | | Manifold | $2.51 \times 10^{-13}$ | $2.16 \times 10^{-11}$ | $4.99 \times 10^{-11}$ |
| | | Proj(Unconstr.) | $2.66 \times 10^{-13}$ | — | $2.46 \times 10^{3}$ |
| | PDE2 | Unconstrained | $1.88 \times 10^{1}$ | $3.99 \times 10^{-3}$ | $9.75 \times 10^{1}$ |
| | | Soft Penalty | $3.49 \times 10^{-16}$ | $6.53 \times 10^{-28}$ | $1.26 \times 10^{-27}$ |
| | | Manifold | $2.51 \times 10^{-13}$ | $2.21 \times 10^{-11}$ | $5.17 \times 10^{-11}$ |
| | | Proj(Unconstr.) | $2.58 \times 10^{-13}$ | — | $9.15 \times 10^{2}$ |

The defect-controlled model improves relative errors in-distribution and yields larger gains under mesh shift, while keeping both structure metrics small. This complements the topology-shift evaluation by probing an orthogonal deployment shift: discretisation change without an explicit intended topology modification.

## M.3. Finite-Difference Poisson Validation Experiment (Hybrid vs. PINN)

Consider the standard 2D Poisson problem on the unit square

$$-\Delta u(x,y) = f(x,y), \qquad (x,y) \in [0,1]^2,$$

with analytic solution $u_{\mathrm{ana}}$ given by

$$u_{\mathrm{ana}}(x,y) = \sin(\pi x)\sin(\pi y),$$
$$f(x,y) = 2\pi^2 u_{\mathrm{ana}}(x,y),$$

and Dirichlet boundary conditions $u = g$ given by the analytic solution on $\partial[0,1]^2$. The domain $[0,1]^2$ is discretised by a uniform $41 \times 41$ Cartesian grid using a standard five-point stencil, and Dirichlet boundary conditions are imposed strongly from $u_{\mathrm{ana}}$. A diagonal mass matrix $M$ is constructed via a tensor-product trapezoidal rule. All runs use double precision on a single GPU. Compared methods:

(a) **Known-operator baseline.** Direct solve $L_{\mathrm{ref}} u_h = f$ with a dense linear solver, giving the best available FD solution on this mesh.

(b) **Hybrid operator-learning.** The operator is perturbed by a diagonal interior correction

$$L_\phi = L_{\mathrm{ref}} + \mathrm{diag}(\delta),$$

with $\delta$ non-zero only on interior nodes. Both interior degrees of freedom of $u_h$ and $\delta$ (boundary values clamped to $g$) are optimised with loss

$$\begin{aligned} \mathcal{L}_{\mathrm{hyb}} = &\tfrac{1}{2} \|L_\phi u_h - f\|_{L^2(M)}^2 + \lambda_{\mathrm{comp}} D_{\mathrm{comp}}(\phi) \\ &+ \lambda_{\mathrm{op}} \|L_\phi - L_{\mathrm{ref}}\|_{\mathrm{op}} \\ &+ \lambda_{\mathrm{data}} \|u_h - u_{\mathrm{ana}}\|_{L^2(M)}^2. \end{aligned}$$

Here $D_{\mathrm{comp}}$ is a proxy cochain-compatibility defect (the $\ell_2$ norm of the interior diagonal perturbation) and the operator deviation is approximated by power iteration. The setup uses $\lambda_{\mathrm{comp}} = \lambda_{\mathrm{op}} = 1$, $\lambda_{\mathrm{data}} = 10^{-2}$, warm-start $u_h$ from the direct solve $u_{\mathrm{ref}}$, and initialise $\delta = 0$ so that $L_\phi = L_{\mathrm{ref}}$ at step 0.

(c) **PINN baseline.** A physics-informed neural network $u_\theta(x,y)$ with Fourier features and a fully-connected

*Table 8.* Diagnostics for the degree-0 discrete Hodge Laplacian $L_0$: $\|L_0\|_{\mathrm{op}}$, deviations w.r.t. $L_{0,\mathrm{ref}}$, and condition numbers (rounded to match reporting precision).

| Mesh | PDE | Regime | $\|L_0\|_{\mathrm{op}}$ | $\|L_0 - L_{0,\mathrm{ref}}\|_{\mathrm{op}}$ | $\|L_0 - L_{0,\mathrm{ref}}\|_F$ | $\lambda_{\min}$ | $\lambda_{\max}$ | $\mathrm{cond}(L_0)$ |
|---|---|---|---|---|---|---|---|---|
| $8 \times 8$ | PDE1 | Unconstrained | 1.250625 | $3.91 \times 10^{-4}$ | $5.85 \times 10^{-4}$ | $8.37 \times 10^{-2}$ | 1.250625 | $1.49 \times 10^1$ |
| | | Soft Penalty | 1.250638 | $1.19 \times 10^{-4}$ | $1.76 \times 10^{-4}$ | $8.37 \times 10^{-2}$ | 1.250638 | $1.49 \times 10^1$ |
| | | Manifold | 1.250635 | $9.95 \times 10^{-7}$ | $6.68 \times 10^{-6}$ | $8.37 \times 10^{-2}$ | 1.250635 | $1.49 \times 10^1$ |
| | PDE2 | Unconstrained | 1.596518 | $6.37 \times 10^{-4}$ | $7.10 \times 10^{-4}$ | $9.08 \times 10^{-2}$ | 1.596518 | $1.76 \times 10^1$ |
| | | Soft Penalty | 1.596509 | $1.22 \times 10^{-4}$ | $1.76 \times 10^{-4}$ | $9.08 \times 10^{-2}$ | 1.596509 | $1.76 \times 10^1$ |
| | | Manifold | 1.596514 | $9.94 \times 10^{-7}$ | $6.66 \times 10^{-6}$ | $9.08 \times 10^{-2}$ | 1.596514 | $1.76 \times 10^1$ |
| $16 \times 16$ | PDE1 | Unconstrained | 0.641943 | $3.03 \times 10^{-3}$ | $4.30 \times 10^{-3}$ | $1.07 \times 10^{-2}$ | 0.641943 | $6.01 \times 10^1$ |
| | | Soft Penalty | 0.641936 | $2.42 \times 10^{-5}$ | $3.58 \times 10^{-5}$ | $1.07 \times 10^{-2}$ | 0.641936 | $6.01 \times 10^1$ |
| | | Manifold | 0.641936 | $9.99 \times 10^{-7}$ | $1.41 \times 10^{-5}$ | $1.07 \times 10^{-2}$ | 0.641936 | $6.01 \times 10^1$ |
| $32 \times 32$ | PDE1 | Unconstrained | 39.71406 | $3.96 \times 10^1$ | $3.96 \times 10^1$ | $2.36 \times 10^{-3}$ | 39.71406 | $1.68 \times 10^4$ |
| | | Soft Penalty | 0.323143 | $1.00 \times 10^{-6}$ | $3.10 \times 10^{-5}$ | $1.34 \times 10^{-3}$ | 0.323143 | $2.41 \times 10^2$ |
| | | Manifold | 0.323143 | $1.00 \times 10^{-6}$ | $2.92 \times 10^{-5}$ | $1.34 \times 10^{-3}$ | 0.323143 | $2.41 \times 10^2$ |

*Table 9.* **Discretization OOD evaluation on unstructured meshes.** Relative $L^2$ error (lower is better) and structure metrics (ideal value near zero) across in-distribution and out-of-distribution settings.

| Setting | Rel. $L^2$ (pressure / potential) | | Rel. $L^2$ (flux / 1-form) | | Structure metrics | |
|---|---|---|---|---|---|---|
| | Best baseline | **Ours** | Best baseline | **Ours** | Defect $\downarrow$ | Cons. resid. $\downarrow$ |
| ID (training-distribution meshes) | 0.022 | **0.020** | 0.035 | **0.033** | $3 \times 10^{-3}$ | $1.8 \times 10^{-3}$ |
| OOD-1 (finer meshes $\uparrow$) | 0.051 | **0.030** | 0.082 | **0.048** | $1.2 \times 10^{-2}$ | $2.5 \times 10^{-3}$ |
| OOD-2 (low-quality meshes) | 0.058 | **0.034** | 0.090 | **0.052** | $1.6 \times 10^{-2}$ | $3.2 \times 10^{-3}$ |

$\tanh$ MLP is trained on interior and boundary collocation points with loss

$$\mathcal{L}_{\mathrm{PINN}} = \lambda_r \, \mathbb{E}\big[(r_\theta/s_{\mathrm{PDE}})^2\big] + \lambda_{\mathrm{bc}} \, \mathbb{E}\big[(u_\theta - g)^2\big],$$

where $r_\theta(x,y) = -\Delta u_\theta(x,y) - f(x,y)$ (via AD) and $s_{\mathrm{PDE}}$ is a normalising scale for $f$. The setup uses $\lambda_r = 1$, $\lambda_{\mathrm{bc}} = 10$, 2048 interior and 512 boundary points per iteration.

Reported are the relative $L^2$ error $\|u_h - u_{\mathrm{ana}}\|_{L^2(M)}/\|u_{\mathrm{ana}}\|_{L^2(M)}$, the relative discrete PDE residual $\|L_{\mathrm{ref}} u_h - f\|_{L^2(M)}/\|f\|_{L^2(M)}$, the maximum boundary violation, the complex-defect proxy $D_{\mathrm{comp}}$, the operator deviation $\|L_\phi - L_{\mathrm{ref}}\|_{\mathrm{op}}$, and wall-clock time.

**Summary.** The direct FD solver achieves relative $L^2$ error $\approx 5.1 \times 10^{-4}$ and essentially zero discrete residual, confirming the correctness of $L_{\mathrm{ref}}$ and $M$. The hybrid scheme stays close to $L_{\mathrm{ref}}$ ($D_{\mathrm{comp}}, \|L_\phi - L_{\mathrm{ref}}\|_{\mathrm{op}} \sim 10^{-4}$) while improving the discrete solution in relative $L^2$ error (to $\approx 1.8 \times 10^{-4}$). The PINN achieves a smaller residual than the hybrid in this metric but a larger solution error and higher cost on this setup. This motivates using the hybrid discretization as a mesh-based baseline in later experiments.

## M.4. Time-Dependent PDE Benchmarks: Burgers and Allen–Cahn

Next, three physics-informed solvers are compared on two standard time-dependent PDEs (1D viscous Burgers and Allen–Cahn):

- **Classic PINN (AD):** fully continuous PINN using automatic differentiation in $(x, t)$.

- **Mesh PINN, known operator:** PINN in $t$ only, with spatial derivatives computed by a fixed finite-difference Laplacian $L_{\mathrm{FD}}$.

- **Mesh PINN, hybrid operator-learning:** same as the mesh PINN, but $L$ is learnable and regularised by (i) a structural defect term (symmetry + constant-nullspace), and (ii) an operator-deviation penalty $\|L - L_{\mathrm{FD}}\|_F$.

All methods are run for five seeds; the mean $\pm$ standard deviation is reported.

**Metrics.** Let $u_\theta$ be the prediction and $u_{\mathrm{ref}}$ a high-resolution FD reference on a space-time grid $\{(x_i, t_j)\}$:

*Table 10.* **OOD mesh-shift evaluation with jointly learned $D_0$ and $D_1$.** PDE-only uses the PDE objective alone. Anchor-only uses operator anchoring without defect control. The setting is distinct from fixed-$D_1$ and projection-only comparisons. Lower is better except for nondegeneracy of $D_1$ and OOD spectral Betti accuracy.

| Metric | PDE-only | Anchor-only | Ours |
|---|---|---|---|
| ID nRMSE | $4.18 \times 10^{-2}$ | $4.25 \times 10^{-2}$ | $4.31 \times 10^{-2}$ |
| OOD nRMSE | $7.16 \times 10^{-2}$ | $6.79 \times 10^{-2}$ | $\mathbf{6.08 \times 10^{-2}}$ |
| Raw compatibility defect | $2.84 \times 10^{-2}$ | $1.47 \times 10^{-2}$ | $\mathbf{3.32 \times 10^{-3}}$ |
| Nondegeneracy of $D_1$ | $8.7 \times 10^{-2}$ | $9.5 \times 10^{-2}$ | $\mathbf{1.07 \times 10^{-1}}$ |
| Realized drift / certified budget, median / q95 | 0.58/0.96 | 0.61/0.99 | **0.47/0.86** |
| Failure rate by certificate threshold, $\rho < 1$ / $\rho \geq 1$ | 0.038/0.361 | 0.026/0.224 | **0.009/0.128** |
| OOD spectral Betti accuracy, $\beta_0 = 2$ / $\beta_0 = 3$ | 0.803/0.681 | 0.902/0.820 | **0.982/0.951** |

*Table 11.* Finite-difference Poisson validation experiment on $[0,1]^2$ with analytic solution $u(x,y) = \sin(\pi x)\sin(\pi y)$ on a $41 \times 41$ grid.

| Method | Steps | Time [s] | Rel. $L^2$ err | Rel. PDE res. | max BC err | $D_{\mathrm{comp}}$ | $\|L_\phi - L_{\mathrm{ref}}\|_{\mathrm{op}}$ |
|---|---|---|---|---|---|---|---|
| Known operator (direct solve) | – | 0.06 | $5.14 \times 10^{-4}$ | $9.55 \times 10^{-14}$ | 0 | 0 | 0 |
| Hybrid operator-learning | 3000 | 7.69 | $1.79 \times 10^{-4}$ | $3.25 \times 10^{-2}$ | 0 | $2.31 \times 10^{-4}$ | $1.79 \times 10^{-4}$ |
| PINN baseline | 5000 | 36.64 | $2.70 \times 10^{-3}$ | $1.38 \times 10^{-2}$ | $3.21 \times 10^{-3}$ | – | – |

- **Space–time $L^2$ error**

$$\text{st-}L^2 \approx \left( \sum_{i,j} (u_\theta(x_i, t_j) - u_{\mathrm{ref}}(x_i, t_j))^2 \, \Delta x \Delta t \right)^{1/2}.$$

- **Final-time relative $L^2$ error**

$$\text{final\_rel-}L^2 \approx \frac{\|u_\theta(\cdot, T) - u_{\mathrm{ref}}(\cdot, T)\|_{L^2}}{\|u_{\mathrm{ref}}(\cdot, T)\|_{L^2}}.$$

- **$L^\infty$ error**: $L^\infty = \max_{i,j} |u_\theta(x_i, t_j) - u_{\mathrm{ref}}(x_i, t_j)|$.

- **PDE residual $L^2$ (AD)**: continuous residual norm $L^2$ evaluated via AD on the grid.

- **Burgers mass discrepancy**: $\sup_{t_j} |M_\theta(t_j) - M_{\mathrm{ref}}(t_j)|$, $M(t) = \int u(x,t)\,dx$.

- **Allen–Cahn energy discrepancy**: $\sup_{t_j} |E_\theta(t_j) - E_{\mathrm{ref}}(t_j)|$, with $E[u] = \int \left( \frac{\varepsilon}{2}|\partial_x u|^2 + \frac{1}{4}(u^2 - 1)^2 \right) dx$.

- **Train time**: wall-clock seconds (Adam + L-BFGS on a single GPU).

- **Structural defect (hybrid only)**: Frobenius norm of symmetry and constant-nullspace violations of the interior Laplacian block.

- **Operator deviation (hybrid only)**: $\|L - L_{\mathrm{FD}}\|_{\mathrm{F}}$.

### M.4.1. BURGERS RESULTS

**Summary (Burgers).** Mesh-based solvers reduce space–time error from $0.123 \pm 0.032$ (classic PINN) to $0.097 \pm 0.008$ (known operator) and $0.100 \pm 0.006$ (hybrid), with

*Table 12.* Burgers benchmark: mean $\pm$ std over 5 random seeds. For the hybrid model, the last column reports *structural defect / operator deviation.*

| Method | st-$L^2$ | final rel-$L^2$ | $L^\infty$ | res-$L^2$ | mass disc. | time [s] | struct / opdev |
|---|---|---|---|---|---|---|---|
| Classic PINN (AD) | $0.123 \pm 0.032$ | $0.198 \pm 0.059$ | $0.530 \pm 0.115$ | $0.288 \pm 0.068$ | $0.098 \pm 0.024$ | $11.15 \pm 0.28$ | – |
| Mesh PINN, known $L_{\mathrm{FD}}$ | $0.097 \pm 0.008$ | $0.096 \pm 0.008$ | $0.511 \pm 0.031$ | $0.336 \pm 0.016$ | $0.068 \pm 0.003$ | $5.28 \pm 0.05$ | – |
| Mesh PINN, hybrid $L$ | $0.100 \pm 0.006$ | $0.100 \pm 0.016$ | $0.541 \pm 0.023$ | $0.343 \pm 0.007$ | $0.074 \pm 0.003$ | $5.73 \pm 0.06$ | $0.060 \pm 0.006 / 0.018 \pm 0.006$ |

*Table 13.* Allen–Cahn benchmark: mean $\pm$ std over 5 random seeds. For the hybrid model, the last column reports *structural defect / operator deviation.*

| Method | st-$L^2$ | final rel-$L^2$ | $L^\infty$ | res-$L^2$ | energy disc. | time [s] | struct / opdev |
|---|---|---|---|---|---|---|---|
| Classic PINN (AD) | $0.017 \pm 0.007$ | $0.036 \pm 0.016$ | $0.042 \pm 0.018$ | $0.034 \pm 0.010$ | $0.0020 \pm 0.0010$ | $11.08 \pm 0.36$ | – |
| Mesh PINN, known $L_{\mathrm{FD}}$ | $0.016 \pm 0.002$ | $0.029 \pm 0.005$ | $0.032 \pm 0.005$ | $0.028 \pm 0.003$ | $0.0019 \pm 0.0005$ | $8.31 \pm 0.02$ | – |
| Mesh PINN, hybrid $L$ | $0.019 \pm 0.001$ | $0.034 \pm 0.003$ | $0.035 \pm 0.004$ | $0.031 \pm 0.002$ | $0.0019 \pm 0.0004$ | $8.65 \pm 0.04$ | $0.183 \pm 0.010 / 0.034 \pm 0.008$ |

roughly a factor-of-two reduction in final-time relative error and a $\sim 2\times$ speed-up in training. The classic PINN achieves the smallest AD residual but the largest solution error and mass discrepancy on this setup, illustrating that a low continuous residual is not a reliable proxy for solution quality. The hybrid model stays close to $L_{\mathrm{FD}}$ (small structural defect and operator deviation) and tracks the accuracy of the known-operator mesh model.

### M.4.2. ALLEN–CAHN RESULTS

**Summary (Allen–Cahn).** All three methods achieve low errors (space–time $L^2$ in the 1.6–1.9% range). The known-operator mesh PINN is marginally most accurate and has the smallest residual and variance; the hybrid model stays close in all metrics while perturbing $L_{\mathrm{FD}}$ more than in Burgers. Mesh-based PINNs are $\sim 25\%$ faster than the classic PINN on this setup.

*Table 14.* High-level summary comparing the classic PINN and mesh-based PINNs across both PDE benchmarks (mean values). "Mesh" aggregates the known-operator and hybrid variants.

| Metric | Classic PINN | Mesh PINNs | Verdict |
|---|---|---|---|
| Training time [s] | $\approx 11.1$ | $\approx 5.5$ (Burgers), $\approx 8.5$ (Allen–Cahn) | $\sim 2\times$ faster on Burgers |
| Stability (st-$L^2$ std, Burgers) | $3.21 \times 10^{-2}$ | $\sim 0.8 \times 10^{-2}$ | lower variance |
| Burgers st-$L^2$ error | $\approx 12.3\%$ | $\approx 9.7\%$–$10.0\%$ | more accurate |
| Allen–Cahn st-$L^2$ error | $\approx 1.7\%$ | $\approx 1.6\%$–$1.9\%$ | comparable accuracy |

*Table 15.* Scaling study for the 2D Poisson problem (trial = 0).

| N | Method | Collocation | st-L2 | rel-L2 | PDE-MSE | Time [s] | Residual evals | Peak mem [MB] | # Params | op_dev |
|---|---|---|---|---|---|---|---|---|---|---|
| 32 | Classic PINN | grid (AD) | 0.001 240 110 3 | 0.002 560 227 6 | $4.4827 \times 10^{-4}$ | 3.92 | 1894500 | 31.48 | 8577 | – |
| 32 | Mesh-known | mesh-discrete | 0.000 414 844 4 | 0.000 856 453 0 | $2.6422 \times 10^{-7}$ | 0.06 | 48600 | 17.73 | 900 | – |
| 32 | Mesh-hybrid | mesh-discrete | 0.000 133 409 5 | 0.000 275 426 0 | $1.5013 \times 10^{-7}$ | 0.10 | 64800 | 17.87 | 901 | $5.806 \times 10^{-4}$ |
| 64 | Classic PINN | grid (AD) | 0.001 184 973 1 | 0.002 407 564 1 | $1.6663 \times 10^{-4}$ | 3.94 | 8091620 | 65.51 | 8577 | – |
| 64 | Mesh-known | mesh-discrete | 0.000 101 271 2 | 0.000 205 757 3 | $4.0335 \times 10^{-6}$ | 0.36 | 591976 | 20.63 | 3844 | – |
| 64 | Mesh-hybrid | mesh-discrete | 0.000 063 112 5 | 0.000 128 228 5 | $3.6981 \times 10^{-6}$ | 0.39 | 699608 | 20.67 | 3845 | $7.871 \times 10^{-5}$ |
| 128 | Classic PINN | grid (AD) | 0.001 007 941 6 | 0.002 031 756 1 | $5.1608 \times 10^{-4}$ | 4.21 | 33450732 | 210.20 | 8577 | – |
| 128 | Mesh-known | mesh-discrete | 0.000 033 802 9 | 0.000 068 138 0 | $9.3577 \times 10^{-7}$ | 1.09 | 5826492 | 30.54 | 15876 | – |
| 128 | Mesh-hybrid | mesh-discrete | 0.000 010 482 0 | 0.000 021 129 0 | $9.6991 \times 10^{-7}$ | 1.15 | 6683796 | 30.79 | 15877 | $3.091 \times 10^{-5}$ |

### M.4.3. HEADLINE COMPARISON AND TAKEAWAYS

Across both PDEs, explicit spatial operators (mesh PINNs) match or improve the classic AD-based PINN in accuracy, are faster to train, and are more stable over seeds in these experiments. In Burgers (steep gradients), they reduce both error and mass drift. In Allen–Cahn (moderately stiff dynamics), all methods are close in accuracy, while mesh PINNs reduce wall time. The hybrid operator remains close to the fixed mesh operator: structural defect and operator deviation remain small, and accuracy tracks the fixed-operator mesh baseline.

## M.5. Poisson Scaling and Ablation Results

Scaling with grid resolution and the effect of collocation layout and residual formulation are studied on a 2D Poisson problem, comparing a classic AD-based PINN, a mesh-based PINN with known operator, and a mesh-based hybrid operator-learning variant. All models are trained on the same analytic Poisson solution and evaluated on structured grids of size $N \times N$.

**Summary.** Across all resolutions, mesh-based PINNs (known and hybrid) yield smaller solution and discrete-residual errors, dramatically fewer residual evaluations, lower memory, and smaller wall time than the classic PINN in these runs. The ablation indicates that grid vs. random collocation has marginal effect compared to switching from AD residuals to mesh-discrete residuals.

## M.6. Topological Stability on an Annulus

The topological robustness of hybrid operator learning is tested on an annulus

$$\Omega_{\text{ann}} = [0,1]^2 \setminus B_{r_{\text{hole}}}(c),$$
$$c = (0.5, 0.5), \quad r_{\text{hole}} = 0.15,$$

which has Betti numbers $\beta_0 = 1$, $\beta_1 = 1$.

**DEC complex and reference Laplacians.** The domain $\Omega_{\text{ann}}$ is discretised on a structured $N \times N$ grid with $N =$

*Table 16.* Ablation study at $N = 64$ (trial = 0): collocation layout and residual formulation.

| N | Method | Collocation | st-L2 | rel-L2 | PDE-MSE | Time [s] | Residual evals | Peak mem [MB] | # Params |
|---|---|---|---|---|---|---|---|---|---|
| 64 | Classic PINN | random (AD) | 0.000 975 196 0 | 0.001 981 350 4 | $3.2784 \times 10^{-4}$ | 3.78 | 8095464 | 65.51 | 8577 |
| 64 | Classic PINN | grid (AD) | 0.001 258 159 6 | 0.002 556 260 7 | $3.1670 \times 10^{-4}$ | 3.82 | 8087776 | 65.51 | 8577 |
| 64 | Mesh-known | mesh-discrete | 0.000 101 271 2 | 0.000 205 757 3 | $4.0335 \times 10^{-6}$ | 0.35 | 591976 | 20.63 | 3844 |

*Table 17.* Betti numbers on the annulus from the reference DEC complex and the hybrid learned complex.

| Complex | $\beta_0 = \dim \ker L_0$ | $\beta_1 = \dim \ker L_1$ |
|---|---|---|
| Reference DEC | 1 | 1 |
| Hybrid learned DEC | 1 | 1 |

20, masking out nodes in the hole. The resulting DEC complex has 376 vertices (0-forms), 700 edges (1-forms) with incidence $D_0 \in \mathbb{R}^{700 \times 376}$, and 324 faces (2-forms) with incidence $D_1 \in \mathbb{R}^{324 \times 700}$. The unweighted Hodge Laplacians are

$$L_0 = D_0^\top D_0, \qquad L_1 = D_1^\top D_1 + D_0 D_0^\top.$$

Counting eigenvalues below $10^{-3}$ gives $\beta_0^{\text{ref}} = 1$, $\beta_1^{\text{ref}} = 1$, with $L_1$ having a single near-zero eigenvalue and a spectral gap at $\approx 1.9 \times 10^{-2}$.

**Hybrid learned complex.** The incidence matrices are perturbed

$$D_0^{(\theta)} = D_0 + P_0, \qquad D_1^{(\theta)} = D_1 + P_1,$$

with trainable dense $P_0, P_1$ (Gaussian initialisation, $\sigma = 10^{-4}$). The learned Laplacians are

$$L_0^{(\theta)} = (D_0^{(\theta)})^\top D_0^{(\theta)},$$
$$L_1^{(\theta)} = (D_1^{(\theta)})^\top D_1^{(\theta)} + D_0^{(\theta)} (D_0^{(\theta)})^\top.$$

Training uses a purely operator-level objective

$$\mathcal{L}_{\text{topo}}(\theta) = \|D_1^{(\theta)} D_0^{(\theta)}\|_F + 0.1 \big(\|P_0\|_F + \|P_1\|_F\big) + 0.1\|P_0\|_F^2,$$

optimised with Adam (lr $10^{-3}$, 1000 epochs). The cochain-compatibility defect decreases from $\approx 9.7 \times 10^{-2}$ at epoch 0 to $\approx 7.3 \times 10^{-2}$ by epoch 800.

**Results.** The hybrid model preserves $\beta_0$ and $\beta_1$ in this setup, and the smallest eigenvalue remains near zero with an essentially unchanged spectral gap. This is consistent with the spectral Betti stability mechanism: small cochain-compatibility defects and small operator deviations need not change the topology encoded by the Laplacian spectrum.

## M.7. Step 1: Elliptic Validation Experiment on a Poisson Problem (Degree-0 Discrete Hodge Laplacian)

Defect-controlled operator learning is tested on a degree-0 discrete Hodge Laplacian for the classical 2D Poisson

*Table 18.* Smallest five eigenvalues of the edge Laplacian $L_1$ on the annulus, before and after hybrid operator learning.

| Complex | $\lambda_0$ | $\lambda_1$ | $\lambda_2$ | $\lambda_3$ | $\lambda_4$ |
|---|---|---|---|---|---|
| Reference DEC | $-2.28 \times 10^{-16}$ | $1.90 \times 10^{-2}$ | $1.90 \times 10^{-2}$ | $4.75 \times 10^{-2}$ | $9.31 \times 10^{-2}$ |
| Hybrid learned DEC | $7.78 \times 10^{-8}$ | $1.90 \times 10^{-2}$ | $1.90 \times 10^{-2}$ | $4.75 \times 10^{-2}$ | $9.31 \times 10^{-2}$ |

problem

$$-\Delta u(x,y) = f(x,y), \qquad (x,y) \in \Omega := [0,1]^2,$$

with Dirichlet boundary conditions $u|_{\partial\Omega} = u_{\text{ana}}$ given by

$$u_{\text{ana}}(x,y) = \sin(\pi x)\sin(\pi y),$$
$$f(x,y) = 2\pi^2 u_{\text{ana}}(x,y),$$

and constant conductivity $\kappa \equiv 1$.

**Discrete setting.** The domain $\Omega$ is discretised with a structured grid of $N \times N$ squares, each split into two triangles, giving $(N+1)^2$ vertices and $2N^2$ faces. The setup uses

$$N \in \{16, 32\},$$

corresponding to:

- $N = 16$: 289 vertices (225 interior, 64 boundary), 800 edges, 512 faces;

- $N = 32$: 1089 vertices (961 interior, 128 boundary), 3136 edges, 2048 faces.

A DEC-/graph-style complex is assembled with incidence matrices $D_0$ (vertices→edges) and $D_1$ (edges→faces) and Hodge-like weights $W_0, W_1$:

- $W_0$: lumped vertex mass (triangle areas split equally across vertices);

- $W_1$: diagonal edge weights $w_e \approx \kappa_e/h_e^2$.

This yields
$$L_0 = D_0^\top W_1 D_0.$$

Homogeneous Dirichlet boundary conditions are imposed by restricting to interior vertices and solving $(L_0)_{\text{ii}} u_{\text{int}} = f_{\text{int}}$, then setting $u|_{\partial\Omega} = 0$.

**Operator variants.** Compared methods:

(i) **Reference operator (`ref`).** $L_0^{\text{ref}} = D_0^\top W_1 D_0$ with $D_1 D_0 = 0$ by construction; a low-order discretization of $-\Delta$.

(ii) **Unconstrained learned operator (`uncon`).** Low-rank perturbation

$$D_0(\theta) = D_0^{\text{ref}} + UV^\top,$$

$U \in \mathbb{R}^{N_e \times r}$, $V \in \mathbb{R}^{N_v \times r}$, $r = 32$, and $L_0(\theta) = D_0(\theta)^\top W_1 D_0(\theta)$. Loss:

$$\mathcal{L}_{\text{PDE}}(\theta) = \frac{1}{|\mathcal{V}_{\text{int}}|} \sum_{i \in \mathcal{V}_{\text{int}}} \big(u_i(\theta) - u_{\text{ana}}(x_i, y_i)\big)^2,$$

with $u(\theta)$ obtained from the Dirichlet solve.

(iii) **Soft-defect regime (`soft`).** Same parameterisation with complex-defect penalty

$$D_{\text{comp}}(D_0) := \|D_1 D_0\|_{\text{F}},$$

and loss

$$\mathcal{L}_{\text{soft}}(\theta) = \mathcal{L}_{\text{PDE}}(\theta) + \lambda_{\text{defect}}\, \widetilde{D}_{\text{comp}}(D_0(\theta))^2,$$

where $\widetilde{D}_{\text{comp}}$ is a mesh-normalized version and $\lambda_{\text{defect}} = 10^3$.

(iv) **Manifold regime (`manifold`).** Compatibility-preserving parameterization

$$D_0(\theta) = K\big(R_{\text{ref}} + \Delta R\big),$$

where $K$ spans $\ker(D_1)$ and $R_{\text{ref}} = K^\top D_0^{\text{ref}}$ is the optimal projection. Loss:

$$\begin{aligned}
\mathcal{L}_{\text{man}}(\theta) = \mathcal{L}_{\text{PDE}}(\theta) \\
+ \lambda_{\text{re}} \left\| L_0(\theta) - L_0^{\text{ref}} \right\|_{\text{F}}^2 \\
+ \lambda_{\Delta R} \left\| \Delta R \right\|_{\text{F}}^2,
\end{aligned}$$

with $(\lambda_{\text{re}}, \lambda_{\Delta R}) = (1, 10^{-2})$, and $D_1 D_0(\theta) = 0$ for all $\theta$ by construction.

(v) **Projection baseline (`proj-uncon`).** Take trained $D_0^{\text{uncon}}$ from the unconstrained run, project onto the chain-compatible subspace,

$$\widetilde{D}_0 := K K^\top D_0^{\text{uncon}},$$

set $\widetilde{L}_0 = \widetilde{D}_0^\top W_1 \widetilde{D}_0$, and re-solve the PDE.

All models use Adam (lr $10^{-3}$, 2000 steps, same seed). Metrics: weighted $L^2$ error, $H^1$ seminorm error, corresponding relative errors, relative flux error (weighted difference of $D_0 u$ vs $D_0 u_{\text{ana}}$), and raw cochain-compatibility defect $D_{\text{comp}} = \|D_1 D_0\|_F$.

**Summary.** The reference Laplacian satisfies $D_{\text{comp}} = 0$ but has $\mathcal{O}(10^{-1})$ relative $L^2/H^1$ errors. Unconstrained learning drives errors down to $\mathcal{O}(10^{-5})$–$\mathcal{O}(10^{-4})$, but at large cochain-compatibility defects $D_{\text{comp}} \sim 10^{-1}$. The soft-defect regime reduces $D_{\text{comp}}$ by two orders of magnitude while keeping relative errors extremely small (typically $\mathcal{O}(10^{-4})$ or better). Post-hoc projection enforces $D_{\text{comp}} \approx 0$ but destroys most of the PDE accuracy, and

*Table 19.* Poisson validation experiment on the $N = 16$ mesh (289 vertices). Relative $L^2/H^1$/flux errors vs analytic solution and cochain-compatibility defect.

| Variant | $D_{\text{comp}}$ | $\|e\|_{L^2}/\|u\|_{L^2}$ | $\|e\|_{H^1}/\|u\|_{H^1}$ | Flux rel. error |
|---|---|---|---|---|
| Ref. | 0 | $3.21 \times 10^{-1}$ | $3.40 \times 10^{-1}$ | $3.40 \times 10^{-1}$ |
| Unconstr. | $1.65 \times 10^{-1}$ | $5.61 \times 10^{-5}$ | $5.64 \times 10^{-5}$ | $5.64 \times 10^{-5}$ |
| Soft | $1.38 \times 10^{-3}$ | $1.32 \times 10^{-5}$ | $5.66 \times 10^{-5}$ | $5.66 \times 10^{-5}$ |
| Manifold | $8.89 \times 10^{-6}$ | $3.21 \times 10^{-1}$ | $3.40 \times 10^{-1}$ | $3.40 \times 10^{-1}$ |
| Proj-uncon. | $8.19 \times 10^{-14}$ | $3.19 \times 10^{-2}$ | $5.46 \times 10^{-2}$ | $5.46 \times 10^{-2}$ |

*Table 20.* Same experiment as Table 19 on the $N = 32$ mesh (1089 vertices).

| Variant | $D_{\text{comp}}$ | $\|e\|_{L^2}/\|u\|_{L^2}$ | $\|e\|_{H^1}/\|u\|_{H^1}$ | Flux rel. error |
|---|---|---|---|---|
| Ref. | 0 | $3.24 \times 10^{-1}$ | $3.44 \times 10^{-1}$ | $3.44 \times 10^{-1}$ |
| Unconstr. | $1.26 \times 10^{-1}$ | $2.85 \times 10^{-4}$ | $3.24 \times 10^{-4}$ | $3.24 \times 10^{-4}$ |
| Soft | $4.66 \times 10^{-2}$ | $3.35 \times 10^{-4}$ | $2.23 \times 10^{-3}$ | $2.23 \times 10^{-3}$ |
| Manifold | $3.87 \times 10^{-5}$ | $3.23 \times 10^{-1}$ | $3.44 \times 10^{-1}$ | $3.44 \times 10^{-1}$ |
| Proj-uncon. | $2.64 \times 10^{-13}$ | $6.79 \times 10^{-2}$ | $8.11 \times 10^{-2}$ | $8.11 \times 10^{-2}$ |

subspace-constrained training stays close to the reference operator and inherits its relatively large PDE error. This illustrates the core trade-off: the cochain-compatibility defect penalty is an effective continuous parameter that reduces complex violations while keeping PDE error in a useful regime, whereas compatibility-by-construction constraints alone are insufficient to guarantee good discretizations.

## M.8. Experiment 2.1: Hole Creation and Destruction

This experiment tests whether learning a discrete operator purely from PDE solution data can infer incorrect topology, and whether the cochain-compatibility defect penalty can prevent such failures.

**Geometry and complexes.** The setup uses a structured triangular mesh on $\Omega = [0, 1]^2$ with $(n_x, n_y) = (32, 32)$ (two triangles per rectangle) and construct two domains:

- a *solid* square (no hole);

- an *annulus* obtained by removing triangles whose centroid lies in a circle of radius $r_{\text{hole}} = 0.25$ centred at $(0.5, 0.5)$.

Both yield a DEC/FEEC complex

$$C^0 \xrightarrow{D_0^{\text{ref}}} C^1 \xrightarrow{D_1^{\text{ref}}} C^2, \qquad D_1^{\text{ref}} D_0^{\text{ref}} = 0,$$

with diagonal $W_1^{\text{ref}}$ and $L_0^{\text{ref}} = (D_0^{\text{ref}})^\top W_1^{\text{ref}} D_0^{\text{ref}}$. For the solid: $(n_0, n_1, n_2) = (1089, 3136, 2048)$; for the annulus: $(1089, 2562, 1646)$. In both cases $D_{\text{comp}}(D_0^{\text{ref}}) = 0$ and $\beta_0 = 1$.

**Learned operator and metrics.** A low-rank perturbation is learned $D_0^\phi = D_0^{\text{ref}} + UV^\top$ ($r = 32$), with fixed $W_1^{\text{ref}}$, and define

$$L_0^\phi = (D_0^\phi)^\top W_1^{\text{ref}} D_0^\phi.$$

*Table 21.* Wall-clock training time (seconds) for each complex-defect regime (2000 optimization steps, RTX 4090).

| Mesh | Unconstr. | Soft-defect | Manifold |
|---|---|---|---|
| $N = 16$ | 1.79 | 1.92 | 1.89 |
| $N = 32$ | 7.86 | 9.38 | 13.96 |

*Table 22.* Hole creation/destruction on the solid domain (no hole).

| Regime | $D_{\text{comp}}$ | $\beta_{0,\text{spec}}$ | $\beta_{0,\text{comb}}$ | $\|L_0^\phi - L_0^{\text{ref}}\|_2$ | rel. $L^2$ (train) | rel. $L^2$ (shift) |
|---|---|---|---|---|---|---|
| Baseline | $4.94 \times 10^1$ | 0 | 0 | $2.63 \times 10^1$ | $5.44 \times 10^{-2}$ | $6.10 \times 10^{-2}$ |
| Defect-reg. | $9.29 \times 10^{-5}$ | 1 | 1 | $1.46 \times 10^{-4}$ | $1.31 \times 10^{-6}$ | $1.28 \times 10^{-6}$ |

Tracked quantities include cochain-compatibility defect $D_{\text{comp}}(\phi)$, spectral and combinatorial Betti numbers $\beta_{0,\text{spec}}, \beta_{0,\text{comb}}$, operator deviations $\|L_0^\phi - L_0^{\text{ref}}\|_2$, and train/shifted PDE errors (MSE and relative $L^2$).

**PDE data.** Solutions are generated by solving $L_0^{\text{ref}} u = f$ with homogeneous Dirichlet boundary conditions, where

$$f(x, y) = \sum_{i=1}^4 a_i \varphi_i(x, y),$$

with trigonometric modes $\varphi_i$ and $a_i \sim \mathcal{N}(0, \sigma^2)$. The setup uses 256 train forcings ($\sigma = 1$) and 64 shifted forcings ($\sigma = 2$).

**Training regimes.**

**Baseline.**
PDE-only loss $\mathcal{L}_{\text{PDE}}(\phi) = \mathbb{E}\|u^\phi - u^{\text{ref}}\|_2^2$, with $\lambda_{\text{defect}} = \lambda_{\text{opdev}} = 0$.

**Defect-regularised.**
Add complex-defect and operator-deviation penalties

$$\mathcal{L}(\phi) = \mathcal{L}_{\text{PDE}}(\phi) + \lambda_{\text{defect}} D_{\text{comp}}(\phi)^2 + \lambda_{\text{opdev}}\|L_0^\phi - L_0^{\text{ref}}\|_F^2,$$

with $\lambda_{\text{defect}} = 10^3, \lambda_{\text{opdev}} = 10^{-1}$.

Training runs for 1000 Adam steps (lr $10^{-3}$, batch size 16).

**Summary.** On the solid domain, PDE-only training drives the complex far from the chain-compatibility condition (substantial defect, incorrect Betti numbers, large operator deviation) and produces relatively large PDE errors. Defect-regularisation nearly restores chain-compatibility ($D_{\text{comp}} \approx 10^{-4}$, correct $\beta_0$), keeps $L_0^\phi$ extremely close to $L_0^{\text{ref}}$, and yields near machine-precision PDE errors. On the annulus, the baseline achieves noticeably small PDE errors but significantly degrades topology and the operator, whereas defect-regularisation reduces defect by four orders of magnitude, recovers the correct *spectral* Betti number,

*Table 23.* Hole creation/destruction on the annulus (square with circular hole).

| Regime | $D_{\mathrm{comp}}$ | $\beta_{0,\mathrm{spec}}$ | $\beta_{0,\mathrm{comb}}$ | $\|L_0^\phi - L_0^{\mathrm{ref}}\|_2$ | rel. $L^2$ (train) | rel. $L^2$ (shift) |
|---|---|---|---|---|---|---|
| Baseline | $8.75 \times 10^1$ | 0 | 0 | $1.35 \times 10^2$ | $1.31 \times 10^{-3}$ | $1.31 \times 10^{-3}$ |
| Defect-reg. | $2.30 \times 10^{-2}$ | 1 | 0 | $7.36 \times 10^{-1}$ | $1.79 \times 10^{-4}$ | $1.77 \times 10^{-4}$ |

*Table 24.* Step 3.1: coefficient/forcing distribution shift. Reference DEC vs learned regimes.

| Regime | $D_{\mathrm{comp}}(\phi)$ | $\|L_0 - L_0^{\mathrm{ref}}\|_F$ | ID $L^2$ | OOD $L^2$ | OOD/ID | ID rel. $L^2$ | OOD rel. $L^2$ | OOD/ID (rel) |
|---|---|---|---|---|---|---|---|---|
| Reference DEC | 0 | 0 | 13.26 | 467.3 | 35.2 | 7.01 | 9.45 | 1.35 |
| Unconstrained | $\approx 45.1$ | $\approx 53.9$ | 0.0113 | 73.5 | $6.49 \times 10^3$ | 0.167 | 3.58 | 21.4 |
| Soft defect | $\approx 0.356$ | $\approx 5.89$ | 0.119 | 616 | $5.17 \times 10^3$ | 0.658 | 10.9 | 16.5 |
| Manifold | 0 | $\approx 0.252$ | 0.0583 | 442 | $7.58 \times 10^3$ | 0.425 | 9.15 | 21.5 |

*Table 25.* Step 3.2: mesh-quality metrics.

| Scenario | Mean aspect ratio | Min angle [rad] | Max angle [rad] |
|---|---|---|---|
| base (reference) | 1.414 | 0.785 | 1.571 |
| stretched_x | 2.390 | 0.108 | 1.571 |
| jittered | 1.663 | 0.145 | 2.709 |

keeps $L_0^\phi$ in a neighbourhood of $L_0^{\mathrm{ref}}$, and improves PDE accuracy by an order of magnitude. This shows that topological failure modes are undetected by standard PDE metrics, but are controlled by chain-condition-defect regularisation. Key finding: defect control better preserves the near-zero spectral structure consistent with the target topology in our setup.

## M.9. Step 3: OOD Robustness under Coefficient and Geometry Shift

An evaluation under distribution shift considers degree-0 discrete Hodge Laplacians under (i) coefficient/forcing distribution shift and (ii) mesh/geometry shift. In all cases

$$L_0(\phi) = D_0(\phi)^\top W_1 D_0(\phi),$$

with homogeneous Dirichlet boundary conditions and discrete $L^2$ errors evaluated via $W_0$. Three regimes are compared:

- **Unconstrained**: $D_0(\phi) = D_0^{\mathrm{ref}} + \Delta D_0(\phi)$ with PDE-only training.

- **Soft defect**: same parameterisation plus quadratic penalty on $D_{\mathrm{comp}}(\phi) = \|D_1 D_0(\phi)\|_F$ (weight $\lambda_{\mathrm{defect}} = 10$).

- **Manifold-constrained**: $D_0(\phi) = K(R_{\mathrm{ref}} + \Delta R)$ with $D_1 D_0(\phi) \equiv 0$ and tether to $L_0^{\mathrm{ref}}$ ($\lambda_{\mathrm{re}} = 1$, $\lambda_{\Delta R} = 0.1$).

All are trained with Adam (lr $10^{-2}$, 200 epochs, 64 PDE samples/epoch).

### 3.1 COEFFICIENT / FORCING DISTRIBUTION SHIFT

The setup uses analytic solutions

$$u(x,y) = \mathtt{amp}\, \sin(k_x \pi x)\, \sin(k_y \pi y)$$

with homogeneous Dirichlet boundary conditions and $-\Delta u = f = \lambda u$ with $\lambda = (k_x^2 + k_y^2)\pi^2$. The in-distribution (ID) training/test uses $\mathtt{amp} \sim \mathcal{U}(0.5, 1.5)$, $(k_x, k_y) \in \{1,2\}^2$; the OOD test uses $\mathtt{amp} \sim \mathcal{U}(3,6)$, $(k_x, k_y) \in \{3,4\}^2$. For each regime, mean ID/OOD squared $L^2$ errors and relative errors arrors over 128 PDE instances.

All learned regimes dramatically reduce ID error relative to Reference DEC; the unconstrained regime is most accurate in absolute terms but has very large cochain-compatibility defect and operator deviation. In absolute OOD error, all learned operators outperform DEC; the manifold regime

matches or slightly improves on DEC while preserving chain-compatible complex structure. However, the relative degradation factors (OOD/ID) are orders of magnitude worse for learned operators, indicating significantly weaker robustness under coefficient shift unless structural constraints are imposed. Key finding: the cochain-compatibility defect penalty maintains low defect and avoids structural violation under severe coefficient shifts.

### 3.2 MESH / GEOMETRY SHIFT

Combinatorics are fixed and only vertex positions change, recomputing $W_0, W_1$:

- **base**: original quasi-uniform mesh;

- **stretched_x**: $(x,y) \mapsto (x^\beta, y)$, $\beta = 1.8$;

- **jittered**: random perturbations of interior vertices.

For each geometry, $L_0^{\mathrm{ref}}$ and $L_0(\phi)$ are recomputed and measure ID/OOD squared $L^2$ errors, relative degradation, spectral shifts, and operator-norm differences.

**Summary.** All learned operators significantly reduce ID error relative to Reference DEC. Under geometry shift, the unconstrained regime attains the smallest absolute OOD errors but exhibits large spectral and operator deviations, indicating strong geometry sensitivity. The manifold regime matches the DEC spectral and operator-norm behaviour very closely and slightly improves average OOD error. The soft-defect regime reduces spectral drift relative to unconstrained but can overshoot in OOD errors. Overall, chain-condition-defect control and manifold parameterisations provide a principled trade-off between accuracy and structural robustness. Key finding: manifold constraints and defect penalties prevent spectral collapse on deformed meshes.

## M.10. Ablation: Complex-Defect Regularisation

Controlled ablations are performed on a canonical elliptic model problem on an $8 \times 8$ triangular mesh, with reference

*Table 26.* Step 3.2: mesh/geometry shift.

| Scenario | Regime | ID $L^2$ | OOD $L^2$ | OOD/ID | $\|$spec shift$\|_\infty$ | $\|L_0^{\text{geom}} - L_0^{\text{base}}\|_2$ |
|---|---|---|---|---|---|---|
| base | Reference DEC | 12.27 | 474.4 | 38.7 | 0 | 0 |
| | Unconstrained | $1.62 \times 10^{-2}$ | 80.3 | $5.0 \times 10^3$ | 0 | 0 |
| | Soft defect | $1.59 \times 10^{-1}$ | 592.2 | $3.72 \times 10^3$ | 0 | 0 |
| | Manifold | $5.20 \times 10^{-2}$ | 446.1 | $8.58 \times 10^3$ | 0 | 0 |
| stretched_x | Reference DEC | 13.39 | 366.0 | 27.3 | 0.186 | 0.261 |
| | Unconstrained | $4.68 \times 10^{-2}$ | 89.9 | $1.92 \times 10^3$ | 0.943 | 3.42 |
| | Soft defect | 5.13 | 435.7 | 85.0 | 0.186 | 0.332 |
| | Manifold | 1.09 | 328.7 | $3.02 \times 10^2$ | 0.186 | 0.261 |
| jittered | Reference DEC | 12.59 | 485.6 | 38.6 | 0.038 | 0.117 |
| | Unconstrained | $1.35 \times 10^{-2}$ | 74.5 | $5.50 \times 10^3$ | 0.646 | 1.16 |
| | Soft defect | $1.30 \times 10^{-1}$ | 633.9 | $4.89 \times 10^3$ | 0.120 | 0.287 |
| | Manifold | $6.93 \times 10^{-2}$ | 450.6 | $6.50 \times 10^3$ | 0.038 | 0.117 |

operator

$$L_0^{\text{ref}} = D_0^\top W_1 D_0 + \mu I, \qquad \mu = 10^{-3},$$

and $f = L_0^{\text{ref}} u_{\text{ref}}$ so that $u_{\text{ref}}$ is the discrete reference solution. The learned Laplacian

$$L_{0,\phi} = D_{0,\phi}^\top W_1 D_{0,\phi} + \mu I$$

is trained from noisy initialisation $D_{0,\phi} = D_0 + 5 \cdot 10^{-3}\Xi$ ($\Xi$ Gaussian). The cochain-compatibility defect is $D_{\text{comp}}(\phi) = \|D_1 D_{0,\phi}\|_F$.

The primary loss is

$$\mathcal{L}_{\text{PDE}}(\phi) = \|u_\phi - u_{\text{ref}}\|_2^2, \qquad u_\phi = L_{0,\phi}^{-1} f,$$

with optional regularisers

$$\mathcal{L}_{\text{defect}}(\phi) = D_{\text{comp}}(\phi)^2,$$
$$\mathcal{L}_{\text{Frob}}(\phi) = \|D_{0,\phi}\|_F^2,$$
$$\mathcal{L}_{\text{spec}}(\phi) = \|L_{0,\phi}\|_2.$$

These ablations allow one to separate the specific effect of penalising the cochain-compatibility defect from more generic weight- or spectrum-based regularisers. The main comparisons emphasise the structurally interpretable choice $\mathcal{L}_{\text{defect}}$.

*Table 29.* **Per-topology breakdown of Betti error rates.**[†] Finer-grained analysis showing how topology preservation varies with test-set complexity. Mean ± std over 5 seeds, averaged within each $\beta_0$ class. Training only on $\beta_0 \in \{0, 1\}$.

| Method | $\beta_0 = 1$ (in-dist) | $\beta_0 = 2$ (mild OOD) | $\beta_0 = 3$ (hard OOD) | $\beta_0 = 4$ (extreme OOD) |
|---|---|---|---|---|
| Unconstrained | 0.042 ± 0.014 | 0.158 ± 0.038 | 0.247 ± 0.051 | 0.319 ± 0.067 |
| Soft Defect ($\lambda = 1.0$) | 0.008 ± 0.006 | 0.061 ± 0.019 | 0.104 ± 0.027 | 0.148 ± 0.034 |
| Soft Defect ($\lambda = 10$) | 0.002 ± 0.003 | 0.021 ± 0.011 | 0.038 ± 0.016 | 0.061 ± 0.021 |
| Manifold | 0.018 ± 0.009 | 0.113 ± 0.031 | 0.184 ± 0.042 | 0.234 ± 0.053 |
| **Ours (Defect-controlled)** | **0.0004 ± 0.0006** | **0.0038 ± 0.0032** | **0.0084 ± 0.0057** | **0.0139 ± 0.0081** |

[†]Clear trend: Betti error increases with topology complexity for all methods, but the defect-controlled model maintains < 2% errors even at extreme $\beta_0 = 4$.

**Discussion** This benchmark is constructed to separate *apparent accuracy* from *structural correctness* under distribution shift. A method can achieve low PDE error on familiar geometries while silently violating the underlying

*Table 27.* **DarcyFlow 2D**, $\beta = 1.0$, $64 \times 64$ **(comprehensive comparison).**[†] Results show mean ± std over 5 random seeds. The proposed defect-controlled model achieves competitive accuracy while maintaining structural guarantees (low cochain-compatibility defect, near-zero topology errors). Training on 1024 samples, testing on 100 held-out instances.

| Method | nRMSE↓ | Time [s]↓ | $D_{\text{comp}}$[‡]↓ | Betti Err.[§]↓ |
|---|---|---|---|---|
| HAMLET | $(1.40 \pm 0.12) \times 10^{-2}$ | 324 ± 19 | – | – |
| OFormer | $(2.05 \pm 0.16) \times 10^{-2}$ | 247 ± 15 | – | – |
| **Ours (Defect-controlled)** | $\mathbf{(2.30 \pm 0.19) \times 10^{-2}}$ | **271 ± 17** | $(3.2 \pm 0.5) \times 10^{-3}$ | **0.008 ± 0.006** |
| U-Net | $(3.30 \pm 0.24) \times 10^{-2}$ | 198 ± 11 | – | – |
| DeepONet | $(5.12 \pm 0.38) \times 10^{-2}$ | 156 ± 8 | – | – |
| FNO | $(6.40 \pm 0.47) \times 10^{-2}$ | 179 ± 10 | – | – |
| GeoFNO | $(6.34 \pm 0.51) \times 10^{-2}$ | 183 ± 12 | – | – |
| MAgNet | $(1.03 \pm 0.09) \times 10^{-1}$ | 142 ± 9 | – | – |

[†]Training: 50 epochs, batch size 8. HAMLET/OFormer from published benchmarks; others reproduced on same hardware (RTX 4090).
[‡]Complex defect measurable only for methods exposing discrete operators; baselines use end-to-end learning.
[§]Betti error rate: fraction of test instances where spectral $\widehat{\beta}_0$ differs from ground-truth $\beta_0$.

chain-complex structure, which in turn can corrupt topology-sensitive behavior on new domains. The evaluation therefore tracks simultaneously (i) complex chain-condition compatibility via $\|D_1 D_0\|_F$, (ii) OOD PDE solution error under a forcing shift, and (iii) whether the learned operator preserves the ground-truth topology (Betti error rate).

**Key findings** Unconstrained learning exhibits large structural violations (defect 1.5–3.0) and nontrivial topology errors (10–20%). Soft defect regularization improves defect only modestly (to 0.30–2.00) and still leaves noticeable topology errors (3–15%), even though its OOD PDE error remains comparable to unconstrained training. Projection and manifold-style baselines can drive defect down to the 0.015–0.030 range (large reductions vs. unconstrained), but incur a clear accuracy penalty (PDE error factor up to 2×) and still retain nonzero Betti error. In contrast, the defect-controlled model achieves the intended regime: it maintains very low defect (0.008–0.020, i.e. ≥ 120× reduction vs. unconstrained), matches or improves unconstrained OOD accuracy (PDE error factor 0.90–1.00×), and drives topology errors to near-zero (0–1%). Together, these results demonstrate that the proposed constraint is not merely a regularizer, but a mechanism for failure prevention under topological distribution shift: it preserves discrete structure and topology without paying a meaningful OOD accuracy penalty.

*Table 28.* Topology-shift evaluation: comprehensive OOD evaluation[†]. Models trained on topologically simple meshes ($\beta_0 \in \{0, 1\}$, 256 training geometries) and evaluated out-of-distribution on harder topologies ($\beta_0 \in \{2, 3, 4\}$, 128 test geometries with 2–4 holes plus skinny-channel regime) under shifted forcing distribution ($\sigma_{\text{train}} = 1.0 \rightarrow \sigma_{\text{test}} = 2.5$). Results show mean $\pm$ std over 5 random seeds. All metrics averaged across OOD test set.

| Method | Mean Defect[‡] ($\|D_1 D_0\|_F$) ↓ | Mean PDE Err.[§] (nRMSE) ↓ | Defect Reduct.[‖] (vs. Unconst.) ↑ | PDE Err. Factor[‖] (vs. Unconst.) ↓ | Betti Err.[††] (Rate) ↓ | Solve Fail[‡‡] (Rate) ↓ | Mean Cond. ($L_0$) ↓ |
|---|---|---|---|---|---|---|---|
| Unconstrained | $2.18 \pm 0.47$ | $0.763 \pm 0.082$ | $1.0\times$ | $1.0\times$ | $0.147 \pm 0.031$ | $0.034 \pm 0.012$ | $(3.84 \pm 1.21) \times 10^3$ |
| Soft Defect ($\lambda = 0.1$) | $1.32 \pm 0.38$ | $0.741 \pm 0.076$ | $1.65\times$ | $0.971\times$ | $0.094 \pm 0.024$ | $0.019 \pm 0.008$ | $(1.97 \pm 0.63) \times 10^3$ |
| Soft Defect ($\lambda = 1.0$) | $0.583 \pm 0.192$ | $0.728 \pm 0.071$ | $3.74\times$ | $0.954\times$ | $0.057 \pm 0.018$ | $0.008 \pm 0.005$ | $(8.42 \pm 2.14) \times 10^2$ |
| Soft Defect ($\lambda = 10$) | $0.178 \pm 0.067$ | $0.734 \pm 0.074$ | $12.2\times$ | $0.962\times$ | $0.021 \pm 0.011$ | $0.002 \pm 0.003$ | $(2.91 \pm 0.74) \times 10^2$ |
| Manifold | $0.0218 \pm 0.0047$ | $0.832 \pm 0.094$ | $100\times$ | $1.090\times$ | $0.112 \pm 0.028$ | $0.003 \pm 0.003$ | $(2.73 \pm 0.51) \times 10^2$ |
| Projection Baseline | $0.0224 \pm 0.0051$ | $1.186 \pm 0.217$ | $97.3\times$ | $1.554\times$ | $0.089 \pm 0.023$ | $0.006 \pm 0.005$ | $(2.68 \pm 0.48) \times 10^2$ |
| **Ours (Defect-controlled)** | $\mathbf{0.0134 \pm 0.0031}$ | $\mathbf{0.731 \pm 0.075}$ | $\mathbf{163\times}$ | $\mathbf{0.958\times}$ | $\mathbf{0.0047 \pm 0.0041}$ | $\mathbf{0.0008 \pm 0.0012}$ | $\mathbf{(2.59 \pm 0.44) \times 10^2}$ |

[†]Training: 200 epochs, Adam optimizer (lr $10^{-3}$), batch size 16. All methods use the same compute budget.

[‡]Raw Frobenius norm of chain-condition violation; lower indicates better structural preservation.

[§]Normalized RMSE on OOD test set under topology and forcing shift.

[‖]Relative to unconstrained baseline; defect reduction $> 1$ is better, PDE error factor $< 1$ is better.

[††]Betti error rate: fraction where spectral $\widehat{\beta}_0$ doesn't match ground-truth $\beta_0(K_h)$.

[‡‡]Solve failure rate: fraction where SPD solve diverges or produces NaN/Inf (condition $> 10^5$ or CG non-convergence).

# N. Additional Experimental Results

*Table 30.* Summary of core structural experiments and the theoretical statements they illustrate. Full protocols and ablations are provided in the appendix.

| Experiment | Theoretical reference | Main observation |
|---|---|---|
| Operator vs. defect | Thm. 4.4 | Operator deviation follows a linear–quadratic defect regime |
| PDE error decomposition | Thm. 5.6 | Defect contribution is negligible in the small-defect regime |
| Learning dynamics | Thm. 5.6 | Explicit defect control limits structural drift |
| Mesh convergence | Thm. 5.6 | Decaying defects preserve FEEC convergence slopes |
| Spectral Betti stability | Thm. 4.10 | Betti count remains correct below the defect threshold |
| Discretization OOD (unstructured) | Thm. 5.6 | OOD mesh shifts show improved errors with small structure metrics |

