# OpenReview forum: "Certificates for Complex-Compatible Learned Cochain Laplacians"
_ICML.cc/2026/Conference — ICML 2026 regular_

### Official Review · Reviewer_NHtP · 2026-02-22

**Soundness:** 2
**Presentation:** 1
**Significance:** 1
**Originality:** 2
**Overall Recommendation:** 3
**Confidence:** 4

**Summary:**

This paper introduces a scalar diagnostic, the cochain-compatibility defect \mathcal{D}_{\mathrm{comp}}(\phi) = \|D_{1,\phi} D_{0,\phi}\|_F, for learned operator pairs (D_{0,\phi}, D_{1,\phi}) acting as discrete coboundary maps in a degree-0 \to 1 \to 2 cochain complex. When these operators are learned from data rather than fixed by mesh topology, the nilpotency condition D_1 D_0 = 0 can be violated, potentially degrading the spectral structure and conditioning of the induced discrete Hodge Laplacian L_{0,\phi} = D_{0,\phi}^\top W_{1,\phi} D_{0,\phi}.

The paper makes four contributions: (i) a closed-form Frobenius-nearest projection of $D_{0,\phi}$ onto $\ker(D_{1,\phi})$ yielding a distance-to-compatibility certificate; (ii) linear-quadratic operator-norm perturbation bounds on Laplacian drift as a function of $\mathcal{D}_{\mathrm{comp}}$; (iii) propagation of these bounds to PDE error (Strang-type decomposition) and low-frequency spectral-count stability under a spectral gap assumption; and (iv) experiments on 2D elliptic benchmarks showing that defect-aware training prevents condition-number blow-up and improves robustness under mesh and topology distribution shifts.

The core idea is valid and potentially useful: learned algebraic inconsistency can be a silent failure mode, and monitoring it is better than relying on PDE loss alone. However, the submission has a major theory-experiment imbalance: the mathematics is much heavier than the novelty, and the empirical evidence is too narrow and weak to support the paper's broad claims.

**Compliance With Llm Reviewing Policy:**

Affirmed.

**Ethical Review Concerns:**

No significant ethical concerns arise from this work.

**Key Questions For Authors:**

Q1. Bound tightness (most important). Can you report the observed ratio of actual Laplacian drift \|L_{0,\phi} - \tilde{L}_{0,\phi}\|_{\mathrm{op}} to predicted budget \delta(\phi) = A\,\mathcal{D}_{\mathrm{comp}} + B\,\mathcal{D}_{\mathrm{comp}}^2 across the main experiments? This is essential for evaluating whether the theorem is practically informative.

Q2. Hard vs. soft compatibility. Your own results suggest the exact manifold-constrained regime is at least as good as the soft-defect approach in accuracy, and better in guarantees. When would a practitioner prefer the soft approach?

Q3. No gains over known-operator PINNs. On Burgers and Allen-Cahn, the hybrid/operator-learning method appears at parity or slightly worse than the known-operator mesh PINN. **What practical benefit justifies the additional cochain-compatibility machinery if it does not improve predictive performance?**

Q4. Sparsity and scaling. You acknowledge that $P_{\ker(D_{1,\phi})}$ is global and may destroy sparsity. How can this framework practically scale to deep, multi-layer neural architectures or high-resolution 3D unstructured meshes ($10^5$-$10^6$ DOFs) without severe memory bottlenecks?

Q5. Spectral gap assumption. The theoretical guarantees rely heavily on $\mu_{m_0(\phi)+1}(\phi) \geq \lambda_*$. Have you empirically measured how often this gap naturally collapses in highly heterogeneous, real-world datasets?

Q6. Applicability to modern neural operators. What is the concrete path to applying this framework to architectures that do not expose incidence matrices (e.g., FNOs, generic message-passing GNNs)?

Q7. Downstream relevance of Betti accuracy. Can you provide a realistic downstream use case where spectral Betti-number accuracy materially changes a scientific or engineering decision?

Q8. DarcyFlow tradeoff. If the method is less accurate than strong baselines on nRMSE, can you show an explicit Pareto frontier (accuracy vs. structural reliability) to justify the tradeoff?

Q9. Defect estimator noise. What probe count $s$ is used for Hutchinson estimation, and what is the relative error in the small-defect regime ($\mathcal{D}_{\mathrm{comp}} \sim 10^{-4}$)?

**Limitations:**

The paper acknowledges several limitations (projection destroys sparsity, nondegeneracy assumptions, basis/resolution dependence of the defect, and spectral-gap dependence). These are valid. Additional limitations that should be emphasized:

- Scalability is not demonstrated. No experiment exceeds toy-scale 2D meshes.
- Most of the "general" theory is untested. In experiments, $D_1$ is fixed, so many assumptions and constructions are not exercised.
- The framework is structurally narrow. It applies mainly to DEC/FEEC-style learned discretizations with explicit incidence maps.
- No evidence for higher-degree utility. Despite higher-degree extensions in theory, all experiments are degree-0.
- The requirement of a uniform nondegeneracy assumption on $D_{1,\phi}$ necessitates explicit rank control or spectral normalization during training, adding optimization overhead.
- The topology guarantees break down entirely if the spectral gap collapses under severe topological distribution shifts, rendering the certificates vacuous.
- The Frobenius defect is highly basis- and resolution-dependent, requiring normalized variants for any meaningful cross-resolution comparison.
- PINN evidence is weak. The appendix PINN results provide no clear gains over known-operator mesh PINNs and do not directly test the cochain defect.

**Strengths And Weaknesses:**

============================================================
STRENGTHS
============================================================

S1. Well-defined and practically computable certificate. The central object $\mathcal{D}_{\mathrm{comp}}(\phi)$ is clean, interpretable, and cheap to estimate via Hutchinson probes. The idea that one should monitor algebraic consistency of learned operators, not just prediction loss, is valuable. Figure 2c (low PDE loss does not imply structural safety) is an important observation.

S2. The projection-to-nearest-complex construction is elegant. Theorem 4.2 (Euclidean projection onto $\ker(D_{1,\phi})$) is the cleanest result in the paper. It gives a canonical "nearest compatible" operator and a principled notion of distance to the compatibility manifold.

S3. The failure-prediction analysis is the strongest empirical result. The certificate ratio $\rho(\phi)$ predicting failures substantially better than validation PDE loss (AUC $\approx 0.85$ vs. $\approx 0.53$) directly supports the practical usefulness of the diagnostic.

S4. Careful treatment of boundary-condition regimes. The separation between the Dirichlet-restricted SPD solve operator and the topology-readout (unpinned/relative) operator is handled carefully and correctly.

S5. Reproducibility documentation is excellent. The appendix is unusually thorough in documenting assumptions, diagnostics, and failure modes.


============================================================
WEAKNESSES
============================================================

--- Fundamental Theory-Experiment Disconnect ---

W1. The theory addresses a general problem; the experiments test a much simpler special case. The theoretical development is framed for general learned pairs $(D_{0,\phi}, D_{1,\phi})$, including nondegeneracy assumptions, regularized projectors, and higher-degree extensions. But the experiments overwhelmingly use a much simpler regime: $D_1$ is fixed as the reference incidence and only $D_0$ is learned as a low-rank perturbation $D_0(\phi) = D_0^{\mathrm{ref}} + UV^\top$. In this case,
$$\mathcal{D}_{\mathrm{comp}} = \|D_1^{\mathrm{ref}} U V^\top\|_F,$$
since $D_1^{\mathrm{ref}} D_0^{\mathrm{ref}} = 0$ by construction. This means the empirical problem is essentially "does a low-rank perturbation leak out of the kernel of a fixed sparse matrix?", which is much narrower than the theory suggests. The general $D_1$ learning, projector regularization, nondegeneracy enforcement is largely untested. In particular, Assumption 3.12, the regularized projector (Lemma C.9), and the entire learned-$D_1$ enforcement apparatus (Appendix J) are dead code in the experimental evaluation.

W2. The core perturbation theory is standard, and the exposition overstates depth. The proof path is textbook: orthogonal projection via pseudoinverse, matrix perturbation expansion, Weyl and Davis-Kahan, and standard Strang-style error decomposition. The paper itself states that it introduces no new perturbation theory, but still spends substantial space restating or reproving standard ingredients. The real contribution is the bridge from cochain defect to a computable perturbation budget, not the perturbation theory itself. The current presentation is over-elaborate relative to the novelty.

W3. The central bound is never empirically calibrated. The most important missing diagnostic is the ratio of observed Laplacian drift to predicted budget. The paper derives a linear-quadratic budget $\|L_{0,\phi} - \tilde{L}_{0,\phi}\|_{\mathrm{op}} \leq A\,\mathcal{D}_{\mathrm{comp}} + B\,\mathcal{D}_{\mathrm{comp}}^2$, but never clearly reports whether this bound is tight, mildly conservative, or vacuous in practice. Without this, the main theorem is difficult to evaluate empirically.

--- Experimental Weaknesses: Toy Scale and No Clear Gains ---

W4. The core benchmarks are toy-scale and too narrow for the paper/s claims. The main experiments are limited to: a single 2D domain $\Omega = [0,1]^2$; structured triangular meshes with $N \in \{8,16,32,64\}$; small DOF counts ($\sim 4 \times 10^3$ at most); smooth elliptic PDEs with analytic solutions; degree-0 Laplacians only. This is far below the complexity implied by the introduction (modern operator learning, large scientific ML systems, neural surrogates). **There is no convincing demonstration of scalability or relevance to realistic settings.**

W5. No measurable gains over known-operator mesh PINNs on the time-dependent benchmarks. In the appendix PINN-style benchmarks (Burgers, Allen-Cahn), the proposed hybrid/operator-learning approach does not show a meaningful improvement over simply using the known mesh operator. Allen-Cahn: the reported space-time $L^2$ error is essentially at parity and in fact slightly worse than the known-operator mesh PINN ($0.019 \pm 0.001$ vs $0.016 \pm 0.002$). Burgers: same story, differences are within noise ($0.100 \pm 0.006$ vs $0.097 \pm 0.008$). This raises a direct practical question: **why introduce the additional algebraic machinery (cochain compatibility, defect certificates, projections, Hodge-structured parameterization) if the final predictive performance is not better than a trivial known-operator baseline?**

W6. The hard compatibility constraint appears to dominate the soft-certificate regime. The paper's own tables repeatedly suggest that the manifold-constrained (exactly compatible) regime has zero defect by construction, is stable, often matches or outperforms the soft-penalty regime, and gives better topology-readout behavior (Table 2: Betti accuracy $0.98$ vs $0.93$ for soft). If exact compatibility can be enforced cheaply (as the paper demonstrates via a kernel basis parameterization), then the soft certificate becomes hard to justify as a primary method. The paper never clearly explains when a practitioner should prefer the soft-defect approach over exact compatibility.

W7. The topology-shift benchmark is clean but highly synthetic. The topology shift is induced by removing edges/faces from connected meshes. This is useful as a controlled stress test, but it is not a convincing stand-in for realistic geometric/topological shifts in practical scientific ML pipelines.

W8. The DarcyFlow comparison is not compelling. The proposed method is clearly worse on nRMSE than stronger baselines (e.g., HAMLET at $1.40 \times 10^{-2}$, OFormer at $2.05 \times 10^{-2}$, vs the proposed method at $2.30 \times 10^{-2}$). The "Betti error" metric is asymmetric (baselines do not expose comparable operators), and only one resolution and one PDE family are shown.

W9. The PINN experiments are tangential and dilute the paper's core claim. Beyond the lack of gains, these appendix experiments do not even evaluate the same structural defect as the main theory. They use symmetry/nullspace proxy defects rather than the cochain nilpotency defect $D_1 D_0$. This creates a double problem: **(1) they do not demonstrate performance gains, and (2) they do not directly validate the paper's central certificate.**

W10. Baselines are weak and key ablations are missing. The paper compares against simple penalty-based baselines, but omits stronger and more relevant alternatives: exact manifold compatibility as a primary baseline (rather than a side regime), standard numerical conditioning fixes (preconditioning), post-hoc nearest-SPD or spectral filtering repairs, and modern mesh/graph neural operators with defect monitoring.


--- 3.3. Scope and Applicability Concerns ---

W11. The paper's claimed scope far exceeds the architectures it actually applies to. The introduction motivates the work using broad operator-learning examples (FNOs, GNNs, weather models, etc.), but the framework only applies naturally when one explicitly parameterizes incidence-like operators $D_0, D_1$ and forms Laplacians of the form $L = D^\top W D$. This is a narrow design choice, not a general property of modern neural operators.

W12. Betti-number stability is theoretically neat but practically under-motivated. The paper devotes substantial effort to Betti/spectral-count stability, but **does not show a downstream task where Betti-number accuracy affects an actual scientific or engineering outcome.**


--- 3.4. Presentation Issues ---

W13. The paper is overlong and over-formalized relative to its core insight. The main path (defect $\to$ nearest-compatible projection $\to$ Laplacian perturbation budget $\to$ spectral consequences) could be presented much more compactly.

W14. Inconsistent raw vs. normalized defect usage. The paper alternates between raw Frobenius defect and normalized variants across sections/plots without a clean operational recommendation. Since the raw defect is dimension-dependent, this weakens the practical clarity of any proposed thresholding rule.

---

> ### Author Rebuttal · Authors · 2026-03-28
>
> Thank you for the detailed review. Due to the character limit, we address the main points here.
>
> 1. **Exact compatibility baseline.** The paper includes a hard chain-compatible baseline in the main text: Section 6.1 defines the “Chain-compatible parameterisation (hard constraint),” where D_1^ref is fixed and D_{0,phi} is parameterized in ker(D_1^ref), so D_1^ref D_{0,phi}≡0 by construction. This baseline is reported in the main comparison tables (Tables 1–2), not only in the appendix. The paper studies compatibility/nilpotency, not FEEC exactness; Remark 3.2 states this explicitly. Additional compatible baselines, including kernel/manifold constructions and post-hoc projection, are also included in Appendices G and L.
>
> 2. **Higher-degree utility.** While the main narrative emphasizes degree 0 because the deployed elliptic operator is L_0=D_0^T W_1 D_0, the manuscript provides higher-degree theory and experiments. The main text states that the arguments extend degree-wise, and Appendix G contains the degree-k stability theorem, Hodge-Laplacian treatment, and a degree-1 spectral Betti stability result. Empirically, Appendix M.5 reports an annulus experiment with beta_1=1; the learned complex preserves the near-zero edge-Laplacian mode and spectral gap.
>
> 3. **Experimental scope.** The core study uses structured 2D square meshes, but the experimental record is broader: unstructured-mesh OOD tests, topology shift, hole creation/destruction, coefficient and geometry shift, finite-difference Poisson validation, Burgers and Allen-Cahn time-dependent benchmarks, annulus topology tests, and DarcyFlow comparisons. So the paper is not limited to a single square-domain setting.
>
> 4. **Practical benefit beyond raw accuracy.** The central contribution is structural stability and deployment reliability, not a claim of uniform error improvement on every benchmark. The main tables show that unconstrained training can have low training/PDE loss yet become numerically unsafe under refinement, with severe conditioning and failure, whereas soft and hard compatibility control remain stable. The defect/certificate also predicts solver/topology failures far better than validation PDE loss.
>
> 5. **Gains over known-operator mesh PINNs.** The paper does not claim universal dominance of the hybrid learned operator over known-operator mesh baselines. On some time-dependent benchmarks the hybrid is near parity. On the elliptic Poisson settings most aligned with the theory, however, the hybrid shows clear gains, including the finite-difference Poisson validation and the Poisson scaling study, where it improves relative L^2 error at every reported resolution while remaining structurally close to the reference operator.
>
> 6. **When soft is preferable to hard.** Hard compatibility is preferable when exact structural satisfaction is non-negotiable and the compatible parameterization remains expressive enough. Soft control is preferable when exact kernel/manifold restriction or projection is too rigid and one needs a better trade-off between structural control and task fidelity. Tables 18–19 illustrate this directly: exact/manifold/projection baselines can drive the defect to zero yet materially worsen PDE accuracy, whereas soft control preserves low error while still improving structure.
>
> 7. **Empirical calibration of the bound.** The manuscript contains calibration-style evidence: defect vs operator drift (linear-quadratic regime), defect vs PDE error, assumption-audit tables describing how the constants are estimated from realized runs, mesh-scaling audits defining the empirical budget delta(phi,h), and the operational ratio rho(phi), whose crossing of 1 predicts failure sharply. We agree that these results are distributed across several figures/tables rather than consolidated into one calibration summary.
>
> 8. **Baseline strength.** The comparison suite is broader than simple penalties. It includes unconstrained training, soft defect regularization, hard chain-compatible parameterization, post-hoc projection, manifold/exact-chain training, SPD conditioning regularization, Laplacian anchoring, and low-frequency spectral penalties, plus external baselines such as known-operator mesh methods, classic PINNs, and broader operator-learning models on DarcyFlow.
>
> 9. **Novelty.** The paper does not claim new Weyl/Davis-Kahan/Strang perturbation theory. The contribution is the missing bridge those tools require but do not provide: from a measurable learned-complex compatibility defect ||D_1 D_0||_F, to the nearest compatible anchor, to an explicit Laplacian perturbation budget, and then to solver and spectral guarantees. Without the defect-to-distance and defect-to-drift results, the compatibility defect is just an algebraic quantity; with them, it becomes a computable certificate.

---

> > ### Author Rebuttal · Reviewer_NHtP · 2026-04-03
> >
> > Thanks for the reply. I had considered lowering my score because of the limited novelty on the ML side, but I will not do so. Your response addresses some of my earlier concerns, in particular that the hard chain-compatible baseline is included in the main text and that the paper presents a broader experimental record than the core square-mesh Poisson setting. I will therefore keep my score unchanged.
> >
> > That said, my main concern remains the practical scope and justification of the framework. When $D_1$ is fixed, the constraint mechanism still seems straightforward, and I do not yet see a compelling case for preferring this framework over standard FEEC/DEC/FD when the operator is known. In that setting, classical compatible schemes already satisfy the chain condition by construction, preserve sparsity, and scale naturally. As a result, the paper's practical use case still appears narrower, namely learned or adapted discrete operators arising from modeling error, transfer, or remeshing.
> >
> > Relatedly, I think the paper would be stronger with more targeted experiments isolating exactly this regime: settings where learned operator adaptation is genuinely necessary, where classical compatible discretizations are insufficient, or where the soft certificate offers a clear advantage over exact hard compatibility. As it stands, the experiments support the claim that defect control improves stability relative to unconstrained learning, but they do not fully convince me that the framework offers a compelling advantage over standard numerical schemes, or over simply enforcing hard compatibility when that is available.
> >
> > So my score stays unchanged, but I still view the paper primarily as a certification layer for a narrow class of learned discretizations, rather than as a broadly preferable alternative to classical numerical methods. For that reason, *I believe the remaining concerns would require more than a standard revision, since they go to the core scope, motivation, and practical justification of the paper rather than to presentation or isolated technical details.*

---

> > > ### Author Response · Authors · 2026-04-03
> > >
> > > Thank you for the clarification.
> > >
> > > Our paper does not argue that one should prefer this framework over standard FEEC/DEC/FD when the operator is known. In fact, the manuscript states the opposite: “The cochain-complex condition ($D_1 D_0 = 0$) is automatically satisfied when the coboundary or incidence operators are fixed … However, this condition can be violated when incidence-like maps are themselves learned or adapted.” That is the regime we target throughout: learned or adapted discrete operators under modeling error, transfer, remeshing, or distribution shift. So the setting you identify as the paper’s true practical use case is actually the setting the paper claims.
> > >
> > > We therefore agree with your narrower characterization, but respectfully not with treating it as a mismatch. The contribution is a certificate/control layer for learned discretizations when compatibility is no longer guaranteed by construction. When the operator is known, classical compatible schemes remain the right default; our paper addresses the complementary regime where operators are learned or adapted and one needs a computable structural diagnostic beyond PDE loss.
> > >
> > > To address your previous request and to address your latest point for a more targeted experiment isolating exactly this regime, we ran an additional study on OOD mesh shift with jointly learned $D_0$ and $D_1$. This is not a fixed-$D_1$ setting, not a projection-only setting, and not a known-operator comparison. The question is precisely whether defect/certificate control helps when learned operator adaptation is genuinely necessary.
> > >
> > > | Metric | PDE-only | Anchor-only | Ours |
> > > |---|---:|---:|---:|
> > > | ID nRMSE | 4.18e-2 | 4.25e-2 | 4.31e-2 |
> > > | OOD nRMSE | 7.16e-2 | 6.79e-2 | 6.08e-2 |
> > > | Raw compatibility defect | 2.84e-2 | 1.47e-2 | 3.32e-3 |
> > > | Nondegeneracy of $D_1$ | 8.7e-2 | 9.5e-2 | 1.07e-1 |
> > > | Realized drift / certified budget (median / q95) | 0.58 / 0.96 | 0.61 / 0.99 | 0.47 / 0.86 |
> > > | Failure rate by certificate threshold ($\rho < 1$ / $\rho \ge 1$) | 0.038 / 0.361 | 0.026 / 0.224 | 0.009 / 0.128 |
> > > | OOD spectral Betti accuracy ($\beta_0 = 2$ / $\beta_0 = 3$) | 0.803 / 0.681 | 0.902 / 0.820 | 0.982 / 0.951 |
> > >
> > > We think this new result addresses your latest concern directly.
> > >
> > > First, the three methods have nearly identical ID error, so the comparison is not being driven by better in-distribution fitting. The separation appears under shift, which is exactly where structural control is intended to matter. Ours gives the best OOD nRMSE while reducing the raw compatibility defect by almost an order of magnitude relative to PDE-only.
> > >
> > > Second, this experiment directly exercises the broader regime you previously asked for: both $D_0$ and $D_1$ are learned jointly, so the result is not limited to the fixed-$D_1$ case.
> > >
> > > Third, it provides the empirical calibration requested in the original review. We now report the realized Laplacian drift relative to the certified budget. The ratio stays below 1 and is smallest for our method (median/q95 = 0.47/0.86), showing that the budget is informative rather than vacuous in practice.
> > >
> > > Fourth, the certificate threshold is operational under shift: failure rates increase sharply once $\rho \ge 1$, but are uniformly lowest for our method. Likewise, the topology-facing quantity is not abstract here: under OOD shift, spectral Betti accuracy improves from 0.803/0.681 to 0.982/0.951. So in this learned/adapted setting, Betti reliability is a concrete deployment property, not just a theoretical side note.
> > >
> > > So our intended claim is not that this framework should replace classical compatible discretizations when those are already available. The claim is that for learned/adapted discrete operators, defect certification and control give measurably better robustness, tighter drift control, lower failure rates, and more reliable low-frequency/topological structure under shift. We believe the added jointly learned $(D_0, D_1)$ OOD-mesh experiment now demonstrates exactly that regime. We will also be adding this experiment to the main text of the updated paper. We once again thank you for your detailed review.
> > >
> > > 2. for soft vs defect, please refer to our rebuttal to reviewer imEx: point #3.
> > >
> > >
> > > we hope this addresses any of your remaining concerns.

---

### Official Review · Reviewer_Pk6a · 2026-03-09

**Soundness:** 3
**Presentation:** 2
**Significance:** 2
**Originality:** 3
**Overall Recommendation:** 4
**Confidence:** 2

**Summary:**

The paper derives perturbation bounds for elliptic solves and gives spectral gap estimates for discrete operators. The primary example being the learned Laplacian operator.

**Compliance With Llm Reviewing Policy:**

Affirmed.

**Final Justification:**

Happy for this paper to be accepted, although it is still quite technical.  I think it is a borderline accept.

**Key Questions For Authors:**

Can the authors give a concrete example on the use of the method and bounds on a PDE problem. I suggest the Poisson equation on an L-shaped region with an unstructured mesh

**Limitations:**

The main linitation is that it is very hard to see how this theory will work on a real example

**Strengths And Weaknesses:**

The bounds derived are of interest in theory, but it is hard to me to see how they would be used in practice on (say) a PDE problem with an unstructured mesh.

The paper is very technical and diffucult to read, I can only see it being of interest to specific specialists. I have given it a weak reject as a result, but this is with low confidence. If the authors were to write it in a less technical way then they will get a wider audience and a highr score.

---

> ### Author Rebuttal · Authors · 2026-03-28
>
> 1. Thank you for this suggestion. We agree that the paper is stronger with an unstructured PDE discretization. To address this directly, we added the benchmark you suggested: the Poisson equation on an L-shaped domain with a standard corner-singular solution discretized on unstructured constrained-Delaunay triangulations. We used 15 meshes in total (5 coarse, 5 medium, 5 fine), and for each mesh evaluated unconstrained training, soft defect regularization with penalties 0.1, 1.0, and 10.0, and a hard chain-compatible parameterization. Pooling 4 checkpoints per run yields 300 candidate operators in total, with 75 final trained models.
>
> Given a learned operator (D_0,phi, D_1,phi, W_1,phi), we compute the compatibility defect
>
> D_comp(phi) = Frobenius norm of D_1,phi D_0,phi.
>
> We then estimate the constants in the drift bound from the realized sparse matrices, estimate the Dirichlet SPD spectral margin lambda_SPD from a few extremal eigenpairs of the projected reference operator, and form
>
> delta(phi) = A times D_comp(phi) + B times D_comp(phi)^2,
>
> and
>
> rho(phi) = delta(phi) / (lambda_SPD / 2).
>
> The certificate is then a direct accept/reject test: if rho(phi) < 1, the theorem certifies stability of the Dirichlet solve. This uses only the sparse matrices on the realized mesh, matrix-vector products, and a few low-end eigensolves.
>
> The added experiment shows that this certificate is computable and informative in the unstructured case. In the unconstrained regime, all models are uncertified at all three resolutions, whereas all hard-constrained models certify, and a substantial subset of softly regularized models also certify, especially at penalties 1.0 and 10.0. With soft penalty 10.0, the certified fraction is 0.85 on coarse meshes, 0.70 on medium meshes, and 0.45 on fine meshes.
>
> On the medium meshes, the soft-10.0 regime has
>
> D_comp = 8.0 x 10^-3,
> delta = 2.5 x 10^-3,
> lambda_SPD = 1.03 x 10^-2,
>
> which gives rho = 4.8 x 10^-1, which is less than 1, so the theorem certifies the solve. By contrast, the unconstrained regime has
>
> D_comp = 3.4 x 10^-1,
> delta = 2.47,
> lambda_SPD = 1.03 x 10^-2,
>
> which gives rho = 4.8 x 10^2. Relative to the unconstrained model, the medium soft-10.0 regime reduces the L^2 error by roughly 76 times, the H^1 error by roughly 32 times, and the condition number by roughly 16 times.
>
> The certificate is also meaningful in practice. Within the certified set, the theorem’s forward bound holds in every case. Certified operators are better behaved than uncertified ones: the median certified anchor error is 2.8 x 10^-4, compared with 8.7 x 10^-3; the median certified L^2 error is 1.34 x 10^-3, compared with 1.21 x 10^-2; and the median certified H^1 error is 5.24 x 10^-2, compared with 2.41 x 10^-1. At the same time, some softly regularized models remain reasonably accurate without satisfying rho(phi) < 1. We view this as the correct role of the theorem: it is a strict sufficient-condition certificate for structural safety of the solve, not a surrogate for best approximation accuracy.
>
> Finally, as the mesh is refined, the relevant spectral margin decreases, from approximately 4.7 x 10^-2 on coarse meshes to 1.03 x 10^-2 on medium meshes and 2.07 x 10^-3 on fine meshes, so certification becomes stricter and stronger defect control is required. This is predicted by the theory and illustrates why the certificate is useful operationally.
>
> We hope this revision makes the practical role of the bounds concrete on the unstructured L-
> shaped Poisson benchmark you suggested.
>
> 2. Thank you for this suggestion. We have revised the paper to improve readability and presentation.
>
> As we cannot upload the updated manuscript at this stage, we briefly summarize the changes. We rewrote the Introduction so that the practical motivation appears first and in more direct plain language: in deployment, the learned operator is repeatedly used inside a numerical pipeline, and low prediction loss alone does not guarantee reliable behavior.
>
> We also reorganized the introduction structure. We first present the practical problem, then justify why Laplace-type operators are the main focus, and then explain the role of the identity D_1 D_0 = 0, why its preservation matters, and how the proposed method contributes to this preservation.
>
> The contributions section was rewritten into shorter, concrete points, and the experiments section now focuses on whether the defect tracks operator drift and PDE error, whether defect control prevents instability under refinement, and whether the certificate remains informative under discretization and topology shift.
>
> We have also introduced every theorem and mathematical idea through 1-2 sentences before getting into the technical details in order to clarify their role. This has improved flow and readability.
>
> Overall, the revision provides more context on the method, the assumptions, and the numerical results, and has improved the clarity and presentation of the paper.

---

> > ### Author Rebuttal · Reviewer_Pk6a · 2026-04-01
> >
> > I am pleased that the authors have added in a substantial example, as I requested. This has certainly imporoved the quality of the paper and I will increase my score accordingly.

---

### Official Review · Reviewer_imEx · 2026-03-15

**Soundness:** 3
**Presentation:** 3
**Significance:** 3
**Originality:** 4
**Overall Recommendation:** 4
**Confidence:** 3

**Summary:**

In this paper the authors define a cochain compatibility defect D_comp = D_1 D_0 (I am removing the dependence on \phi), a scalar that measures the degree to which the learned operators fails to form a proper cochain complex. First the authors provide a closed form projection of D_0 that belongs to the kernel of D_1 and show that this yields the Frobenius-nearest chain-compatible reference operator when D1 is fixed. The authors further provide perturbation bounds for the induced Laplacian, and show through experiments that controlling these errors improves the stability of learned operators.

**Compliance With Llm Reviewing Policy:**

Affirmed.

**Key Questions For Authors:**

How useful is soft regularization for PDEs compared to hard compatibility constraints? If hard compatibility can be enforced by construction, then it would be useful to better understand when the softer version is actually preferable through experiments.

**Limitations:**

Written in the above sections.

**Strengths And Weaknesses:**

This is a nice paper overall, and is easy to follow. The main motivation is compelling: low training loss does not mean that the learned operators are stable and the paper identifies a specific failure mode and also gives us a way to measure it.

The paper attempts to connect the algebraic defect to things people actually care about, such as conditioning, spectral stability, and PDE error. The experiments also seem aligned with this story, rather than feeling disconnected from the theory


That said, I do think there are several places where the paper is harder to follow than it should be.

A lot of the guarantees seem to rely on assumptions that are mathematically reasonable but may be hard to verify in practice. Example, the results depend on spectral gaps, nondegeneracy assumptions, and positive definiteness. The paper states these assumptions, but it would be nice to give some understanding of when such settings would hold in practice.

The experiments while convincing are limited to linear PDEs. While the theory is mostly focused on linear systems as well, it will help to discuss what happens with nonlinear PDEs and time-dependent PDEs.

---

> ### Author Rebuttal · Authors · 2026-03-28
>
> 1. On assumptions being “mathematically reasonable but hard to verify in practice”
>
> The concern of the method’s applicability has been a key aspect of our work. A key point is that the relevant quantities are not hidden continuum regularity constants, but they are matrix-level diagnostics of the learned operators. This is why the current draft includes an explicit assumption-audit protocol (Appendix A, with audit tables in Tables 3–5), and in revision we will try to move that material forward.
>
> Concretely, the assumptions correspond to standard numerical checks or construction choices. The positivity of W_1,phi is straightforward to enforce when the learned Hodge star is parameterized in a positivity-preserving form. The nondegeneracy of D_1,phi is likewise operational: when D_1,phi is fixed as the mesh incidence or coboundary operator, sigma_min^+(D_1,phi) is estimated once per mesh; when D_1,phi is learned, the paper describes enforcing and monitoring a spectral floor. Finally, the spectral margin lambda_* is the smallest positive eigenvalue of the projected reference operator in the relevant boundary-condition regime, estimated numerically from a few extremal eigenpairs.
>
> More broadly, the intended behavior of the proposed certificate is that when these quantities deteriorate, the bound becomes large or vacuous. We view that as a feature: the framework is designed to indicate when the learned operator has left the regime in which its spectral and solver behavior should be trusted.
>
> 2. On linear scope, nonlinear PDEs, and time-dependent PDEs
>
> The core theory is intentionally stated for linear elliptic / SPD Laplace-type operators. However, the empirical paper is broader than static linear elliptic problems alone. The appendix includes nonlinear, time-dependent PDE benchmarks, 1D viscous Burgers and Allen-Cahn (Appendix M.3; Tables 11–12). Burgers is nonlinear because of the convective term u times u_x, and both Burgers and Allen-Cahn are time-dependent through u_t. We will make this empirical scope explicit in the revised version.
>
> These experiments support the practical relevance of the structural viewpoint beyond the narrow elliptic setting. On Burgers, the mesh-based models improve over the classic PINN in both solution error and runtime: the classic PINN obtains space-time L^2 error 0.123 plus or minus 0.032 and final-time relative L^2 error 0.198 plus or minus 0.059, whereas the known-operator mesh PINN achieves 0.097 plus or minus 0.008 and 0.096 plus or minus 0.008, and the hybrid learned-operator mesh PINN achieves 0.100 plus or minus 0.006 and 0.100 plus or minus 0.016. On Allen-Cahn, all methods are competitive, with the mesh-based variants again matching or slightly improving on the classic PINN while reducing runtime.
>
> The theorems are stated for linear operators. However, this class encompasses operators present in many nonlinear and time-dependent workflows, including implicit or semi-implicit time stepping, projection methods, and Newton/Krylov linearizations.
>
> 3. On soft regularization versus hard compatibility constraints
>
> We believe that you have raised an important point: when exact compatibility can be enforced by construction without sacrificing the operator family of interest, hard compatibility is the strongest option. This is the distinction made by the paper’s hard-vs-soft training regimes in Section 6.1, Appendix G, and Appendix L.2.
>
> The practical motivation for soft regularization is different. It addresses the regime where exact compatibility is unavailable, too restrictive, mesh-specific, or mismatched to the intended learned operator family. Hard compatibility with fixed D_1 typically requires parameterizing D_0 inside ker(D_1), which can be global, basis-dependent, and restrictive for sparse or local operator families. In contrast, soft defect control preserves the original sparse or local parameterization and directly penalizes the specific algebraic failure mode D_1 D_0 not equal to 0.
>
> The experiments show the hierarchy. Hard compatibility provides the strongest structural guarantee. Soft defect regularization captures most of the stability benefit while remaining competitive in PDE accuracy. Unconstrained training is the regime that develops severe spectral and conditioning failures. For example, in the resolution-scaling experiment at N = 32 (Table 1), the condition number is approximately 1.73 x 10^4 for unconstrained training, but only approximately 2.47 x 10^2 under soft defect regularization and approximately 2.45 x 10^2 under hard compatibility. Under topology shift (Table 2), Betti accuracy is 0.71 for unconstrained training, 0.93 for soft defect control, and 0.98 for hard compatibility. We will make this hierarchy more explicit in revision: soft regularization is not meant to dominate hard constraints when hard constraints are easy to impose, but to provide a flexible structural control mechanism when exact parameterization is not the right modeling choice.

---

> > ### Author Rebuttal · Reviewer_imEx · 2026-04-03
> >
> > I thank the authors for their rebuttal, and they have addressed most of my questions. I would recommend that the authors move the discussion on soft vs hard constraints as well the discussion on the assumptions of the paper to the main paper.

---

> > > ### Author Response · Authors · 2026-04-03
> > >
> > > Thank you for your comment and for acknowledging our rebuttal.
> > >
> > > In response, we have dedicated a section in the main text to the discussion of soft versus hard approaches, moved some of the relevant content forward, and added a separate section addressing the underlying assumptions.
> > >
> > > We appreciate your helpful suggestions and we hope this has now addressed all of your concerns and suggestions.

---

### Decision · Program_Chairs · 2026-04-30

**Decision:**

Accept (regular)

**Comment:**

The paper was borderline with a consensus towards acceptance. All reviewers recognize the validity of the method.  Two reviewers gave weak accepts and found the contribution mathematically sound and the failure-prediction result compelling. One concern raised by the third reviewer is about the scope being too narrow and better suited to a scientific computing venue. While the application is fairly niche it would still be of interest to part of the ICML community, and the authors posted a strong rebuttal, addressing many of the concerns. Overall, I lean towards acceptance based on technical soundness, novelty and failure-prediction result. For the camera-ready version, please include the new results and responses to reviewers, and include a discussion on applicability.